# THE STABILITY AND CONVERGENCE OF TWO-TIMESCALE STOCHASTIC APPROXIMATION WITH MARKOVIAN NOISE FOR REINFORCEMENT LEARNING

## ABSTRACT

Stochastic approximations (SA)–algorithms which derive their power through the use of random, incremental updates–are at the heart of reinforcement learning (RL). Expanding the theory of SA has established rigorous results concerning the most important algorithms in RL, including stochastic gradient descent and temporal difference learning. In this work, we focus on two-timescale stochastic approximations, a class which notably includes temporal difference learning with gradient correction (TDC) and actor-critic methods. Prior work has developed stability (boundedness) and convergence criteria for two-timescale SA under i.i.d. noise, but analogous results for Markovian noise have remained elusive–a critical issue since RL data are generated by a Markov chain, making i.i.d. assumptions unrealistic. To address this gap, we present the first stability result and the first asymptotic convergence result for two-timescale schemes with Markovian noise under general, verifiable conditions–notably, without resorting to projected variants of the schemes or requiring the noise to be in a compact space. As a key application, we contribute the first asymptotic convergence proof of TDC, an off-policy prediction algorithm with linear approximation and eligibility traces. Together, our results extend SA theory, establishing the first theoretical foundation for analysis of two-timescale algorithms with the realistic noise models inherent to RL.

## 1 INTRODUCTION

The theory of stochastic approximation (SA, Robbins & Monro (1951); Benveniste et al. (1990); Kushner & Yin (2003); Borkar (2009)) concerns algorithms that recursively and randomly update a vector of parameters. Prominent SA algorithms include stochastic gradient descent (Kiefer & Wolfowitz, 1952) and temporal difference learning (Sutton, 1988). Many SA algorithms involve computations in nested loops, a structure central in reinforcement learning (RL, Sutton & Barto (2018)). For example, take actor-critic algorithms (Konda (2002)), where the inner loop (the critic) estimates the value of the agent's policy, and the outer loop (the actor) updates the policy. Such algorithms fall under the wide umbrella of *two-timescale* SA (Borkar (1997)). We represent algorithms with nested loops by updating two vectors of parameters: the *fast timescale* has large step sizes and acts as the inner loop, while the *slow timescale* has small step sizes and serves as the outer loop.

The community has long been concerned with proving the asymptotic convergence of SA schemes to ensure the practical reliability of many widely-used algorithms. A central challenge in establishing convergence is proving stability, i.e., that the iterates remain bounded, guaranteeing the algorithm will not diverge to infinity. Several works have addressed the convergence of two-timescale SA under independent, identically distributed (i.i.d.) noise, beginning with the Borkar-Meyn Theorem (Borkar, 1997; 2009; Tadic, 2004). All these works assume stability in their proofs of convergence. The challenge of proving stability has been addressed for the i.i.d. case, where the i.i.d. noise allows rewriting the terms as a Martingale difference sequence and the application of the Martingale Convergence Theorem (Borkar (2009), Lakshminarayanan & Bhatnagar (2017), Deb & Bhatnagar (2021)). Unfortunately, the applicability of such results to RL is quite limited, since RL algorithms are run on Markov chains, where the noise is state-dependent, and a very different approach from using the Martingale Convergence Theorem is required. Attempts have been made to establish stability and convergence results in the Markovian case (Karmakar & Bhatnagar, 2018; 2021; Panda & Bhatnagar,

2025), but they cannot be applied to many important RL algorithms–in particular, they are insufficient for off-policy algorithms with eligibility traces. By contrast, our results are general enough to cover this important class, including TDC with linear approximation and eligibility trace (Sutton et al., 2009; Yu, 2017). TDC is a gradient-based, state-of-the-art temporal difference (TD) algorithm for policy evaluation; it uses a gradient correction on a fast timescale to speed up convergence. It also converges where traditional TD methods diverge (when combining bootstrapping, function approximation, and off-policy learning–also known as the deadly triad (Baird, 1995; Sutton & Barto, 2018)).

Table 1 situates our work in the broader SA literature, highlighting the crucial missing piece: stability and convergence guarantees for two-timescale algorithms under Markovian noise.

Table 1: Comparison of our work's stability results to those of existing works

|  | i.i.d. Noise | Markovian Noise |
|---|---|---|
| Single-Timescale | Borkar (2009) | Liu et al. (2025) |
| Two-Timescale | Lakshminarayanan & Bhatnagar (2017) | **This Work** |

We address this gap with three significant contributions:

1. We provide the first set of criteria required to show the stability of two-timescale stochastic approximation schemes with Markovian noise, without the use of projections (Theorem 2).

2. We use the stability to provide the first proof of the convergence of two-timescale schemes with Markovian noise in uncountable, unbounded spaces (Corollary 1).

3. We provide the first proof of TDC with eligibility trace, an important off-policy RL algorithm, as an application of our results (Theorem 3).

**Technical Innovation.** We use the well-established ODE method, which has resulted in many advances in SA (Borkar, 2009; Liu et al., 2025). The key technical innovation of our work is that we connect the timescales by showing that the fast iterates are bounded by *the maximal slow iterate seen previously* (see Theorem 1). Our methodology is a novel contribution to the literature, as no prior work—focused either on stability (Kushner & Yin, 2003; Lakshminarayanan & Bhatnagar, 2017; Yaji & Bhatnagar, 2016; Panda & Bhatnagar, 2025) or other two-timescale results (Mokkadem & Pelletier, 2006; Dalal et al., 2018; Doan, 2021; Zeng et al., 2024)—has applied it to the ODE method since its inception more than two decades ago. To connect the timescales, those prior works have either taken stronger assumptions or have been forced to show that the iterates of one timescale bound the other, while we are able to relax this to just being bound by the maximal slow iterate seen so far. The Markovian two-timescale case itself necessitates this method as, unlike in the single-timescale case addressed by Liu et al. (2025), we need a rescaling scheme that uses the maximal scaling factor seen thus far, so Theorem 1 tracks the maximal slow iterate previously encountered. This necessarily engenders many creative divergences in our approach when compared to prior work. These include the necessity of showing that the discretization error diminishes along the entire sequence (unlike in Liu et al. (2025) where it only diminishes along a subsequence), a unique definition of the scaling factor in the slow timescale, and some creative approaches of analyzing the ODEs to link the timescales (see Lemmas 12.8 to 12.10 and 6.1).

In totality, our results establish the first general and rigorous theoretical foundation for two-timescale SA algorithms under realistic noise–closing a long-standing gap in RL theory, securing the convergence of widely used RL methods, and contributing a methodological novelty back to the community.

## 2 RELATED WORK

We divide this section into two parts: the works focusing on stability–ensuring the boundedness of the iterates–and those focusing on convergence–that the iterates converge in the limit almost surely.

**Stability Results.** Since stability is difficult to verify but necessary for convergence, many works assume stability in their proofs (Borkar, 1997; 2009; 2024; Tadic, 2004; Karmakar & Bhatnagar,

2018). However, it is possible to prove stability with verifiable criteria–see Chapter 3 of Borkar (2009) for the single-timescale i.i.d. case, Lakshminarayanan & Bhatnagar (2017) for the two-timescale i.i.d. case, and Liu et al. (2025) for the single-timescale Markovian case (visualized in Table 1). *We extend this to the two-timescale Markovian case, which has not been done before.*

Another strategy is to project the SA iterates to ensure their boundedness, e.g., in Yu (2017), which can prevent convergence to the true solution (see Section 5.4 of Borkar (2009) for details about the problems inherent to projected schemes). *Our work does not require the SA scheme to have projections, ensuring that the algorithm in our application converges to the true optimum.* In Karmakar & Bhatnagar (2021), the authors generalize to a multi-timescale Markovian problem. However, in their formulation, while the slower timescale affects the faster timescale, the faster timescale does not impact the slower timescale. *Our formulation is more general, as we maintain the coupled dynamics key to two-timescale SA–both the slow and fast iterates impact each other.*

**Asymptotic Convergence Results.** The convergence of two-timescale SA under i.i.d. noise is well-established; see Chapter 6 of Borkar (2009). While attempts have been made to extend the theory to Markovian noise, no prior work is general enough to be suitable for our purposes. Karmakar & Bhatnagar (2018) establish convergence for Markovian noise, but requires the noise to be in a compact space, limiting the applicability. *In contrast, our results allow the noise to evolve in an uncountable, non-compact space, a necessity for application to off-policy RL algorithms with eligibility traces.* Panda & Bhatnagar (2025) attempt to show stability and convergence, but they require a projection on the fast timescale. *In comparison, our work does not project the iterates.*

Prior work covers i.i.d. or uncoupled cases, but no existing result establishes stability or convergence for general two-timescale SA with coupled dynamics under Markovian noise. Our work fills this gap.

## 3 BACKGROUND

In this work, we study the general two-timescale stochastic approximation scheme: given an initial $x_0 \in \mathbb{R}^{d_1}$ and $y_0 \in \mathbb{R}^{d_2}$, we recursively generate sequences of vectors $\{x_n\}$ and $\{y_n\}$ by

$$x_{n+1} = x_n + \alpha(n)H(x_n, y_n, W_{n+1}) \tag{1}$$
$$y_{n+1} = y_n + \beta(n)G(x_n, y_n, W_{n+1}) \tag{2}$$

We also define for convenience the concatenation $z_n \doteq (x_n, y_n)$. Each $z_n$ is an element of $\mathbb{R}^{d_1+d_2}$.

In this scheme, $\{\alpha(n)\}_{n=0}^{\infty}$ and $\{\beta(n)\}_{n=0}^{\infty}$ are sequences of deterministic learning rates and $\{W_n\}_{n=1}^{\infty}$ is a sequence of random noise in a general space $\mathcal{W}$ (not necessarily compact), while $H : \mathbb{R}^{d_1+d_2} \times \mathcal{W} \to \mathbb{R}^{d_1}$ is a function that maps the current iterate $z_n$ and noise $W_{n+1}$ to the actual incremental update of $x$ (the fast timescale iterate) and $G : \mathbb{R}^{d_1+d_2} \times \mathcal{W} \to \mathbb{R}^{d_2}$ maps $z_n$ and $W_{n+1}$ to the incremental update of $y$ (the slow timescale iterate).

This is a two-timescale scheme since the stepsizes satisfy $\lim_{n\to\infty} \frac{\beta(n)}{\alpha(n)} = 0$, which is the mechanism that produces the same effect as a nested loop. Thus, since we can consider $y_n$ to remain almost constant relative to $x_n$ for large $n$ in the fast timescale, the behavior of $x_n$ can be studied by analyzing the ordinary differential equation (ODE)

$$\frac{\mathrm{d}x(t)}{\mathrm{d}t} = h(x(t), y) \tag{3}$$

where $y$ is a fixed constant and $h(x, y) \doteq \mathbb{E}[H(x, y, \omega)]$ (with respect to a specific probability distribution). In the slow timescale, we can consider $x_n$ to have already converged to a value that depends on $y_n$, so we also study the ODE

$$\frac{\mathrm{d}y(t)}{\mathrm{d}t} = g(\lambda(y(t)), y(t)) \tag{4}$$

where $\lambda(y)$ denotes the equilibrium of (3) (i.e., the $x$ such that $h(x, y) = 0$ for fixed $y$) and $g(x, y) \doteq \mathbb{E}[G(x, y, \omega)]$. The asymptotic behavior of the discrete, stochastic iterates $\{z_n\}$ can then be characterized by the continuous, deterministic trajectories of the ODEs (3) and (4)–this is called

the ODE method (see Borkar (2009) for a comprehensive background). For this ODE approximation to be valid, the iterates must be bounded; this is known as *stability*. One needs to show

$$\sup_n \|z_n\| < \infty \quad \text{a.s.,}$$

which is widely considered challenging (Borkar (2009)), but stability is a critical prerequisite for convergence and many works in the realm of two-timescale stochastic approximation rely on it (Tadic, 2004; Borkar, 2009; Karmakar & Bhatnagar, 2018; Borkar, 2024; Hu et al., 2024; Borkar, 2025).

# 4 MAIN RESULTS

We begin by giving a structural overview of the rest of the paper. In this section, we will describe our assumptions and present our main results on stability and convergence of two-timescale algorithms. In Section 5, we give a detailed overview of the proof of Theorem 1, which states that the current fast timescale iterate is bounded by the largest slow timescale iterate seen previously. This proof is a fast timescale analysis. In Section 6, we give an overview of the proof of Theorem 2, which establishes stability (boundedness) over all the iterates with probability one–this is a slow timescale analysis. We then utilize the stability results to prove Corollary 1, which shows the convergence of two-timescale SA algorithms under Markovian noise, in Section 7. Finally, in Section 8, we apply our results to prove the convergence of the off-policy RL algorithm TDC with eligibility trace.

**Assumptions.** We now briefly discuss our assumptions (the full details are in Appendix 11). First, we assume that our Markov chain $\{W_n\}$ has a unique stationary distribution, a standard assumption in RL. We also assume our learning rates follow standard SA conditions such as square-summability. Importantly, since we are analyzing a two-timescale scheme, we have $\lim_{n \to \infty} \frac{\beta(n)}{\alpha(n)} = 0$.

Now, we make assumptions about the functions $H$ and $G$. For any $c \in [1, \infty)$, define

$$H_c(x, y, w) \doteq \frac{H(cx, cy, w)}{c}, \quad G_c(x, y, w) \doteq \frac{G(cx, cy, w)}{c}.$$

The functions $H_c$ and $G_c$ are rescaled versions of the functions $H$ and $G$ and will be used to construct rescaled iterates, a key technique in the ODE method (see, e.g., Borkar & Meyn (2000); Borkar (2009); Liu et al. (2025)). Just as in those works, we need the existence of some sort of limiting functions $H_\infty$ and $G_\infty$ for $H_c$ and $G_c$ respectively, when $c \to \infty$. We take $H_c, G_c, H_\infty, G_\infty$ to be Lipschitz–so that their growth is well-characterized and to guarantee existence and uniqueness of the ODEs of interest–and define their respective expectations $h_c, g_c, h_\infty, g_\infty$ with respect to the stationary distribution of the Markov chain.

We now define the limiting ODEs–these are central to the ODE method:

$$\frac{dx(t)}{dt} = h_\infty(x(t), y), \quad \frac{dy(t)}{dt} = g_\infty(\lambda_\infty(y(t)), y(t))$$

which have the unique globally asymptotically stable equilibria $\lambda_\infty(y)$ (where $\lambda_\infty : \mathbb{R}^{d_2} \to \mathbb{R}^{d_1}$ is a homogeneous Lipschitz map with $\lambda_\infty(0) = 0$) and 0 respectively.

These assumptions are mild, minimal, and well-supported in literature that studies stability of SA (Borkar, 2009; Lakshminarayanan & Bhatnagar, 2017; Liu et al., 2025). Most importantly, we can easily verify that important RL algorithms like TDC satisfy them rather than simply assuming that stability holds, so our results are widely applicable to general methods.

We present our first result, which is a *key methodological innovation*:

**Theorem 1.** *Let the Assumptions in Appendix 11 hold. There exists a constant $K$ such that for all $n$,*

$$\|x_n\| \le K(1 + \|y_n^{max}\|) \quad a.s.$$

*where $y_n^{max} \doteq y_m$ such that $m = \arg\max_{i \le n} \|y_i\|$.*

We discuss its proof in Section 12. Intuitively, this result tells us that the size of the fast timescale iterates is bounded by the largest slow timescale iterate seen thus far–in the long history of the ODE

method, no previous work has attempted to establish and use a result bounding one timescale by the maximum of the other to tie the timescales together (see the introduction for more detail). Note that we can recover the main theorem of Liu et al. (2025) from Theorem 1 by setting all $y_n$ to 0 (then the scheme is identical to the single-timescale case). Having established the slow timescale iterates control the fast timescale iterates, we prove stability:

**Theorem 2.** *The iterates $\{z_n\}$ are stable, i.e.,*

$$\sup_n \|z_n\| < \infty \quad a.s.$$

See Section 6 for a discussion of the proof. Once the boundedness of the iterates has been established, convergence can be shown. We can now guarantee that the iterates converge to a *bounded, nonempty* invariant set since the iterates evolve in a compact set, ensuring the existence of subsequential limits.

**Corollary 1.** *Let Assumptions 1 - 6 hold. Then the iterates $\{z_n\}$ generated by (1) converge almost surely to a (sample path dependent) bounded invariant set of the ODE system*

$$\frac{\mathrm{d}x(t)}{\mathrm{d}t} = h(x(t), y(t)), \quad \frac{\mathrm{d}y(t)}{\mathrm{d}t} = 0 \tag{5}$$

*and the iterates $\{y_n\}$ generated by (1) converge almost surely to a (sample path dependent) bounded invariant set of the ODE*

$$\frac{\mathrm{d}y(t)}{\mathrm{d}t} = g(\lambda(y(t)), y(t)). \tag{6}$$

The discussion of the proof is in Section 7. As with stability, we first establish a convergence result for the fast timescale and then the slow timescale separately. This type of result is the goal of the ODE method–Corollary 1 shows convergence to bounded invariant sets, and the assumptions we place on the ODEs (in particular, the existence of unique globally asymptotically stable equilibria) characterize the invariant set. In particular, since (6) has a unique globally asymptotically stable equilibrium, its invariant set is the singleton set $\{y^*\}$, ensuring convergence to the optimal value.

## 5 OVERVIEW OF PROOF OF THEOREM 1

In this section, we will give a discussion of the proof of Theorem 1–see Appendix 12 for the full details. To prove Theorem 1, we conduct analysis in the fast timescale. This allows us to treat the slow timescale iterates as almost constant relative to the fast iterates. Note that the lemmas are derived on an arbitrary sample path $\{x_0, y_0, \{W_i\}_{i=1}^\infty\}$ such that the assumptions in Section 4 hold, so we omit "$a.s.$" from the lemma statements for conciseness.

**Setting up the Timeline.** We begin by splitting the positive real axis $[0, \infty)$ into chunks of length $\{\alpha(i)\}_{i=0,1,\dots}$. We then collect these segments together into larger intervals $\{[T_n, T_{n+1})\}_{n=0,1,\dots}$, where the sequence $\{T_n\}$ has the property that as $n$ grows large, we have $T_{n+1} - T_n \approx T$. We define

$$t(0) \doteq 0, \quad t(n) \doteq \sum_{i=0}^{n-1} \alpha(i) \quad n = 1, 2, \dots$$

For all $T > 0$, define

$$m(T) = \max\{i | T \geq t(i)\}$$

to be the maximal $i$ where $t(i)$ is no greater than $T$. We now define

$$T_0 = 0, \quad T_{n+1} = t(m(T_n + T) + 1).$$

Intuitively, $T_{n+1}$ is a little to the right of $T_n + T$ on the real axis. As $n$ grows large, the interval $[T_n, T_{n+1})$ is able to contain more time steps as $\alpha(i)$ is decreasing to 0.

**Defining the Scaled Iterates.** We start by fixing a sample path $\{x_0, y_0, \{W_i\}_{i=1}^{\infty}\}$. We take $\bar{z}(t)$ to be the piecewise constant interpolation of $z_n$ at points $\{t(n)\}_{n=0,1,\ldots}$, i.e.,

$$\bar{z}(t) \doteq (\bar{x}(t), \bar{y}(t)) \doteq \begin{cases} (x_0, y_0) & t \in [0, t(1)) \\ (x_1, y_1) & t \in [t(1), t(2)) \\ (x_2, y_2) & t \in [t(2), t(3)) \\ \vdots \end{cases}$$

Now we describe our rescaling of $\bar{z}(t)$.

**Definition 1.** $\forall n \in \mathbb{N}, t \in [0, T+1)$, *define*

$$r_n \doteq \max\{1, r_{n-1}, \|\bar{z}(T_n)\|\}, \quad \tilde{z}_n(t) \doteq (\tilde{x}_n(t), \tilde{y}_n(t)) \doteq \frac{\bar{z}(T_n + t)}{r_n}.$$

This implies

$$\forall n \in \mathbb{N}, \|\tilde{z}_n(0)\| \leq 1.$$

Note that we depart from Liu et al. (2025) in our definition of the rescaling factor $r_n$. Through the definition, we ensure that the sequence $\{r_n\}$ is monotonic, which forces the (to be defined) ODE discretization error to diminish over the entire sequence rather than just a subsequence, which is necessary in the two-timescale case.

We can regard $\tilde{z}_n(t)$ as the Euler discretization of $z_n(t)$ defined below.

**Definition 2.** $\forall n \in \mathbb{N}, t \in [0, T+1)$, *define* $z_n(t) = (x_n(t), y_n(t))$ *as the solution to the ODE system*

$$\frac{dx_n(t)}{dt} = h_{r_n}(x_n(t), y_n(t)), \quad \frac{dy_n(t)}{dt} = 0$$

*with initial condition* $z_n(0) = \tilde{z}_n(0)$.

We want to show that the error of the discretization diminishes asymptotically. Precisely speaking, the discretization error is defined as $f_n(t) \doteq \tilde{z}_n(t) - z_n(t)$, and we want to show that $f_n(t)$ diminishes to 0 uniformly in $t$ as $n \to \infty$. So, we need to analyze the following three sequences of functions:

$$\{t \mapsto \tilde{z}_n(t)\}_{n=0}^{\infty}, \{z_n(t)\}_{n=0}^{\infty}, \{f_n(t)\}_{n=0}^{\infty}.$$

In particular, we show that they are all *equicontinuous in the extended sense*, as this is required to apply the Arzelà-Ascoli theorem (Appendix 10.4). We defer the relevant definitions, statements, and proofs to the Appendix.

**A Convergent Subsequence.** To prove stability, we need to show that $\sup_n \|z_n\| < \infty$. Observe the inequality $\|z_{m(T_n)}\| = \|\bar{z}(T_n)\| \leq r_n$. Therefore, if we had $\sup_n r_n < \infty$, the result would come instantly. So, we assume that $\sup_n r_n = \infty$ to show that Theorem 1 holds even in this case. According to the Arzela-Ascoli theorem in the extended sense (Appendix 10.4), a sequence of equicontinuous functions always has a subsequence of functions that uniformly converges to a continuous limit. We use this to identify a subsequence of interest.

Note that in the following lemma, we mention taking an arbitrary subsequence as the first step– this sets us up to be able to apply Lemma 10.1 (a standard result from real analysis) later to get convergence along the entire sequence, which is something Liu et al. (2025) do not do.

**Lemma 5.1.** *Suppose* $\sup_n r_n = \infty$. *Take an arbitrary subsequence* $\{n_{k,0}\}_{k=0}^{\infty} \subseteq \{0, 1, 2, \ldots\}$. *Then there is a subsequence* $\{n_k\}_{k=0}^{\infty} \subseteq \{n_{k,0}\}_{k=0}^{\infty}$ *such that there exist some continuous functions* $f^{\lim}(t)$ *and* $\tilde{z}^{\lim}(t)$ *such that* $\forall t \in [0, T+1)$,

$$\lim_{k \to \infty} f_{n_k}(t) = f^{\lim}(t), \quad \lim_{k \to \infty} \tilde{z}_{n_k}(t) = \tilde{z}^{\lim}(t),$$

*where both convergences are uniform in $t$ on $[0, T+1)$. Furthermore, let $z^{\lim}(t) = (x^{\lim}(t), y^{\lim}(t))$ denote the unique solution to the ODE system*

$$\frac{dx^{\lim}(t)}{dt} = h_\infty(x^{\lim}(t), y^{\lim}(t)), \quad \frac{dy^{\lim}(t)}{dt} = 0$$

*with the initial condition $z^{\lim}(0) = \tilde{z}_n^{\lim}(0)$. Then $\forall t \in [0, T+1)$, we have*

$$\lim_{k \to \infty} z_{n_k}(t) = z^{\lim}(t),$$

*where the convergence is uniform in $t$ on $[0, T+1)$.*

Its proof is in Appendix 16.4. We will use the subsequence $\{n_k\}$ intensively.

**Diminishing Discretization Error.** We show that $\lim_{n \to \infty} \|f_n(t)\| = 0$ for all $t \in [0, T+1)$. We start by first proving that the discretization error diminishes along the sequence $\{n_k\}$, i.e., that

$$\lim_{k \to \infty} \|f_{n_k}(t)\| = \|f^{\lim}(t)\| = 0.$$

This means $\tilde{z}_{n_k}(t)$ is close to $z_{n_k}(t)$ as $k \to \infty$. The details of the argument are quite technical and many borrow from Liu et al. (2025). At the end of it all, from Lemma 16.9, we obtain $\lim_{k \to \infty} \|f_{n_k}(t)\| = 0$ uniformly in $t$ on $[0, T+1)$, so the discretization error goes to 0 along $\{n_k\}$.

Now, since we had chosen an arbitrary subsequence $\{f_{n_{k,0}}\}$ and it has a subsequence $\{f_{n_k}\}$ that converges to 0, by Lemma 10.1 we know that $\{f_n\}$ also converges to 0. Thus, the discretization error diminishes along the entire sequence. That is,

$$\lim_{k \to \infty} \|f_n(t)\| = 0$$

for all $t \in [0, T+1)$.

**ODEs with External Inputs.** We present some notation concerning ODEs with external inputs–this is useful since, in the limit, we can treat the slow iterates as constant external inputs.

**Definition 3.** *We use the notation $\eta_c^{y(t)}(t, x)$ to denote the solution to the ode*

$$\frac{dx(t)}{dt} = h_c(x(t), y(t))$$

*with the initial condition $x$.*

Note that under this notation, $x_n(t)$ can be written $\eta_{r_n}^{\tilde{y}_n(0)}(t, \tilde{x}_n(0))$. This notation (borrowed from Lakshminarayanan & Bhatnagar (2017)) is useful since it identifies a trajectory of an ODE parameterized by the rescaling factor $c$ and the external input $y(t)$.

The following lemma shows that within a certain amount of time, the trajectory of the ODE will be pulled close to the equilibrium determined by the external input.

**Lemma 5.2** (Lemma 1 from Chapter 3.2 of Borkar (2009)). *Let $K \subset \mathbb{R}^{d_1}$ be compact and fix $y \in \mathbb{R}^{d_2}$. Given any $\epsilon > 0$, there exists a $T_\epsilon > 0$ such that for all initial conditions $x \in K$, we have $\eta_\infty^y(t, x) \in B(\lambda_\infty(y), \epsilon)$[1] for all $t \geq T_\epsilon$.*

The next lemma shows that if the external input is close to some constant $y$, then the trajectory remains close to the trajectory we would obtain if $y$ were the external input.

**Lemma 5.3** (Lemma 2 from Chapter 3.2 of Borkar (2009)). *Let $y \in \mathbb{R}^{d_2}, [0, T]$ be a given time interval, and $\rho$ be a small positive constant with $y'(t) \in B(y, \rho)$. Then,*

$$\left\| \eta_c^{y'(t)}(t, x) - \eta_\infty^y(t, x) \right\| \leq (L\rho + \epsilon(c)) T e^{LT}.$$

*for any initial $x \in \mathbb{R}^{d_1}$ and for all $t \in [0, T]$.*

---

[1]The notation $B(x, r)$ denotes the open ball centered at $x$ with radius $r$.

**Completing the Proof.** The most involved result left is Lemma 5.4, which tells us that if the $x$ component is too much larger than the $y$ component, the norm of the iterate at $T_{n+1}$ cannot be larger than that of the iterate at $T_n$. We use the results about ODEs with external inputs to ensure that the $x$ component of the trajectory gets pulled to the equilibrium, which is close to 0.

**Lemma 5.4.** *There exists a $C_1 > 1$ such that if $\|\bar{x}(T_n)\| > C_1(1 + \|\bar{y}(T_n)\|)$, then $r_{n+1} = r_n$.*

Lemma 5.5 tells us that the iterate at each $T_n$ is bounded by the largest $y$ component seen so far in the sequence of times $T_n$. To do this, we use an inductive argument–intuitively, we combine the previous lemma, which prevents $z$ from growing if the $x$ component is too much larger than the $y$ component, with Lemma 17.15 which prevents $z$ from growing too much within one period $T$, ensuring that $x$ can never get too much larger than $y$.

**Lemma 5.5.** *There exists a constant $C_2$ such that for all $n$,*
$$\|\bar{z}(T_n)\| \leq C_1 C_2 (\max_{m \leq n} \|\bar{y}(T_m)\| + 1).$$

Finally, while we have ensured that for all $T_n$, $\bar{z}(T_n)$ is not too much larger than $\bar{y}(T_n)$, we need the result to also hold for the iterates that lie in between $T_n$ and $T_{n+1}$. Lemma 5.6 takes care of this.

**Lemma 5.6.** *There exists a constant $C_3$ such that for all $n$,*
$$\|x_n\| \leq C_1 C_2 C_3 (\|y_n^{max}\| + 1),$$

By setting $K$ equal to $C_1 C_2 C_3$, this concludes the proof of Theorem 1.

**Bridging the Gap.** We present the following result, showing that eventually, in the fast timescale, the fast iterates will track the equilibria $\lambda_\infty(\tilde{y}_n(t))$. In the fast timescale analysis, the slow iterates were regulated by the fact that the slow step sizes are small. However, in the slow timescale analysis, the fast step sizes will become very large. So, this result is necessary to provide a way to regulate the behavior of the fast iterates.

**Lemma 5.7.** *For any $\epsilon > 0$, there is some $T_\epsilon$ and some $N_\epsilon$ such that for all $n > N_\epsilon$, $\|\tilde{x}_n(t) - \lambda_\infty(\tilde{y}_n(t))\| \leq \epsilon$ for all $t \in [0, T_\epsilon]$.*

# 6 OVERVIEW OF PROOF OF THEOREM 2

In this section, we discuss the proof of Theorem 2. We now conduct analysis in the slow timescale, but the structure is very similar to the fast timescale analysis, so we only highlight the key differences. Note that we are still working with the same sample path $\{x_0, y_0, \{W_i\}_{i=1}^\infty\}$ from the last section.

We again split the positive real axis $[0, \infty)$, but this time into chunks of length $\{\beta(i)\}_{i=0,1,...}$ instead of $\{\alpha(i)\}_{i=0,1,...}$. We then redefine the scaled iterates for the slow timescale, except this time the scaling factor is defined differently, so that it will be at least as large as the largest iterate seen so far:
$$r_n \doteq \max\left\{1, \left\|z_{m(T_n)}^{max}\right\|\right\}.$$
where $z_n^{max} \doteq z_m$ such that $m = \arg\max_{i \leq n} \|z_i\|$. This way of defining the scaling factor in the slow timescale is another innovation. It ensures that at each time $T_n$, the iterate, when scaled by the slow timescale's $\{r_n\}$ values, is no larger when scaled by the appropriate fast timescale's $\{r_n\}$ values. This will be critical for linking the timescales in Lemma 6.1.

We can regard $\tilde{y}_n(t)$ as the Euler's discretization of $y_n(t)$ defined below.

**Definition 4.** $\forall n \in \mathbb{N}, t \in [0, T+1)$, *define $y_n(t)$ as the solution to the ODE*
$$\frac{dy_n(t)}{dt} = g_{r_n}(\lambda_\infty(y_n(t)), y_n(t))$$
*with initial condition $y_n(0) = \tilde{y}_n(0)$.*

The discretization error is then defined as $f_n(t) \doteq \tilde{y}_n(t) - y_n(t)$; just considering the slow component. Again, we show equicontinuity and apply Arzela-Ascoli to obtain a convergent subsequence.

To show that the discretization error diminishes, we must prove an involved result, requiring us to work in both timescales. We show that even in the slow timescale, the fast timescale iterates converge to the equilibria dependent on the slow timescale iterates:

**Lemma 6.1.** *For all $\epsilon > 0$, there exists some $N$ such that for all $n > N$,*

$$\|\tilde{x}_n(t) - \lambda_\infty(\tilde{y}_n(t))\| < \epsilon$$

*for all $t \in [0, T]$.*

With this result, we are able to deal with the error term (45) unique to the slow timescale, and show the discretization error decreases to zero (46). We then obtain ODE results similar to single-timescale stability results (Chapter 3 of Borkar (2009)). Since the limiting ODE is attracted to its globally asymptotically stable equilibrium, 0, our iterates cannot grow without bound. Thus, the sequence $r_n$ is bounded, creating a contradiction. We conclude that Theorem 2 holds true.

## 7    CONVERGENCE (PROOF OF COROLLARY 1)

In this section, we discuss the proof of Corollary 1. The proof mirrors the stability proof but is much simpler. We first show a convergence result in the fast timescale–namely, that the fast iterates $x_n$ converge to $\lambda(y_n)$. Then, we obtain the full convergence result through analysis in the slow timescale. The difference is that we no longer need to use rescaling, since we know that the iterates are bounded by Theorem 2, and we now prove results on $t \in (-\infty, \infty)$ as this makes it easier to show convergence to an invariant set.

Once we show that the discretization error diminishes, due to Theorem 2, we can argue that since the set of limit points $Z$ of $\{z_n\}$ is bounded and nonempty, it is invariant. We can then show that it is not possible for any subsequence of $\{z_n\}$ to converge to a point not in $Z$, so we know that $\{z_n\}$ converges to $Z$. Since the invariant set of (5) is $\{(\lambda(y), y) : y \in \mathbb{R}^{d_2}\}$, we have

$$\lim_{n \to \infty} \|x_n - \lambda(y_n)\| = 0.$$

We then pursue the slow timescale argument and show that $\{y_n\}$ converges to the invariant set of (6), which is the singleton containing $y^*$, the equilibrium of the map $\lambda$. This, combined with the fast timescale convergence result, yields our desired result:

$$\lim_{n \to \infty} \|z_n - (\lambda(y^*), y^*)\| = 0.$$

## 8    CONVERGENCE OF TDC WITH ELIGIBILITY TRACE

In this section, we prove the convergence of an important off-policy RL algorithm, TDC with eligibility trace, using our main results (Section 4). Vanilla TDC was first proposed in Sutton et al. (2008b) as a modification of gradient temporal difference learning (GTD) (Sutton et al., 2008a). GTD was developed to break the deadly triad–divergence that can arise when combining off-policy learning, function approximation, and bootstrapping, each of which are critical components in successful RL algorithms. While GTD mitigates the deadly triad, it is slow. TDC, on the other hand, is nearly as fast as regular TD learning and converges. It is also a two-timescale algorithm, as the gradient correction runs on a faster timescale. Although vanilla TDC is known to converge, the best prior work was only able to establish the convergence of projected variants of TDC with eligibility traces (Yu, 2017).

Eligibility traces are a powerful tool for credit assignment, a critical challenge in RL, and have been a fundamental part of RL since the inception of the field (Barto & Sutton, 1981). Although eligibility traces are useful, they introduce difficulties in analysis. Even if the state space of the Markov chain $\{(S_t, A_t)\}$ is finite, with eligibility traces, we must instead consider the chain $\{(S_t, A_t, e_t)\}$, which now evolves in an uncountable space. In the case of off-policy learning, the importance sampling ratio can cause the state space to be unbounded as well. Our results, therefore, are the first to be able to handle the important case of off-policy RL algorithms with eligibility traces. We demonstrate this with TDC, defined as follows:

**Definition 5.** *(TDC with Eligibility Trace)*

$$e_t = \lambda \gamma \rho_{t-1} e_{t-1} + \phi_t,$$
$$\delta_t = R_{t+1} + \gamma \phi_{t+1}^\top \theta_t - \phi_t^\top \theta_t,$$
$$\nu_{t+1} = \nu_t + \alpha_t \left( \rho_t \delta_t e_t - \phi_t \phi_t^\top \nu_t \right),$$
$$\theta_{t+1} = \theta_t + \beta_t \rho_t (\phi_t - \gamma \phi_{t+1}) e_t^\top \nu_t.$$

The eligibility trace $e_t$ is an exponential average of the linear features $\phi_t$ (weighted by importance sampling ratio $\rho_t$–necessary in off-policy learning), which represent the state $S_t$. $\delta_t$ is the TD error.

**Theorem 3.** *Take the assumptions in Appendix 15. Then, by applying Corollary 1, TDC with eligibility trace converges.*

The assumptions we take here are quite minimal. We assume the state space $S$ and the action space $A$ are finite, the Markov chain $\{S_t\}$ is irreducible, and that from any state, all possible actions have some positive probability of being chosen (since this is an off-policy algorithm, this ensures coverage of the target policy we are estimating the value of). These simple assumptions are well-established in the literature (Yu, 2017; Liu et al., 2025). From these basic assumptions, it is very easy to verify that TDC with eligibility trace satisfies all assumptions in Appendix 11, so Corollary 1 applies. By applying Corollary 1, we confirm that TDC with eligibility trace converges (see Appendix 15).

## 9 CONCLUSION

Stochastic approximation methods constitute a broad set of algorithms that recursively and randomly update a parameter vector. Of particular interest in reinforcement learning are two-timescale stochastic approximation methods, referring to those algorithms which involve nested loops, including actor-critic algorithms and TDC. Proving the asymptotic convergence of two-timescale stochastic approximation schemes provides rigorous guarantees of the reliability of deeply popular algorithms–for this, we need stability, which refers to the boundedness of the iterates. We prove the first stability and convergence results for general two-timescale stochastic approximation schemes under Markovian noise–critical to RL since the uncertainty in RL is Markovian. Our assumptions are minimally restrictive, allowing us to handle noise that evolved in an unbounded, uncountable space and the standard, unprojected two-timescale scheme. Our results, therefore, are broadly applicable to RL, as we enable analysis of critical off-policy RL algorithms with eligibility traces. We demonstrate this applicability by supplying the first proof of convergence of TDC with eligibility trace.

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

APPENDIX

## 10 MATHEMATICAL BACKGROUND

**Theorem 10.1** (**Gronwall Inequality**). *(Lemma 6 in Section 11.2 in* Borkar *(2009)) For a continuous function $u(\cdot) \geq 0$ and scalars $C, K, T \geq 0$,*

$$u(t) \leq C + K \int_0^t u(s)ds \quad \forall t \in [0, T]$$

*implies*

$$u(t) \leq Ce^{tK}, \forall t \in [0, T].$$

**Theorem 10.2** (**Gronwall Inequality in the Reverse Time**). *For a continuous function $u(\cdot) \geq 0$ and scalars $C, K, T \geq 0$,*

$$u(t) \leq C + K \int_t^0 u(s)ds \quad \forall t \in [-T, 0]$$

*implies*

$$u(t) \leq Ce^{-tK}, \forall t \in [-T, 0].$$

For a proof, see Appendix A.2 of Liu et al. (2025).

**Theorem 10.3** (**Discrete Gronwall Inequality**). *(Lemma 8 in Section 11.2 in* Borkar *(2009)) For nonnegative sequences $\{x_n, n \geq 0\}$ and $\{a_n, n \geq 0\}$ and scalars $C, L \geq 0$,*

$$x_{n+1} \leq C + L \sum_{i=0}^{n} a_i x_i \quad \forall n$$

*implies*

$$x_{n+1} \leq Ce^{L \sum_{i=0}^{n} a_i} \quad \forall n.$$

**Theorem 10.4** (The Arzelà-Ascoli Theorem in the Extended Sense on $[0, T)$). *Let $\{t \in [0, T) \mapsto g_n(t)\}$ be equicontinuous in the extended sense. Then, there exists a subsequence $\{g_{n_k}(t)\}$ that converges to some continuous limit $g^{\lim}(t)$, uniformly in $t$ on $[0, T)$.*

For a proof, see Appendix A.4 of Liu et al. (2025).

**Theorem 10.5** (Moore-Osgood Theorem for Interchanging Limits). *If $\lim_{n \to \infty} a_{n,m} = b_m$ uniformly in $m$ and $\lim_{m \to \infty} a_{n,m} = c_n$ for each large $n$, then both $\lim_{m \to \infty} b_m$ and $\lim_{n \to \infty} c_n$ exists and are equal to the double limit, i.e.,*

$$\lim_{m \to \infty} \lim_{n \to \infty} a_{n,m} = \lim_{n \to \infty} \lim_{m \to \infty} a_{n,m} = \lim_{\substack{n \to \infty \\ m \to \infty}} a_{n,m}.$$

**Lemma 10.1** (Sub-subsequence Lemma). *Let $x_n$ be a sequence in some metric space. If every subsequence $x_{n_k}$ itself has a subsequence $x_{n_{k_j}}$ that converges to the same limit $x$, then $x_n$ converges to $x$.*

*Proof.* Suppose we have a sequence $x_n$ where every subsequence $x_{n_k}$ itself has a subsequence $x_{n_{k_j}}$ that converges to the same limit $x$. For contradiction, assume that $x_n$ does not converges to $x$, so there is a subsequence $x_{n_{k,0}}$ that is always at least some distance $\epsilon$ away from $x$. By assumption, $x_{n_{k,0}}$ has a subsequence $x_{n_{k,1}}$ that converges to $x$, which contradicts that $x_{n_{k,0}}$ is always at least some $\epsilon$ away from $x$. So, it must be that $x_n$ converges to $x$. $\square$

## 11 MAIN ASSUMPTIONS

**Assumption 1.** *The Markov chain $\{W_n\}$ has a unique invariant probability measure (i.e., stationary distribution), denoted by $d_{\mathcal{W}}$.*

Although the uniqueness and even the existence of the invariant probability measure can be relaxed, we use Assumption 1 for simplification. Additionally, due to the way updates are defined ((1) and (2)), we start counting $\{W_n\}$ from $n = 1$.

**Assumption 2.** *The learning rates $\{\alpha(i)\}, \{\beta(i)\}$ are positive, decreasing, and satisfy*

$$\sum_{i=0}^{\infty} \alpha(i) = \sum_{i=0}^{\infty} \beta(i) = \infty,$$

$$\lim_{i \to \infty} \alpha(i) = \lim_{i \to \infty} \beta(i) = 0$$

$$\frac{\alpha(i) - \alpha(i+1)}{\alpha(i)} = \mathcal{O}\left(\alpha(i)\right), \tag{7}$$

$$\frac{\beta(i) - \beta(i+1)}{\beta(i)} = \mathcal{O}\left(\beta(i)\right),$$

$$\lim_{i \to \infty} \frac{\beta(i)}{\alpha(i)} = 0. \tag{8}$$

These conditions on the learning rates are quite common in stochastic approximation. The last condition is necessary for the two-timescale formulation we are using.

**Remark 1.** *For any $\alpha(n) = \frac{B_1}{(n+B_2)^{\gamma_\alpha}}, \beta(n) = \frac{B_3}{(n+B_4)^{\gamma_\beta}}$ with $\gamma_\alpha, \gamma_\beta \in (0.5, 1], \gamma_\beta > \gamma_\alpha$, and all $B_i$ positive, one can verify that all the conditions in Assumption 2 are satisfied.*

Now, we make assumptions about the functions $H$ and $G$. For any $c \in [1, \infty)$, define

$$H_c(x, y, w) \doteq \frac{H(cx, cy, w)}{c} \tag{9}$$

$$G_c(x, y, w) \doteq \frac{G(cx, cy, w)}{c}. \tag{10}$$

The functions $H_c$ and $G_c$ are rescaled versions of the functions $H$ and $G$ and will be used to construct rescaled iterates, a key technique in the ODE method (see, e.g., Borkar & Meyn (2000); Borkar (2009); Liu et al. (2025)). Just as in those works, we need the existence of some sort of limiting functions for $H_c$ and $G_c$ when $c \to \infty$.

**Assumption 3.** *There exist measurable functions $H_\infty(x, y, w)$ and $G_\infty(x, y, w)$, functions $\kappa_H(c), \kappa_G(c) : \mathbb{R} \to \mathbb{R}$, and measurable functions $b_H(x, y, w), b_G(x, y, w)$ such that for all $x, y, w$*

$$H_c(x, y, w) - H_\infty(x, y, w) = \kappa_H(c) b_H(x, y, w) \tag{11}$$

$$G_c(x, y, w) - G_\infty(x, y, w) = \kappa_G(c) b_G(x, y, w)$$

$$\lim_{c \to \infty} \kappa_H(c) = \lim_{c \to \infty} \kappa_G(c) = 0$$

*There also exists a measurable function $L_b(w)$ such that for all $x, x', y, y', w$*

$$\|b_H(x, y, w) - b_H(x', y', w)\| \le L_b(w)\|(x, y) - (x', y')\|,$$

$$\|b_G(x, y, w) - b_G(x', y', w)\| \le L_b(w)\|(x, y) - (x', y')\|.$$

*Additionally, the expectation $L_b \doteq \mathbb{E}_{w \sim d_\mathcal{W}}[L_b(w)]$ is well-defined and finite.*

Assumption 3 provides details on how $H_c$ and $G_c$ converge to $H_\infty$ and $G_\infty$ when $c \to \infty$.

We now assume that the functions $H_c$ and $G_c$ are Lipschitz. This will guarantee that the corresponding ODEs exist and are unique.

**Assumption 4.** *There exists a measurable function $L(w)$ such that for any $x, x', y, y', w$,*

$$\|H(x, y, w) - H(x', y', w)\| \le L(w)\|(x, y) - (x', y')\|, \tag{12}$$

$$\|H_\infty(x, y, w) - H_\infty(x', y', w)\| \le L(w)\|(x, y) - (x', y')\|, \tag{13}$$

$$\|G(x, y, w) - G(x', y', w)\| \le L(w)\|(x, y) - (x', y')\|,$$

$$\|G_\infty(x, y, w) - G_\infty(x', y', w)\| \le L(w)\|(x, y) - (x', y')\|.$$

*Moreover, the following expectations are well-defined and finite for any $x, y$:*

$$h(x, y) \doteq \mathbb{E}_{w \sim d_{\mathcal{W}}}[H(x, y, w)],$$

$$h_\infty(x, y) \doteq \mathbb{E}_{w \sim d_{\mathcal{W}}}[H_\infty(x, y, w)],$$

$$g(x, y) \doteq \mathbb{E}_{w \sim d_{\mathcal{W}}}[G(x, y, w)],$$

$$g_\infty(x, y) \doteq \mathbb{E}_{w \sim d_{\mathcal{W}}}[G_\infty(x, y, w)],$$

$$L \doteq \mathbb{E}_{w \sim d_{\mathcal{W}}}[L(w)].$$

The functions $x, y \mapsto H_c(x, y, w)$ and $x, y \mapsto G_c(x, y, w)$ share the same Lipschitz constant $L(w)$ as the functions $x, y \mapsto H(x, y, w)$ and $x, y \mapsto G(x, y, w)$. Similarly to (9) and (10), we define

$$h_c(x) \doteq \frac{h(cx, cy)}{c},$$

$$g_c(x) \doteq \frac{g(cx, cy)}{c}.$$

The following assumption is necessary to ensure that the the rescaled iterates can converge to the trajectory of the limiting ODE.

**Assumption 5.** *As $c \to \infty$, $h_c(x, y)$ converges to $h_\infty(x, y)$ uniformly in $(x, y)$ on any compact subsets of $\mathbb{R}^{d_1 + d_2}$. The limiting ODE*

$$\frac{dx(t)}{dt} = h_\infty(x(t), y)$$

*has a unique globally asymptotically stable equilibrium $\lambda_\infty(y)$ where $\lambda_\infty : \mathbb{R}^{d_2} \to \mathbb{R}^{d_1}$ is a Lipschitz map. $\lambda_\infty$ is homogeneous, i.e., $\lambda_\infty(cy) = c\lambda_\infty(y)$. And $\lambda_\infty(0) = 0$.*

*As $c \to \infty$, $g_c(x, y)$ converges uniformly to $g_\infty(x, y)$ on compact subsets of $\mathbb{R}^{d_1 + d_2}$. The limiting ODE*

$$\frac{dy(t)}{dt} = g_\infty(\lambda_\infty(y(t)), y(t))$$

*has 0 as its unique globally asymptotically stable equilibrium.*

The map $\lambda_\infty$ tells us, for a particular external input $y$, what $x$ (the parameters of the inner loop) should converge to. This idea and the notation for it come from Chapter 6 of Borkar (2009).

**Assumption 6.** *Let $\gamma$ denote any of the following functions:*

$$w \mapsto H(x, y, w) \quad (\forall x, y),$$

$$w \mapsto G(x, y, w) \quad (\forall x, y),$$

$$w \mapsto L_b(w),$$

$$w \mapsto L(w).$$

*We have, for any initial condition $W_1$,*

$$\lim_{n \to \infty} \alpha(n) \sum_{i=1}^{n} (\gamma(W_i) - \mathbb{E}_{w \sim d_{\mathcal{W}}}[\gamma(w)]) = 0 \quad a.s.$$

$$\lim_{n \to \infty} \beta(n) \sum_{i=1}^{n} (\gamma(W_i) - \mathbb{E}_{w \sim d_{\mathcal{W}}}[\gamma(w)]) = 0 \quad a.s. \tag{14}$$

**Remark 2.** *Once more, consider $\beta(n) = \frac{B_1}{(n+B_2)^{\gamma_\beta}}$ as an example. For $\gamma_\beta = 1$, (14) is implied by the following Law of Large Numbers (LLN)*

$$\lim_{n \to \infty} \frac{1}{n} \sum_{i=1}^{n} (\gamma(W_i) - \mathbb{E}_{w \sim d_{\mathcal{W}}}[\gamma(w)]) = 0 \quad a.s. \tag{LLN}$$

*For $\gamma_\beta \in (0.5, 1]$, (14) is implied by the following Law of the Iterated Logarithm (LIL)*

$$\left\| \sum_{i=1}^{n} (\gamma(W_n) - \mathbb{E}_{w \sim d_{\mathcal{W}}}[\gamma(w)]) \right\| \leq \zeta \sqrt{n \log \log n} \quad a.s., \tag{LIL}$$

*where $\zeta$ is a sample path dependent finite constant.*

## 12 PROOF OF THEOREM 1

This is a more detailed version of Section 5. In this section, we will prove Theorem 1. Section 12.1 sets up the notation and explains our fast timescale analysis. Section 12.2 defines the rescaled iterates and some important functions. Section 12.3 assumes for contradiction that stability does not hold and identifies a resulting subsequence of interest. Section 12.4 demonstrates convergence along this subsequence and uses this to show that the convergence holds for the entire sequence. Section 12.5 defines some notation and provides some results about ODEs with external inputs. Finally, Section 12.6 uses results from the previous sections to complete the proof. Lemmas in this section are derived on an arbitrary sample path $\{x_0, y_0, \{W_i\}_{i=1}^{\infty}\}$ such that the assumptions in Section 4 hold. Thus, we omit "$a.s.$" from the lemma statements for conciseness.

### 12.1 SPLITTING UP THE TIMELINE

Here, we perform a fast timescale analysis. First, we split the positive real axis $[0, \infty)$ into chunks of length $\{\alpha(i)\}_{i=0,1,\ldots}$. We then collect these segments together into larger intervals $\{[T_n, T_{n+1})\}_{n=0,1,\ldots}$, where the sequence $\{T_n\}$ has the property that as $n$ grows large, we have $T_{n+1} - T_n \approx T$. We define

$$t(0) \doteq 0,$$

$$t(n) \doteq \sum_{i=0}^{n-1} \alpha(i) \quad n = 1, 2, \ldots \quad .$$

For all $T > 0$, define

$$m(T) = \max\{i | T \geq t(i)\} \tag{15}$$

to be the maximal $i$ where $t(i)$ is no greater than $T$. To visualize, $t(m(T))$ is a bit to the left of $T$ on the real axis. So, $t(m(T))$ satisfies the following:

$$t(m(T)) \leq T < t(m(T) + 1) = t(m(T)) + \alpha(m(T)),$$
$$t(m(T)) > T - \alpha(m(T)).$$

Define

$$T_0 = 0,$$
$$T_{n+1} = t(m(T_n + T) + 1).$$

Intuitively, $T_{n+1}$ is a little to the right of $T_n + T$ on the real axis. We define

$$\alpha(i) = \beta(i) = 0 \quad \forall i < 0,$$
$$m(t) = 0 \quad \forall t \leq 0, \tag{16}$$

for simplifying notations. For any given function $f$ with domain $\mathcal{W}$, its asymptotic rate of change is defined as

$$\limsup_n \sup_{-\tau \leq t_1 \leq t_2 \leq \tau} \left\| \sum_{i=m(t(n)+t_1)}^{m(t(n)+t_2)-1} \alpha(i)[f(W_{i+1}) - \mathbb{E}_{w \sim d_{\mathcal{W}}}[f(w)]] \right\|.$$

This asymptotic rate of change helps us to describe the asymptotic regularity of $\{f(W_n)\}$ and we lean on its usefulness in studying stochastic approximation. We refer the reader to Sections 5.3.2 and 6.2 of Kushner & Yin (2003) for a detailed exposition of this tool. In this work, we have deferred the contents concerning asymptotic rate of change to Appendices 16.1 and 16.2 since the statements and proofs are very similar to results in Liu et al. (2025).

### 12.2 DEFINING THE SCALED ITERATES

We start by fixing a sample path $\{x_0, y_0, \{W_i\}_{i=1}^{\infty}\}$. We will take $\bar{z}(t)$ to be the piecewise constant interpolation[2] of $z_n$ at points $\{t(n)\}_{n=0,1,\ldots}$, i.e.,

---

[2]It also works if we consider a piecewise linear interpolation following Borkar (2009). The piecewise linear interpolation, however, will significantly complicate the presentation. We, therefore, follow Kushner & Yin (2003) and Liu et al. (2025) and use the piecewise constant interpolation.

$$\bar{z}(t) \doteq (\bar{x}(t), \bar{y}(t)) \doteq \begin{cases} (x_0, y_0) & t \in [0, t(1)) \\ (x_1, y_1) & t \in [t(1), t(2)) \\ (x_2, y_2) & t \in [t(2), t(3)) \\ \vdots \end{cases}$$

By (15), we also have

$$\bar{z}(t) \doteq z_{m(t)}. \tag{17}$$

By (1), $\forall n \geq 0$, we have

$$\bar{z}(t(n+1)) = \bar{z}(t(n)) + (\alpha(n)H(\bar{z}(t(n)), W_{n+1}), \beta(n)G(\bar{z}(t(n)), W_{n+1})).$$

Now we describe our rescaling of $\bar{z}(t)$.

**Definition 6.** $\forall n \in \mathbb{N}, t \in [0, T+1)$, *define*

$$\tilde{z}_n(t) \doteq (\tilde{x}_n(t), \tilde{y}_n(t)) \doteq \frac{\bar{z}(T_n + t)}{r_n}$$

*where*

$$r_n \doteq \max\{1, r_{n-1}, \|\bar{z}(T_n)\|\}. \tag{18}$$

This implies

$$\forall n \in \mathbb{N}, \|\tilde{z}_n(0)\| \leq 1. \tag{19}$$

Moreover[3],

$\forall n \in \mathbb{N}, t \in [0, T+1)$,

$$\tilde{z}_n(t) = \frac{\bar{z}(T_n) + \sum_{i=m(T_n)}^{m(T_n+t)-1}(\alpha(i)H(\bar{z}(t(i)), W_{i+1}), \beta(i)G(\bar{z}(t(i)), W_{i+1}))}{r_n}$$

$$= \tilde{z}_n(0) + \sum_{i=m(T_n)}^{m(T_n+t)-1}(\alpha(i)H_{r_n}(\tilde{z}_n(t(i) - T_n), W_{i+1}), \beta(i)G_{r_n}(\tilde{z}_n(t(i) - T_n), W_{i+1})).$$

**Remark 3.** *Note that we depart from Liu et al. (2025) in our definition of the rescaling factor $r_n$ (18). Through the definition, we ensure that the sequence $\{r_n\}$ is monotonic, which enables us to obtain convergence over the entire sequence rather than just a subsequence.*

*We also depart from prior works by defining the sequence of rescaled functions $\{t \mapsto \tilde{z}_n(t)\}$ which share the domain $[0, T+1)$, as opposed to rescaling the function $\bar{z}(t)$ directly (often denoted as $\hat{z}$). This consistent, larger domain greatly simplifies our arguments, while prior works must handle the diminishing excess part $[T, T_{n+1} - T_n)$, which can get messy.*

We can regard $\tilde{z}_n(t)$ as the Euler's discretization of $z_n(t)$ defined below.

**Definition 7.** $\forall n \in \mathbb{N}, t \in [0, T+1)$, *define* $z_n(t) = (x_n(t), y_n(t))$ *as the solution to the ODE system*

$$\frac{dx_n(t)}{dt} = h_{r_n}(x_n(t), y_n(t))$$

$$\frac{dy_n(t)}{dt} = 0$$

*with initial condition*

$$z_n(0) = \tilde{z}_n(0). \tag{20}$$

---

[3]In this paper, we use the convention that $\sum_{k=i}^{j} \alpha(k) = \sum_{k=i}^{j} \beta(k) = 0$ when $j < i$.

We can also write $z_n(t)$ as

$$z_n(t) = \tilde{z}_n(0) + \int_0^t (h_{r_n}(z_n(s)), 0)ds. \tag{21}$$

Ideally, we would like to see that the error of the discretization diminishes asymptotically. Precisely speaking, the discretization error is defined as

$$f_n(t) \doteq \tilde{z}_n(t) - z_n(t)$$

and we want to show that $f_n(t)$ diminishes to 0 uniformly in $t$ as $n \to \infty$. To accomplish this, we need to analyze the following three sequences of functions

$$\{t \mapsto \tilde{z}_n(t)\}_{n=0}^\infty, \{z_n(t)\}_{n=0}^\infty, \{f_n(t)\}_{n=0}^\infty.$$

In particular, we show that they are all *equicontinuous in the extended sense*. We defer the relevant definitions, statements, and proofs to Appendix 16.3 as they are quite similar to the analogous sections in Liu et al. (2025).

## 12.3   A Convergent Subsequence

The ultimate goal would be to show that

$$\sup_n \|z_n\| < \infty.$$

Observe the inequality

$$\forall n, \quad \|z_{m(T_n)}\| = \|\bar{z}(T_n)\| \leq r_n.$$

Therefore, if we had

$$\sup_n r_n < \infty,$$

then the result would come easily. So, we assume for contradiction that $\sup_n r_n = \infty$. However, we can't obtain a contradiction in this section and must first show Theorem 1.

According to the Arzela-Ascoli theorem in the extended sense (Theorem 10.4), a sequence of equicontinuous functions always has a subsequence of functions that uniformly converge to a continuous limit. In the following, we use this to identify a particular subsequence of interest.

**Lemma 12.1 (5.1 Restated).** *Suppose $\sup_n r_n = \infty$. Take an arbitrary subsequence $\{n_{k,0}\}_{k=0}^\infty \subseteq \{0, 1, 2, \dots\}$. Then there is a subsequence $\{n_k\}_{k=0}^\infty \subseteq \{n_{k,0}\}_{k=0}^\infty$ such that there exist some continuous functions $f^{\mathrm{lim}}(t)$ and $\tilde{z}^{\mathrm{lim}}(t)$ such that $\forall t \in [0, T+1)$,*

$$\lim_{k \to \infty} f_{n_k}(t) = f^{\mathrm{lim}}(t),$$

$$\lim_{k \to \infty} \tilde{z}_{n_k}(t) = \tilde{z}^{\mathrm{lim}}(t), \tag{22}$$

*where both convergences are uniform in $t$ on $[0, T+1)$. Furthermore, let $z^{\mathrm{lim}}(t) = (x^{\mathrm{lim}}(t), y^{\mathrm{lim}}(t))$ denote the unique solution to the ODE system*

$$\frac{dx^{\mathrm{lim}}(t)}{dt} = h_\infty(x^{\mathrm{lim}}(t), y^{\mathrm{lim}}(t))$$

$$\frac{dy^{\mathrm{lim}}(t)}{dt} = 0$$

*with the initial condition*

$$z^{\mathrm{lim}}(0) = \tilde{z}_n^{\mathrm{lim}}(0),$$

*in other words,*

$$z^{\mathrm{lim}}(t) = \tilde{z}^{\mathrm{lim}}(0) + \int_0^t (h_\infty(z^{\mathrm{lim}}(s)), 0)ds. \tag{23}$$

*Then $\forall t \in [0, T+1)$, we have*

$$\lim_{k \to \infty} z_{n_k}(t) = z^{\mathrm{lim}}(t),$$

*where the convergence is uniform in $t$ on $[0, T+1)$.*

Its proof is in Appendix 16.4. We use the subsequence $\{n_k\}$ intensively in the remaining proofs.

## 12.4 Diminishing Discretization Error

Recall that $f_n(t)$ denotes the discretization error between $\tilde{z}_n(t)$ and $z_n(t)$. In this section, we will show that $\lim_{n \to \infty} \|f_n(t)\| = 0$ for all $t \in [0, T+1)$. We start by first proving that the discretization error diminishes along the sequence $\{n_k\}$, i.e., that

$$\lim_{k \to \infty} \|f_{n_k}(t)\| = \|f^{\lim}(t)\| = 0.$$

This means $\tilde{z}_{n_k}(t)$ is close to $z_{n_k}(t)$ as $k \to \infty$. For any $t \in [0, T+1)$, we have

$$\lim_{k \to \infty} \|f_{n_k}(t)\|$$

$$= \lim_{k \to \infty} \left\| \sum_{i=m(T_{n_k})}^{m(T_{n_k}+t)-1} (\alpha(i) H_{r_{n_k}}(\tilde{z}_{n_k}(t(i) - T_{n_k}), W_{i+1}), \beta(i) G_{r_{n_k}}(\tilde{z}_{n_k}(t(i) - T_{n_k}), W_{i+1})) \right.$$

$$\left. - \int_0^t (h_{r_{n_k}}(z_{n_k}(s)), 0) ds \right\| \qquad \text{(by (21))}$$

$$\leq \lim_{k \to \infty} \left\| \sum_{i=m(T_{n_k})}^{m(T_{n_k}+t)-1} \alpha(i) H_{r_{n_k}}(\tilde{z}_{n_k}(t(i) - T_{n_k}), W_{i+1}) - \int_0^t h_{r_{n_k}}(\tilde{z}^{\lim}(s)) ds \right\|$$

$$+ \lim_{k \to \infty} \left\| \int_0^t h_{r_{n_k}}(\tilde{z}^{\lim}(s)) ds - \int_0^t h_{r_{n_k}}(z_{n_k}(s)) ds \right\| \qquad (24)$$

$$+ \lim_{k \to \infty} \left\| \sum_{i=m(T_{n_k})}^{m(T_{n_k}+t)-1} \beta(i) G_{r_{n_k}}(\tilde{z}_{n_k}(t(i) - T_{n_k}), W_{i+1}) \right\|.$$

Here, we will bound the last term. This term is a novelty of the two-timescale setting and so there is no analogous result in Liu et al. (2025).

**Lemma 12.2.** $\forall t \in [0, T+1)$,

$$\lim_{k \to \infty} \left\| \sum_{i=m(T_{n_k})}^{m(T_{n_k}+t)-1} \beta(i) G_{r_{n_k}}(\tilde{z}_{n_k}(t(i) - T_{n_k}), W_{i+1}) \right\| = 0.$$

Its proof is in Appendix 16.5. We defer the rest of the argument to Appendix 16.6 since it is quite similar to the relevant section in Liu et al. (2025). At the end of it all, from Lemma 16.9, we obtain

$$\lim_{k \to \infty} \|f_{n_k}(t)\| = 0,$$

for all $t \in [0, T+1)$ showing that the discretization error goes to 0 along $\{n_k\}$.

Now, since we had chosen an arbitrary subsequence $\{f_{n_{k,0}}\}$ and it has a subsequence $\{f_{n_k}\}$ that converges to 0, by Lemma 10.1 we know that $\{f_n\}$ also converges to 0. Thus, the discretization error diminishes along the entire sequence. That is,

$$\lim_{k \to \infty} \|f_n(t)\| = 0 \qquad (25)$$

for all $t \in [0, T+1)$.

## 12.5 ODEs with External Inputs

This section contains some notation and results concerning ODEs with external inputs, which we need for the next section. First, we will define some new notation.

**Definition 8.** *We use the notation $\eta_c^{y(t)}(t, x)$ to denote the solution to the ode*

$$\frac{dx(t)}{dt} = h_c(x(t), y(t))$$

*with the initial condition $x$.*

Note that under this notation, $x_n(t)$ (7) can be written $\eta_{r_n}^{\tilde{y}_n(0)}(t, \tilde{x}_n(0))$. This notation (borrowed from Lakshminarayanan & Bhatnagar (2017)) is useful since it identifies a trajectory of an ODE parameterized by the rescaling factor $c$ and the external input $y(t)$.

The following lemma shows that within a certain amount of time, the trajectory of the ODE will be pulled close to the equilibrium determined by the external input.

**Lemma 12.3 (Lemma 5.2 Restated).** *Let $K \subset \mathbb{R}^{d_1}$ be compact and fix $y \in \mathbb{R}^{d_2}$. Given any $\epsilon > 0$, there exists a $T_\epsilon > 0$ such that for all initial conditions $x \in K$, we have $\eta_\infty^y(t, x) \in B(\lambda_\infty(y), \epsilon)$[4] for all $t \geq T_\epsilon$.*

Its proof is in Appendix 16.7. The next lemma shows that if the external input is close to some constant $y$, then the trajectory will remain close to the trajectory we would obtain if $y$ was the external input.

**Lemma 12.4 (Lemma 5.3 Restated).** *Let $y \in \mathbb{R}^{d_2}, [0, T]$ be a given time interval, and $\rho$ be a small positive constant with $y'(t) \in B(y, \rho)$. Then,*

$$\left\| \eta_c^{y'(t)}(t, x) - \eta_\infty^y(t, x) \right\| \leq (L\rho + \epsilon(c)) T e^{LT}.$$

*for any initial $x \in \mathbb{R}^{d_1}$ and for all $t \in [0, T]$.*

Its proof is in 16.8. Armed with these results, we are ready for the next section.

### 12.6 COMPLETING THE PROOF

We now proceed to complete the proof of Theorem 1. The most involved result in this section is Lemma 12.5, which tells us that if the $x$ component is too much larger than the $y$ component, the norm of the iterate at $T_{n+1}$ cannot be larger than that of the iterate at $T_n$. We use the results about ODEs with external inputs to ensure that the $x$ component of the trajectory gets pulled to the equilibrium, which is close to 0.

**Lemma 12.5 (Lemma 5.4 Restated).** *There exists a constant $C_1 > 1$ such that if $\|\bar{x}(T_n)\| > C_1(1 + \|\bar{y}(T_n)\|)$, then $r_{n+1} = r_n$.*

*Proof.* Suppose that for some $n$, $\|\bar{x}(T_n)\| > C_1(1 + \|\bar{y}(T_n)\|)$, which implies that $\|\tilde{x}_n(0)\| > C_1(\frac{1}{r_n} + \|\tilde{y}_n(0)\|)$. Since $1 \geq \|\tilde{x}_n(0)\|$ and $\frac{\|\tilde{x}_n(0)\|}{C_1} > \|\tilde{y}_n(0)\|$, we have

$$\|\tilde{y}_n(0)\| < \frac{1}{C_1}. \tag{26}$$

We know from Lemma 5.2 that there exists some $T_{\frac{1}{4}}$ such that for all $t \geq T_{\frac{1}{4}}$,

$$\left\| \eta_\infty^0(t, \tilde{x}_n(0)) \right\| \leq \frac{1}{4}. \tag{27}$$

So if we set $T \doteq T_{\frac{1}{4}}$, then (27) holds for all $t \in [T, T+1)$.

From Lemma 5.3 we know that there exist $\rho_{\frac{1}{4}}$ and $c_{\frac{1}{4}}$ such that if $y(t) \in B(0, \rho_{\frac{1}{4}})$ for all $t \in [0, T+1)$ and $r_n > c_{\frac{1}{4}}$ then

$$\left\| \eta_{r_n}^{y(t)}(t, \tilde{x}_n(0)) - \eta_\infty^0(t, \tilde{x}_n(0)) \right\| \leq \frac{1}{4} \tag{28}$$

for all $t \in [0, T+1)$. Since $\lim_{n \to \infty} r_n = \infty$, there is some finite $n_1$ such that if $n > n_1$, we can be sure that $r_n > c_{\frac{1}{4}}$. By (25), we know that there exists $n_2$ such that for $n > n_2$, the discretization error will diminish enough so that

$$\tilde{y}_n(t) \in B\left( \tilde{y}_n(0), \min\left( \frac{\rho_{\frac{1}{4}}}{2}, \frac{1}{8} \right) \right) \tag{29}$$

---

[4]In case the reader is unfamiliar, the notation $B(x, r)$ denotes the open ball centered at $x$ with radius $r$.

for all $t \in [0, T+1)$. So, if we choose $C_1$ large enough so that $C_1 > 16$ and $\frac{1}{C_1} < \frac{\rho_{\frac{1}{4}}}{2}$, then we have $\tilde{y}_n(t) \in B(0, \rho_{\frac{1}{4}})$ (by (26) and (29)) for all $t \in [0, T+1)$. So, we know that for all $n > \max(n_1, n_2)$, (28) holds.

From (25), we know that there is some finite $n_3$ such that for $n > n_3$,

$$\left\| \tilde{x}_n(t) - \eta_{r_n}^{\tilde{y}_n(0)}(t, \tilde{x}_n(0)) \right\| \leq \frac{1}{4} \tag{30}$$

for all $t \in [0, T+1)$.

By (27), (28), and (30), since $\tilde{x}_n(T_{n+1} - T_n) = \frac{\bar{x}(T_{n+1})}{r_n}$, we have $\left\| \frac{\bar{x}(T_{n+1})}{r_n} \right\| \leq \frac{3}{4}$.

For $n > n_1$, since $\tilde{y}_n(T_{n+1} - T_n) = \frac{\bar{y}(T_{n+1})}{r_n}$ we also have $\left\| \frac{\bar{y}(T_{n+1})}{r_n} \right\| \leq \frac{1}{4}$ (by (26), $\frac{1}{C_1} < \frac{1}{16}$, and (29)).

So we conclude $\left\| \frac{\bar{z}(T_{n+1})}{r_n} \right\| \leq 1$, telling us that $r_{n+1} = r_n$ as desired. If we let $n_0 = \max(n_1, n_2, n_3)$ and ensure that $C_1 > \max_{i \leq n_0} \|\bar{x}(T_i)\|$, then the result holds for all $n$. $\qquad\square$

Lemma 12.6 tells us that the iterate at each $T_n$ is bounded by the largest $y$ component seen so far in the sequence of times $T_n$. To do this, we use an inductive argument–intuitively, we combine the previous lemma, which prevents $z$ from growing if the $x$ component is too much larger than the $y$ component, with Lemma 17.15 which prevents $z$ from growing too much within one period $T$, ensuring that $x$ can never get too much larger than $y$.

**Lemma 12.6** (**Lemma 5.5** Restated). *There exists a constant $C_2$ such that for all $n$,*

$$\|\bar{z}(T_n)\| \leq C_1 C_2 (\max_{m \leq n} \|\bar{y}(T_m)\| + 1).$$

*Proof.* From Lemma 17.15, we know that there are some constants $A, B$ such that for all $n$,

$$\|\bar{z}(T_{n+1})\| \leq A\|\bar{z}(T_n)\| + B.$$

Our argument is inductive in nature. Let $C_1, C_2$ and ensure that $C_1 > A + B$ and $C_2 > 2A + B + 2$. To make sure the base case ($n = 0$), holds, we also ensure that $C_1 C_2 > \|z_0\|$. From Lemma 12.5 we know that if $\|\bar{x}(T_n)\| \geq C_1(1 + \|\bar{y}(T_n)\|)$, then $r_{n+1} = r_n$. Thus, if the result holds for all $i$ less than some $n$, i.e., we have $\|\bar{z}(T_i)\| \leq C_1 C_2(\max_{m \leq i} \|\bar{y}(T_m)\| + 1)$ for all $i \leq n$, then we can conclude that $\|\bar{z}(T_{n+1})\| \leq C_1 C_2(\max_{m \leq n} \|\bar{y}(T_m)\| + 1)$ also.

To address the other case, assume for some $n$ that $\|\bar{x}(T_n)\| \leq C_1(\|\bar{y}(T_n)\| + 1)$. Then we have

$$\begin{aligned}
\|\bar{z}(T_{n+1})\| &\leq A\|\bar{z}(T_n)\| + B \\
&\leq A\|\bar{x}(T_n)\| + A\|\bar{y}(T_n)\| + B \\
&\leq AC_1\|\bar{y}(T_n)\| + AC_1 + A\|\bar{y}(T_n)\| + B \\
&\leq C_1 C_2(\|\bar{y}(T_n)\| + 1) \\
&\leq C_1 C_2(\max_{m \leq n+1} \|\bar{y}(T_m)\| + 1).
\end{aligned}$$

$\qquad\square$

Finally, while we've ensured that for all the $T_n$ that $\bar{z}(T_n)$ is not to much larger than $\bar{y}(T_n)$, we need the result to also hold for all the iterates that lie in between $T_n$ and $T_{n+1}$. Lemma 12.7 takes care of this.

**Lemma 12.7** (**Lemma 5.6** Restated). *There exists a constant $C_3$ such that for all $n$,*

$$\|x_n\| \leq C_1 C_2 C_3(\|y_n^{max}\| + 1),$$

*Proof.* From Lemma 17.15 we know that for all $m$ such that $T_n \leq m \leq T_{n+1}$, there exist constants $A, B$ such that

$$\|z_m\| \leq A\|\bar{z}(T_n)\| + B.$$

Let $C_3 > A + B$. This means that

$$\|x_m\| \leq \|z_m\| \leq AC_1C_2(\max_{l \leq n} \|\bar{y}(T_l)\| + 1) + B$$

$$\leq C_1C_2C_3(\max_{l \leq n} \|\bar{y}(T_l)\| + 1)$$

$$\leq C_1C_2C_3(\|y_n^{\max}\| + 1).$$

$\square$

By setting $K$ equal to $C_1C_2C_3$, this concludes the proof of Theorem 1.

**Lemma 12.8.** *Let $K_y \subset \mathbb{R}^{d_2}$ where $K_y$ is compact. Given any $\epsilon > 0$, there exists a $T_\epsilon > 0$ such that for all constant external inputs $y \in K_y$, we have $\eta_\infty^y(t, x) \in B(\lambda_\infty(y), \epsilon)$ for any initial condition $x \in \lambda_\infty(K_y)$ and for all $t \geq T_\epsilon$.*

*Proof.* By Lyapunov stability, we know that there is some $\delta$ with $\frac{\epsilon}{2} > \delta > 0$ such that if $\eta_\infty^y(t, x) \in B(\lambda_\infty(y), \delta)$, then for all $t' > t$, $\eta_\infty^y(t', x) \in B(\lambda_\infty(y), \frac{\epsilon}{2})$. By Lemma 5.2 we know that there exists some time $T_\delta$ such that for all $x \in \lambda_\infty(K_y)$ (image of a compact set under a continuous map is compact) and $t \geq T_\delta$, we have

$$\eta_\infty^y(t, x) \in B(\lambda_\infty(y), \frac{\delta}{2}). \tag{31}$$

By Lemma 5.3, we can select $\rho_y$ small enough such that for all $y_1 \in B(y, \rho_y)$,

$$\|\eta_\infty^y(t, x) - \eta_\infty^{y_1}(t, x)\| \leq \frac{\delta}{2} \tag{32}$$

for all $t \in [0, T_\delta]$, $x \in \lambda_\infty(K_y)$.

We can split up the timeline into chunks of size $T_\delta$. The insight we will rely on is that if $\eta_\infty^y(T_\delta, x) = x_1$, then $\eta_\infty^y(T_\delta + t, x) = \eta_\infty^y(t, x_1)$ for all $t \geq 0$. The way our logic proceeds is as follows: By (31) and (32) we have that

$$\eta_\infty^{y_1}(T_\delta, x) \in B(\lambda_\infty(y), \delta)$$

Let $x_1 \doteq \eta_\infty^{y_1}(T_\delta, x)$. For all $t \in [0, T_\delta]$, we know by Lyapunov stability that

$$\eta_\infty^y(t, x_1) \in B(\lambda_\infty(y), \frac{\epsilon}{2}),$$

so by (32) we know (since we can reuse the $\rho_y$ selected earlier) that

$$\eta_\infty^{y_1}(t, x_1) \in B(\lambda_\infty(y), \frac{3\epsilon}{4}).$$

for all $t \in [0, T_\delta]$, implying that

$$\eta_\infty^{y_1}(t, x) \in B(\lambda_\infty(y), \frac{3\epsilon}{4}) \tag{33}$$

for all $t \in [T_\delta, 2T_\delta]$.

By (31) and (32) we know that

$$\eta_\infty^{y_1}(T_\delta, x_1) \in B(\lambda_\infty(y), \delta).$$

We can then define $x_2 \doteq \eta_\infty^{y_1}(T_\delta, x_1)$ and repeat the above arguments to see that (33) holds for all $t \in [2T_\delta, 3T_\delta]$. We can continue repeating this argument for all $x_n$ to see that (33) holds for all $t \geq T_\delta$.

Let $L_\lambda$ be the Lipschitz constant for $\lambda_\infty$. Then for all $y_1 \in B(y, \frac{\epsilon}{4L_\lambda})$ we have

$$\|\lambda_\infty(y) - \lambda_\infty(y_1)\| \leq \frac{\epsilon}{4}. \tag{34}$$

To summarize, by (33) and (34), we know that if $y_1 \in B(y, \min(\rho_y, \frac{\epsilon}{4L_\lambda}))$ then

$$\eta_\infty^{y_1}(t, x) \in B(\lambda_\infty(y_1), \epsilon).$$

Each $y$ gives us such a ball $B(y, \min(\rho_y, \frac{\epsilon}{4L_\lambda}))$ along with a $T_\delta$ and these balls cover $K$. By compactness, we can obtain a finite subcover and a finite number of times $T_1, \ldots, T_n$. Taking $T_\epsilon$ to be the maximum of these times completes the proof. $\square$

**Lemma 12.9.** *Let $K_y \subset \mathbb{R}^{d_2}$ where $K_y$ is compact. Given any $\epsilon > 0$, there exists a $\delta$ such that for all constant external inputs $y \in K_y$, if the initial condition $x \in B(\lambda_\infty(y), \delta)$, then $\eta_\infty^y(t, x) \in B(\lambda_\infty(y), \epsilon)$ for all $t \geq 0$.*

*Proof.* Fix $y$ and $\epsilon$. From Lemma 12.8 we know that the result holds for $t \geq T_\epsilon$, so we just need to show that it holds for $0 \leq t < T_\epsilon$. By Lyapunov stability, there exists $\delta_y$ such that if $x \in B(\lambda_\infty(y), \delta_y)$, then for all $t \geq 0$, $\eta_\infty^y(t, x) \in B(\lambda_\infty(y), \frac{\epsilon}{3})$.

By Lemma 5.3, there exists $\rho$ small enough that for all $y' \in B(y, \rho)$,

$$\|\eta_\infty^y(t, x) - \eta_\infty^{y_1}(t, x)\| < \frac{\epsilon}{3}$$

for all $x \in B(\lambda_\infty(y), \delta_y)$, $t \in [0, T_\delta]$. Finally, since $\lambda_\infty$ is Lipschitz, if $y' \in B(y, \frac{\epsilon}{3L_\lambda})$,

$$\|\lambda_\infty(y) - \lambda_\infty(y_1)\| < \frac{\epsilon}{3}.$$

Combining these facts, we know that if $y_1 \in B(y, \min(\rho, \frac{\epsilon}{3L_\lambda}))$, then $\eta_\infty^y(t, x) \in B(\lambda_\infty(y), \epsilon)$ for all $t \geq 0$. Each $y$ comes with such a neighborhood and a distance $\delta$. By compactness, we can extract a finite subcover and a finite number of distances $\delta_1, \ldots, \delta_n$, and taking $\delta$ to be the smallest one gives us our result. $\qquad\square$

**Lemma 12.10** (**Lemma 5.7 Restated**). *For any $\epsilon > 0$, there is some $T_\epsilon$ and some $N_\epsilon$ such that for all $n > N_\epsilon$, $\|\tilde{x}_n(t) - \lambda_\infty(\tilde{y}_n(t))\| \leq \epsilon$ for all $t \in [0, T_\epsilon]$.*

*Proof.* By 12.9, we know that for all $y \in [-2, 2]^{d_2}$ there is some $\delta$ with $\frac{\epsilon}{2} > \delta > 0$ such that if $\eta_\infty^y(t, x) \in B(\lambda_\infty(y), \delta)$, then for all $t' > t$, $\eta_\infty^y(t', x) \in B(\lambda_\infty(y), \frac{\epsilon}{2})$. By Lemma 12.8 we know that there exists some time $T_\delta$ such that for all $x \in \lambda_\infty(K_y)$ (image of a compact set under a continuous map is compact) and $t \geq T_\delta$, we have, for all $y \in [-2, 2]^{d_2}$,

$$\eta_\infty^y(t, x) \in B(\lambda_\infty(y), \frac{\delta}{4}). \tag{35}$$

By Lemma 5.3, there exists $n_1$ such that for all $n > n_1$,

$$\left\|\eta_{r_n}^y(t, x) - \eta_\infty^y(t, x)\right\| < \frac{\delta}{4} \tag{36}$$

for all $t \in [0, T_\delta]$.

By (25), there exists $n_2$ such that for all $n > n_2$,

$$\left\|\tilde{x}_n(t) - \eta_{r_n}^{\tilde{y}_n(0)}(t, \tilde{x}_n(0))\right\| < \frac{\delta}{4} \tag{37}$$

and

$$\|\lambda_\infty(\tilde{y}_n(t)) - \lambda_\infty(\tilde{y}_n(0))\| \leq L_\lambda \|\tilde{y}_n(t) - \tilde{y}_n(0)\| < \frac{\delta}{4} \tag{38}$$

for all $t \in [0, T_\delta]$. Combining (35), (36), (37), and (38), we have

$$\|\tilde{x}_n(T_\delta) - \lambda_\infty(\tilde{y}_n(T_\delta))\| < \delta$$

for all $n > \max(n_1, n_2)$. This means that

$$\|\tilde{x}_{n+1}(0) - \lambda_\infty(\tilde{y}_{n+1}(0))\| = \frac{r_n}{r_{n+1}} \|\tilde{x}_n(T_\delta) - \lambda_\infty(\tilde{y}_n(T_\delta))\| < \delta$$

when $n > \max(n_1, n_2)$, giving us

$$\|\tilde{x}_n(0) - \lambda_\infty(\tilde{y}_n(0))\| < \delta$$

for all $n > N_\epsilon = \max(n_1, n_2) + 1$. So,

$$\left\|\eta_{r_n}^{\tilde{y}_n(0)}(t, \tilde{x}_n(0)) - \lambda_\infty(\tilde{y}_n(0))\right\| < \frac{\epsilon}{2}$$

for all $t \geq 0$. Combining this with (37) and (38) gives us

$$\|\tilde{x}_n(t) - \lambda_\infty(\tilde{y}_n(t))\| \leq \epsilon$$

for all $t \in [0, T_\delta]$, as desired. Taking $T_\epsilon \doteq T_\delta$ completes the proof.

$\square$

## 13 PROOF OF THEOREM 2

In this section we will prove Theorem 2.

### 13.1 SLOW TIMESCALE SETUP

Here we are working in the slow timescale. We reuse the notation from Section 12.1 but redefine some things in terms of the slow timescale. We split the positive real axis $[0, \infty)$ into chunks of length $\{\beta(i)\}_{i=0,1,\dots}$. We then collect these segments together into larger intervals $\{[T_n, T_{n+1})\}_{n=0,1,\dots}$, where the sequence $\{T_n\}$ has the property that as $n$ grows large, we have $T_{n+1} - T_n \approx T$. We define

$$t(0) \doteq 0,$$

$$t(n) \doteq \sum_{i=0}^{n-1} \beta(i) \quad n = 1, 2, \dots \quad .$$

For all $T > 0$, define

$$m(T) = \max\{i | T \geq t(i)\}$$

to be the maximal $i$ where $t(i)$ is no greater than $T$.

Define

$$T_0 = 0,$$
$$T_{n+1} = t(m(T_n + T) + 1).$$

For any given function $f$ with domain $\mathcal{W}$, its asymptotic rate of change is defined as

$$\limsup_n \sup_{-\tau \leq t_1 \leq t_2 \leq \tau} \left\| \sum_{i=m(t(n)+t_1)}^{m(t(n)+t_2)-1} \beta(i)[f(W_{i+1}) - \mathbb{E}_{w \sim d_{\mathcal{W}}}[f(w)]] \right\|.$$

### 13.2 DEFINING THE SLOW ITERATES

Here, we take $\bar{z}(t)$ to be the piecewise constant interpolation of $z_n$ at points $\{t(n)\}_{n=0,1,\dots}$, i.e.,

$$\bar{z}(t) \doteq (\bar{x}(t), \bar{y}(t)) \doteq \begin{cases} (x_0, y_0) & t \in [0, t(1)) \\ (x_1, y_1) & t \in [t(1), t(2)) \\ (x_2, y_2) & t \in [t(2), t(3)) \\ \vdots \end{cases}$$

Now we describe our rescaling of $\bar{z}(t)$.

**Definition 9.** $\forall n \in \mathbb{N}, t \in [0, T+1)$, *define*

$$\tilde{z}_n(t) \doteq (\tilde{x}_n(t), \tilde{y}_n(t)) \doteq \frac{\bar{z}(T_n + t)}{r_n}.$$

*We will define $r_n$ differently, such that it will be at least as large as the largest iterate seen so far:*

$$r_n \doteq \max\left\{1, \left\| z_{m(T_n)}^{max} \right\| \right\}.$$

*where $z_n^{max} \doteq z_m$ such that $m = \arg\max_{i \leq n} \|z_i\|$.*

We can regard $\tilde{y}_n(t)$ as the Euler's discretization of $y_n(t)$ defined below.

**Definition 10.** $\forall n \in \mathbb{N}, t \in [0, T+1)$, *define $y_n(t)$ as the solution to the ODE*

$$\frac{dy_n(t)}{dt} = g_{r_n}(\lambda_\infty(y_n(t)), y_n(t))$$

*with initial condition*

$$y_n(0) = \tilde{y}_n(0).$$

We can also write $y_n(t)$ as

$$y_n(t) = \tilde{y}_n(0) + \int_0^t g_{r_n}(\lambda_\infty(y_n(t)), y_n(t))ds. \tag{39}$$

The discretization error is defined as

$$f_n(t) \doteq \tilde{y}_n(t) - y_n(t)$$

and we want to show that $f_n(t)$ diminishes to 0 uniformly in $t$ as $n \to \infty$. To accomplish this, we need to analyze the following three sequences of functions

$$\{t \mapsto \tilde{y}_n(t)\}_{n=0}^\infty, \{y_n(t)\}_{n=0}^\infty, \{f_n(t)\}_{n=0}^\infty.$$

In particular, we show that they are all *equicontinuous in the extended sense*. We defer the relevant definitions, statements, and proofs to Appendix 16.3 as they are quite similar to the analogous sections in Liu et al. (2025).

**Lemma 13.1.** $\{\tilde{y}_n(t)\}_{n=0}^\infty$ *is equicontinuous in the extended sense on* $[0, T + 1)$.

*Proof.* We know that

$$\sup_n \|\tilde{y}_n(0)\| \leq 1.$$

Without loss of generality, let $t_1 \leq t_2$.

$$\limsup_n \sup_{0 \leq t_2 - t_1 \leq \delta} \|\tilde{y}_n(t_1) - \tilde{y}_n(t_2)\| = \limsup_n \sup_{0 \leq t_2 - t_1 \leq \delta} \left\| \sum_{i=m(T_n+t_1)}^{m(T_n+t_2)-1} \beta(i)G_{r_n}(\tilde{z}_n(t(i) - T_n), W_{i+1}) \right\|$$

$\forall \xi > 0$, by (95), $\exists \delta_0$, such that $\forall 0 < \delta \leq \delta_0$,

$$\sup_{c \geq 1} \limsup_n \sup_{0 \leq t_2 - t_1 \leq \delta} \left\| \sum_{i=m(T_n+t_1)}^{m(T_n+t_2)-1} \beta(i)G_c(0, W_{i+1}) \right\| \leq \xi. \tag{40}$$

By (101), $\exists \delta_1$, such that $\forall 0 < \delta \leq \delta_1$,

$$\limsup_n \sup_{0 \leq t_2 - t_1 \leq \delta} \sum_{i=m(T_n+t_1)}^{m(T_n+t_2)-1} \beta(i)L(W_{i+1}) \leq \xi. \tag{41}$$

Without loss of generality, let $t_1 \leq t_2$. Then $\forall \delta \leq \min\{\delta_0, \delta_1\}$, we have

$$\limsup_n \sup_{0 \leq t_2 - t_1 \leq \delta} \left\| \sum_{i=m(T_n+t_1)}^{m(T_n+t_2)-1} \beta(i) G_{r_n}(\tilde{z}_n(t(i) - T_n), W_{i+1}) \right\|$$

$$\leq \limsup_n \sup_{0 \leq t_2 - t_1 \leq \delta} \left\| \left\| \sum_{i=m(T_n+t_1)}^{m(T_n+t_2)-1} \beta(i) G_{r_n}(\tilde{z}_n(t(i) - T_n), W_{i+1}) \right\| - \left\| \sum_{i=m(T_n+t_1)}^{m(T_n+t_2)-1} \beta(i) G_{r_n}(0, W_{i+1}) \right\| \right\|$$

$$+ \limsup_n \sup_{0 \leq t_2 - t_1 \leq \delta} \left\| \sum_{i=m(T_n+t_1)}^{m(T_n+t_2)-1} \beta(i) G_{r_n}(0, W_{i+1}) \right\|$$

$$\leq \limsup_n \sup_{0 \leq t_2 - t_1 \leq \delta} \left\| \left\| \sum_{i=m(T_n+t_1)}^{m(T_n+t_2)-1} \beta(i) G_{r_n}(\tilde{z}_n(t(i) - T_n), W_{i+1}) \right\| - \left\| \sum_{i=m(T_n+t_1)}^{m(T_n+t_2)-1} \beta(i) G_{r_n}(0, W_{i+1}) \right\| \right\|$$

$$+ \sup_{c \geq 1} \limsup_n \sup_{0 \leq t_2 - t_1 \leq \delta} \left\| \sum_{i=m(T_n+t_1)}^{m(T_n+t_2)-1} \beta(i) G_c(0, W_{i+1}) \right\|$$

$$\leq \limsup_n \sup_{0 \leq t_2 - t_1 \leq \delta} \left\| \left\| \sum_{i=m(T_n+t_1)}^{m(T_n+t_2)-1} \beta(i) G_{r_n}(\tilde{z}_n(t(i) - T_n), W_{i+1}) \right\| - \left\| \sum_{i=m(T_n+t_1)}^{m(T_n+t_2)-1} \beta(i) G_{r_n}(0, W_{i+1}) \right\| \right\|$$

$$+ \xi \qquad\qquad\qquad\qquad\qquad\qquad\qquad\qquad\qquad\qquad\qquad \text{(by (40))}$$

$$\leq \limsup_n \sup_{0 \leq t_2 - t_1 \leq \delta} \left\| \sum_{i=m(T_n+t_1)}^{m(T_n+t_2)-1} \beta(i) G_{r_n}(\tilde{z}_n(t(i) - T_n), W_{i+1}) - \sum_{i=m(T_n+t_1)}^{m(T_n+t_2)-1} \beta(i) G_{r_n}(0, W_{i+1}) \right\|$$

$$+ \xi$$

$$\leq \limsup_n \sup_{0 \leq t_2 - t_1 \leq \delta} \sum_{i=m(T_n+t_1)}^{m(T_n+t_2)-1} \beta(i) \| G_{r_n}(\tilde{z}_n(t(i) - T_n), W_{i+1}) - G_{r_n}(0, W_{i+1}) \| + \xi$$

$$\leq \limsup_n \sup_{0 \leq t_2 - t_1 \leq \delta} \sum_{i=m(T_n+t_1)}^{m(T_n+t_2)-1} \beta(i) L(W_{i+1}) \| \tilde{z}_n(t(i) - T_n) \| + \xi$$

$$\leq E \limsup_n \sup_{0 \leq t_2 - t_1 \leq \delta} \sum_{i=m(T_n+t_1)}^{m(T_n+t_2)-1} \beta(i) L(W_{i+1}) + \xi \qquad\qquad\qquad \text{(by Lemma 13.11)}$$

$$\leq E\xi + \xi. \qquad\qquad\qquad\qquad\qquad\qquad\qquad\qquad\qquad\qquad\qquad\qquad \text{(by (41))}$$

$$\square$$

Equicontinuity for the other two sequences of functions follows more similarly to the fast timescale arguments.

### 13.3 A CONVERGENT SUBSEQUENCE

The ultimate goal is to show that

$$\sup_n \|z_n\| < \infty.$$

Once again, observe the inequality

$$\forall n, \quad \|z_{m(T_n)}\| = \|\bar{z}(T_n)\| \leq r_n.$$

Therefore, if we had

$$\sup_n r_n < \infty,$$

then the result would come easily. So, we assume for contradiction from now on that $\sup_n r_n = \infty$.

According to the Arzela-Ascoli theorem in the extended sense (Theorem 10.4), a sequence of equicontinuous functions always has a subsequence of functions that uniformly converge to a continuous limit. In the following, we use this to identify a particular subsequence of interest.

**Lemma 13.2.** *Suppose* $\sup_n r_n = \infty$. *Take an arbitrary subsequence* $\{n_{k,0}\}_{k=0}^\infty \subseteq \{0, 1, 2, \dots\}$. *Then there is a subsequence* $\{n_k\}_{k=0}^\infty \subseteq \{n_{k,0}\}_{k=0}^\infty$ *such that there exist some continuous functions* $f^{\lim}(t)$ *and* $\tilde{y}^{\lim}(t)$ *such that* $\forall t \in [0, T+1)$,

$$\lim_{k \to \infty} f_{n_k}(t) = f^{\lim}(t),$$

$$\lim_{k \to \infty} \tilde{y}_{n_k}(t) = \tilde{y}^{\lim}(t), \tag{42}$$

*where both convergences are uniform in* $t$ *on* $[0, T+1)$. *Furthermore, let* $y^{\lim}(t)$ *denote the unique solution to the ODE*

$$\frac{dy^{\lim}(t)}{dt} = g_\infty(\lambda_\infty(y^{\lim}(t)), y^{\lim}(t))$$

*with the initial condition*

$$y^{\lim}(0) = \tilde{y}_n^{\lim}(0),$$

*in other words,*

$$y^{\lim}(t) = \tilde{y}^{\lim}(0) + \int_0^t g_\infty(\lambda_\infty(y^{\lim}(s)), y^{\lim}(s))ds.$$

*Then* $\forall t \in [0, T+1)$, *we have*

$$\lim_{k \to \infty} y_{n_k}(t) = y^{\lim}(t),$$

*where the convergence is uniform in* $t$ *on* $[0, T+1)$.

*Proof.* Since $\sup_n r_n = \infty$ and $r_n$ is monotonic, $\lim_{n \to \infty} r_n = \infty$, and every subsequence also converges to infinity.

Since $\{f_{n_{k,0}}\}$ is equicontinuous, by the Arzelà-Ascoli theorem (see Appendix 10.4), there exists a subsequence $n_{k,1} \subseteq n_{k,0}$ such that $\{f_{n_{k,1}}\}$ converges uniformly to a continuous limit $f^{\lim}$. Similarly, since $\{\tilde{y}_{n_{k,1}}(t)\}$ is equicontinuous, there is a subsequence $n_k \subseteq n_{k,1}$ such that $\{\tilde{z}_{n_k}(t)\}$ converges uniformly in $t$ to a continuous limit $\tilde{y}^{\lim}(t)$.

The proof that $\lim_{n \to \infty} y_{n_k}(t) = y^{\lim}(t)$ uniformly is by lemma 13.5. $\qquad \square$

**Lemma 13.3.** $\sup_{t \in [0, T+1)} \left\| y^{\lim}(t) \right\| \leq C$.

*Proof.* $\forall t \in [0, T+1)$,

$$\|y^{\lim}(t)\|$$

$$= \left\| y^{\lim}(0) + \int_0^t g_\infty(\lambda_\infty(y^{\lim}(s)), y^{\lim}(s)) ds \right\|$$

$$\leq \|y^{\lim}(0)\| + \left\| \int_0^t g_\infty(\lambda_\infty(y^{\lim}(s)), y^{\lim}(s)) ds \right\|$$

$$= \|y^{\lim}(0)\| + \left\| \int_0^t \left[ g_\infty(\lambda_\infty(y^{\lim}(s)), y^{\lim}(s)) - g_\infty(0) \right] ds + \int_0^t g_\infty(0) ds \right\|$$

$$\leq \|y^{\lim}(0)\| + \int_0^t \left\| g_\infty(\lambda_\infty(y^{\lim}(s)), y^{\lim}(s)) - g_\infty(0) \right\| ds + \int_0^t \|g_\infty(0)\| ds$$

$$\leq \|y^{\lim}(0)\| + \int_0^t L\|\lambda_\infty(y^{\lim}(s))\| + L\|y^{\lim}(s)\| ds + \int_0^t \|g_\infty(0)\| ds \qquad \text{(by Lemma 17.2)}$$

$$\leq 1 + \int_0^t L \cdot L_\lambda \|y^{\lim}(s)\| + L\|y^{\lim}(s)\| ds + \int_0^t \|g_\infty(0)\| ds$$

$$\leq 1 + \int_0^t L(L_\lambda + 1)\|y^{\lim}(s)\| ds + (T+1)\|g_\infty(0)\|$$

$$\leq 1 + \int_0^t L(L_\lambda + 1)\|y^{\lim}(s)\| ds + C_H \qquad \text{(by Assumption 5)}$$

$$\leq [1 + C_H] e^{\int_0^t L(L_\lambda + 1) ds} \qquad \text{(by Gronwall inequality in Theorem 10.1)}$$

$$\leq [1 + C_H] e^{L(L_\lambda + 1)(T+1)}$$

$$\leq C. \qquad \text{(by (103))}$$

$\square$

**Lemma 13.4.** $\lim_{k \to \infty} g_{r_{n_k}}(\lambda_\infty(y^{\lim}(t)), y^{\lim}(t)) = g_\infty(\lambda_\infty(y^{\lim}(t)), y^{\lim}(t))$ *uniformly in* $t \in [0, T+1)$.

*Proof.* By Assumption 5, $\lim_{k \to \infty} g_{r_{n_k}}(v) = g_\infty(v)$ uniformly in a compact set $\{v | v \in \mathbb{R}^{d_1 + d_2}, \|v\| \leq C\}$. By Lemma 13.3, $\{y^{\lim}(t) | t \in [0, T+1)\} \subseteq \{v | v \in \mathbb{R}^d, \|v\| \leq C\}$. Additionally, $\lambda_\infty$ is Lipschitz, so $\{(\lambda_\infty(y^{\lim}(t)), y^{\lim}(t)) | t \in [0, T+1)\} \subseteq \{v | v \in \mathbb{R}^{d_1 + d_2}, \|v\| \leq C\}$.

Therefore, $\lim_{k \to \infty} g_{r_{n_k}}(\lambda_\infty(y^{\lim}(t)), y^{\lim}(t)) = g_\infty(\lambda_\infty(y^{\lim}(t)), y^{\lim}(t))$ uniformly in $\{y^{\lim}(t) | t \in [0, T+1)\}$ and in $t \in [0, T+1)$. $\square$

**Lemma 13.5.** $\forall t \in [0, T+1)$, *we have*

$$\lim_{k \to \infty} y_{n_k}(t) = y^{\lim}(t).$$

*Moreover, the convergence is uniform in $t$ on $[0, T+1)$.*

*Proof.* By (42), $\forall \delta > 0$, there exists a $k_1$ such that $\forall k \geq k_1, \forall t \in [0, T+1)$,

$$\|\tilde{y}_{n_k}(t) - \tilde{y}^{\lim}(t)\| \leq \delta. \qquad (43)$$

By Lemma 13.4, there exists a $k_2$ such that $\forall k \geq k_2, \forall t \in [0, T+1)$,

$$\left\| g_{n_k}(\lambda_\infty(y^{\lim}(t)), y^{\lim}(t)) - g_\infty(\lambda_\infty(y^{\lim}(t)), y^{\lim}(t)) \right\| \leq \delta. \qquad (44)$$

$\forall k \geq \max\{k_1, k_2\}, \forall t \in [0, T+1)$

$$\left\| y_{n_k}(t) - y^{\lim}(t) \right\|$$

$$= \left\| \tilde{y}_{n_k}(0) + \int_0^t g_{r_{n_k}}(\lambda_\infty(y_{n_k}(s)), y_{n_k}(s))ds - \tilde{y}^{\lim}(0) - \int_0^t g_\infty(\lambda_\infty(y^{\lim}(s)), y^{\lim}(s))ds \right\|$$

$$\leq \left\| \tilde{y}_{n_k}(0) - \tilde{y}^{\lim}(0) \right\| + \left\| \int_0^t g_{r_{n_k}}(\lambda_\infty(y_{n_k}(s)), y_{n_k}(s))ds - \int_0^t g_\infty(\lambda_\infty(y^{\lim}(s)), y^{\lim}(s))ds \right\|$$

$$\leq \delta + \left\| \int_0^t g_{r_{n_k}}(\lambda_\infty(y_{n_k}(s)), y_{n_k}(s)) - g_\infty(\lambda_\infty(y^{\lim}(s)), y^{\lim})ds \right\| \qquad \text{(by (43))}$$

$$\leq \delta + \int_0^t \left\| g_{r_{n_k}}(\lambda_\infty(y_{n_k}(s)), y_{n_k}(s)) - g_{r_{n_k}}(\lambda_\infty(y^{\lim}(s)), y^{\lim}) \right\| ds$$

$$\quad + \int_0^t \left\| g_{r_{n_k}}(\lambda_\infty(y^{\lim}(s)), y^{\lim}) - g_\infty(\lambda_\infty(y^{\lim}(s)), y^{\lim}) \right\| ds$$

$$\leq \delta + L \int_0^t \left\| (\lambda_\infty(y_{n_k}(s)), y_{n_k}(s)) - (\lambda_\infty(y^{\lim}(s)), y^{\lim})) \right\| ds$$

$$\quad + \int_0^t \left\| g_{r_{n_k}}(\lambda_\infty(y^{\lim}(s)), y^{\lim}) - g_\infty(\lambda_\infty(y^{\lim}(s)), y^{\lim}) \right\| \qquad \text{(by Lemma 17.2)}$$

$$\leq \delta + t\delta + L(L_\lambda + 1) \int_0^t \left\| y_{n_k}(s) - y^{\lim}(s) \right\| ds \qquad \text{(by (44))}$$

$$\leq (\delta + t\delta)e^{L(L_\lambda+1)t} \qquad \text{(by Gronwall inequality in Theorem 10.1)}$$

$$\leq (\delta + T\delta)e^{L(L_\lambda+1)T},$$

which completes the proof. $\qquad\qquad\qquad\qquad\qquad\qquad\qquad\qquad\qquad\qquad\qquad\qquad\quad \square$

We use the subsequence $\{n_k\}$ intensively in the remaining proofs.

### 13.4 DIMINISHING DISCRETIZATION ERROR

Recall that $f_n(t)$ denotes the discretization error between $\tilde{y}_n(t)$ and $y_n(t)$. In this section, we will show that $\lim_{n\to\infty} \|f_n(t)\| = 0$ for all $t \in [0, T+1)$. We start by first proving that the discretization error diminishes along the sequence $\{n_k\}$, i.e., that

$$\lim_{k\to\infty} \|f_{n_k}(t)\| = \|f^{\lim}(t)\| = 0.$$

This means $\tilde{y}_{n_k}(t)$ is close to $y_{n_k}(t)$ as $k \to \infty$. For any $t \in [0, T+1)$, we have

$$\lim_{k\to\infty} \|f_{n_k}(t)\|$$

$$= \lim_{k\to\infty} \left\| \sum_{i=m(T_{n_k})}^{m(T_{n_k}+t)-1} \beta(i)G_{r_{n_k}}(\tilde{z}_{n_k}(t(i) - T_{n_k}), W_{i+1}) - \int_0^t g_{r_{n_k}}(\lambda_\infty(y_{n_k}(s)), y_{n_k}(s))ds \right\|$$

$$\text{(by (39))}$$

$$\leq \lim_{k\to\infty} \left\| \sum_{i=m(T_{n_k})}^{m(T_{n_k}+t)-1} \beta(i)[G_{r_{n_k}}(\tilde{z}_{n_k}(t(i) - T_{n_k}), W_{i+1}) - G_{r_{n_k}}(\lambda_\infty(\tilde{y}_{n_k}(t(i) - T_{n_k})), \tilde{y}_{n_k}(t(i) - T_{n_k}), W_{i+1})] \right\|$$

$$(45)$$

$$\quad + \left\| \sum_{i=m(T_{n_k})}^{m(T_{n_k}+t)-1} \beta(i)G_{r_{n_k}}(\lambda_\infty(\tilde{y}_{n_k}(t(i) - T_{n_k})), \tilde{y}_{n_k}(t(i) - T_{n_k}), W_{i+1}) - \int_0^t g_{r_{n_k}}(\lambda_\infty(y_{n_k}(s)), y_{n_k}(s))ds \right\|.$$

**Lemma 13.6** (**Lemma 6.1** Restated). *For all $\epsilon > 0$, there exists some $N$ such that for all $n > N$,*
$$\|\tilde{x}_n(t) - \lambda_\infty(\tilde{y}_n(t))\| < \epsilon$$
*for all $t \in [0, T]$*

*Proof.* For clarity, since we will work with both fast and slow timescale objects, we will use $\alpha$ and $\beta$ in superscripts to indicate the fast and slow timescale variants respectively. We also define some notation to help in converting between timescales:
$$\tau^\alpha(t) \doteq t^\alpha(m^\beta(t))$$
$$\tau^\beta(t) \doteq t^\beta(m^\alpha(t))$$
By the difference in how the rescaling is defined between the timescales, we know that $\left\|\tilde{z}_n^\beta(0)\right\| \leq \left\|\tilde{z}_m^\alpha(\tau^\alpha(T_n^\beta) - T_m^\alpha)\right\|$, where $m$ is the largest $v$ such that $T_v^\alpha \leq \tau^\alpha(T_n^\beta)$. Essentially, whenever the slow timescale is rescaled, the iterate has a smaller norm than the fast timescale's iterate at that index. Now we need to show that for the rest of that period in the slow timescale (i.e. when $T_n^\beta \leq t \leq T_{n+1}^\beta$) the slow timescale iterates, despite not being rescaled during this period, never get too much larger than the fast timescale iterates which may be rescaled many times.

Thus, given $t \in [0, T^\beta]$, we are interested in the ratio $\frac{r_l^\alpha}{r_n^\beta}$, where $l$ is the largest $v$ where $T_v^\alpha \leq \tau^\alpha(T_n^\beta + t)$. By Lemma 13.11 we know that
$$r_l^\alpha \leq \left\|z_{m^\beta(T_n^\beta+t)}^{max}\right\| \leq C\left\|z_{m^\beta(T_n^\beta)}^{max}\right\| + D = C \cdot r_n^\beta + D$$
implying that
$$\frac{r_l^\alpha}{r_n^\beta} \leq C + \frac{D}{r_n^\beta}.$$
There exists $n_1$ such that for all $n > n_1$, $\frac{D}{r_n^\beta} \leq C$, so for all $n > n_1$,
$$\frac{r_l^\alpha}{r_n^\beta} \leq 2C.$$
By Lemma 12.10, there exist $n_2$ and $T_\epsilon$ such that for all $n > n_\epsilon$, we have
$$\|\tilde{x}_n^\alpha(t) - \lambda_\infty(\tilde{y}_n^\alpha(t))\| \leq \frac{\epsilon}{2C}$$
for all $t \in [0, T_\epsilon]$. Now, set $N = \max\{n_1, n_2\}$, where $n_2$ is the smallest $v$ such that $T_v^\beta \geq \tau^\beta(T_{n_\epsilon}^\alpha)$ and $T^\alpha$ is set to $T_\epsilon$. We then have, for all $t \in [0, T^\beta]$,
$$\left\|\tilde{x}_n^\beta(t) - \lambda_\infty(\tilde{y}_n^\beta(t))\right\| \leq 2C\left\|\tilde{x}_l^\alpha(\tau^\alpha(T_n^\beta + t)) - \lambda_\infty(\tilde{y}_l^\alpha(\tau^\alpha(T_n^\beta + t)))\right\| \leq \epsilon$$
for all $n > N$. Since we set $T^\alpha$ to $T_\epsilon$ without modifying $T^\beta$, the result holds for all $\epsilon$, regardless of what value we set $T^\beta$ to at the start. $\square$

**Lemma 13.7.** *The first error term* (45) *converges to 0.*

*Proof.*
$$\lim_{k\to\infty}\left\|\sum_{i=m(T_{n_k})}^{m(T_{n_k}+t)-1}\beta(i)[G_{r_{n_k}}(\tilde{z}_{n_k}(t(i) - T_{n_k}), W_{i+1}) - G_{r_{n_k}}(\lambda_\infty(\tilde{y}_{n_k}(t(i) - T_{n_k})), \tilde{y}_{n_k}(t(i) - T_{n_k}), W_{i+1})]\right\|$$

$$\leq \lim_{k\to\infty}\sum_{i=m(T_{n_k})}^{m(T_{n_k}+t)-1}\beta(i)L(W_i)\|\tilde{x}_{n_k}(t(i) - T_{n_k}) - \lambda_\infty(\tilde{y}_{n_k}(t(i) - T_{n_k}))\|$$

$$\leq \lim_{k\to\infty}\epsilon(k)\sum_{i=m(T_{n_k})}^{m(T_{n_k}+t)-1}\beta(i)L(W_i) \qquad\qquad\text{(Lemma 13.6)}$$

$$\leq C\lim_{k\to\infty}\epsilon(k) \qquad\qquad\qquad\qquad\qquad\qquad (99)$$

$$= 0.$$

$\square$

The rest of the proof, including the verification that the other term diminishes is the same as liu et al and the fast timescale's analogous term.

Now, since we had chosen an arbitrary subsequence $\{f_{n_{k,0}}\}$ and it has a subsequence $\{f_{n_k}\}$ that converges to 0, by Lemma 10.1 we know that $\{f_n\}$ also converges to 0. Thus, the discretization error diminishes along the entire sequence. That is,

$$\lim_{k \to \infty} \|f_n(t)\| = 0 \tag{46}$$

for all $t \in [0, T+1)$.

### 13.5 COMPLETING THE PROOF

**Definition 11.** *We use the notation $\chi_c(t, y)$ to denote the solution to the ode*

$$\frac{dy(t)}{dt} = g_c(\lambda_\infty(y(t)), y(t))$$

*with the initial condition $y$.*

Note that under this notation, $y_n(t)$ (10) can be written $\chi_{r_n}(t, \tilde{y}_n(0))$.

**Lemma 13.8.** *Let $K \subset \mathbb{R}^{d_2}$ be compact. Given any $\epsilon > 0$, there exists a $T_\epsilon$ such that for all initial conditions $y \in K$, $\chi_\infty(t, y) \in B(0, \epsilon)$.*

*Proof.* By Lyapunov stability, there is a $\delta > 0$ such that any trajectory beginning within $B(0, \delta)$ stays within $\frac{\epsilon}{2}$ of the equilibrium 0.

For an initial condition $y$, let $T_y$ be a time at which the trajectory is within $\frac{\delta}{2}$ of the equilibrium. Let $y_1$ be some other initial condition. By definition, Lipschitzness, and the Gronwall inequality, we have

$$\|\chi_\infty(t, y) - \chi_\infty(t, y_1)\| \leq \|y - y_1\| + L(L_\lambda + 1) \int_0^t \|\chi_\infty(s, y) - \chi_\infty(s, y_1)\| ds$$

$$\leq \|y - y_1\| e^{L(L_\lambda + 1)T_y}$$

for all $t \leq T_y$.

So there is a neighborhood $V_y$ such that for all $y_1 \in V_y$, $\chi_\infty(T_y, y_1)$ is within $\delta$ of the equilibrium, which by Lyapunov stability implies that it will always be within $\epsilon$ of the equilibrium after $T_y$.

By compactness, we can cover the set $K$ by a finite number of such intervals and obtain a finite number of times $T_{y_1}, \ldots, T_{y_n}$ and then take the maximum time to be the value of $T_\epsilon$. $\square$

**Lemma 13.9.** *Let $[0, T]$ be a given time interval. Then,*

$$\|\chi_c(t, y) - \chi_\infty(t, y)\| \leq \epsilon(c) \cdot T e^{L(L_\lambda + 1)T}.$$

*for any initial $y \in \mathbb{R}^{d_2}$ and for all $t \in [0, T]$.*

*Proof.* We have

$$\chi_c(t, y) = y + \int_0^t g_c(\lambda_\infty(\chi_c(s, y)), \chi_c(s, y)) ds,$$

$$\chi_\infty(t, y) = y + \int_0^t g_\infty(\lambda_\infty(\chi_\infty(s, y)), \chi_\infty(s, y)) ds.$$

Let us define the error term:

$$e(t) = \|\chi_c(t, y) - \chi_\infty(t, y)\|.$$

We can bound $e(t)$ by two terms:

$$e(t) \leq \int_0^t \|g_c(\lambda_\infty(\chi_c(s, y)), \chi_c(s, y)) - g_c(\lambda_\infty(\chi_\infty(s, y)), \chi_\infty(s, y))\| ds$$

$$+ \int_0^t \|g_c(\lambda_\infty(\chi_\infty(s, y)), \chi_\infty(s, y)) - g_\infty(\lambda_\infty(\chi_\infty(s, y)), \chi_\infty(s, y))\| ds$$

To bound the first term, we use Lipschitzness:

$$\int_0^t \|g_c(\lambda_\infty(\chi_c(s,y)), \chi_c(s,y)) - g_c(\lambda_\infty(\chi_\infty(s,y)), \chi_\infty(s,y))\| ds$$

$$\leq L(L_\lambda + 1) \int_0^t \|\chi_c(t,y) - \chi_\infty(t,y)\| ds$$

$$= L \int_0^t e(s) ds.$$

To bound the second term, we use Lipschitzness and Assumption 3:

$$\int_0^t \|g_c(\lambda_\infty(\chi_\infty(s,y)), \chi_\infty(s,y)) - g_\infty(\lambda_\infty(\chi_\infty(s,y)), \chi_\infty(s,y))\| ds$$

$$\leq \int_0^t \epsilon(c) ds$$

$$\leq T\epsilon(c).$$

To conclude, we will use the Gronwall Inequality (Appendix 10.1):

$$e(t) \leq T\epsilon(c) + L(L_\lambda + 1) \int_0^t e(s) ds$$

$$\leq \epsilon(c) \cdot T e^{L(L_\lambda + 1)T}.$$

$\square$

Since in this section, $r_n$ is defined as the largest iterate norm seen so far, if it is bounded, then the iterate norms are bounded, verifying Theorem 2.

**Lemma 13.10.** *The sequence $r_n$ is bounded, creating a contradiction.*

*Proof.* By Lyapunov stability there is some $\delta$ with $\frac{1}{2(L_\lambda + 1)} > \delta > 0$ such that if $\chi_\infty(t,y) \in B(0,\delta)$ at some time $t$, then for all times $t' > t$, we have

$$\chi_\infty(t',y) \in B\left(0, \frac{1}{4(L_\lambda + 1)}\right). \tag{47}$$

By Lemma 13.8 we know that there is a $T_\delta$ such that when we set $T = T_\delta$,

$$\chi_\infty(T,y) \in B(0, \frac{\delta}{3}) \tag{48}$$

for all $y \in [-2,2]^{d_2}$.

By Lemma 13.9 we know that there is an $n_1$ such that for all $n > n_1$,

$$\|\chi_{r_n}(t,y) - \chi_\infty(t,y)\| < \frac{\delta}{3} \tag{49}$$

for all $y \in [-2,2]^{d_2}$ and $t \in [0, T_\delta]$.

By (46), we know that there is an $n_2$ such that for all $n > n_2$, we have

$$\|\tilde{z}_n(t) - (\lambda_\infty(\chi_{r_n}(t, \tilde{y}_n(0))), \chi_{r_n}(t, \tilde{y}_n(0)))\| < \frac{\delta}{3} \tag{50}$$

for all $t \in [0, T_\delta]$.

From (48), (49), and (50), we have

$$\tilde{y}_n(0) \in B(0, \delta)$$

for all $n \geq \max\{n_1, n_2\} + 1$. Thus, from (47), we have

$$\chi_\infty(t, \tilde{y}_n(0)) \in B\left(0, \frac{1}{4(L_\lambda + 1)}\right)$$

for all $t \in [0, T_\delta]$, which implies that

$$(\lambda_\infty(\chi_\infty(t, \tilde{y}_n(0))), \chi_\infty(t, \tilde{y}_n(0))) \in B\left(0, \frac{1}{4}\right).$$

Combining this with (50) gives us

$$\tilde{z}_n(t) \in B(0, \frac{1}{2}),$$

for all $n \geq \max\{n_1, n_2\} + 1$ and $t \in [0, T_\delta]$, which, by definition, prevents the sequence $r_n$ from increasing, contradicting the assumption that $r_n$ increases to infinity. $\qquad\square$

**Lemma 13.11.** *There are some constants $A, B$ such that*

$$\|y_l^{max}\| \leq A\left\|y_{m(T_n)}^{max}\right\| + B$$

*where $m(T_n) \leq l \leq m(T_{n+1})$. As a consequence, there are constants $C$ and $D$ such that*

$$\|z_l^{max}\| \leq C\left\|z_{m(T_n)}^{max}\right\| + D. \tag{51}$$

*This implies that $\|\tilde{z}_n(t)\|$ is bounded since for all $n$, $\|\tilde{z}_n(0)\| \leq 1$.*

*Proof.* We know that

$$\|y_l\| = \left\|y_{m(T_n)} + \sum_{i=m(T_n)}^{l} \beta(i)G(x_i, y_i, W_{i+1})\right\|$$

$$\leq \|y_n^{\max}\| + \sum_{i=m(T_n)}^{l} \beta(i)\|G(x_i, y_i, W_{i+1})\| \tag{52}$$

Since the second term of (52) is monotonically increasing in $l$, we know that the following holds:

$$\|y_l^{\max}\| \leq \left\|y_{m(T_n)}^{\max}\right\| + \sum_{i=m(T_n)}^{l} \beta(i)\|G(x_i, y_i, W_{i+1})\|$$

$$\leq \left\|y_{m(T_n)}^{\max}\right\| + \sum_{i=m(T_n)}^{l} \beta(i)L(W_{i+1})(\|x_i\| + \|y_i\|) + \sum_{i=m(T_n)}^{l} \beta(i)G(0, W_{i+1})$$

$$\leq \left\|y_{m(T_n)}^{\max}\right\| + \sum_{i=m(T_n)}^{l} \beta(i)L(W_{i+1})(\|x_i\| + \|y_i\|) + E \qquad (94)$$

$$\leq \left\|y_{m(T_n)}^{\max}\right\| + \sum_{i=m(T_n)}^{l} \beta(i)L(W_{i+1})(K(1 + \|y_i^{\max}\|) + \|y_i^{\max}\|) + E \qquad \text{(Theorem 1)}$$

$$\leq \left\|y_{m(T_n)}^{\max}\right\| + \sum_{i=m(T_n)}^{l} \beta(i)L(W_{i+1})(K + (K+1)\|y_i^{\max}\|) + E$$

$$\leq \left\|y_{m(T_n)}^{\max}\right\| + K\sum_{i=m(T_n)}^{l} \beta(i)L(W_{i+1}) + (K+1)\sum_{i=m(T_n)}^{l} \beta(i)L(W_{i+1})\|y_i^{\max}\| + E$$

$$\leq \left\|y_{m(T_n)}^{\max}\right\| + KC' + (K+1)\sum_{i=m(T_n)}^{l} \beta(i)L(W_{i+1})\|y_i^{\max}\| + E \qquad (106)$$

$$\leq \left(\left\|y_{m(T_n)}^{\max}\right\| + KC'\right) e^{(K+1)\sum_{i=m(T_n)}^{l} \beta(i)L(W_{i+1})} + E$$
$$\text{(by discrete Grownwall inequality in Appendix 10.3)}$$

$$\leq \left(\left\|y_{m(T_n)}^{\max}\right\| + KC'\right) e^{(K+1)C'} + E \qquad (106)$$

Now we show (51):

$$\|z_l^{max}\| \leq \|y_l^{\max}\| + K(1 + \|y_l^{\max}\|) \qquad \text{(Theorem 1)}$$
$$\leq K + (K+1)(A\left\|y_{m(T_n)}^{\max}\right\| + B)$$
$$\leq K + (K+1)(A\left\|z_{m(T_n)}^{max}\right\| + B)$$

$\square$

## 14 CONVERGENCE

### 14.1 FAST TIMESCALE CONVERGENCE

Returning to fast timescale definitions here.

**Corollary 2.** *Let Assumptions 1 - 6 hold. Then the iterates $\{z_n\}$ generated by (1) converge almost surely to a (sample path dependent) bounded invariant set of the ODE system*

$$\frac{\mathrm{d}x(t)}{\mathrm{d}t} = h(x(t), y(t)) \qquad (53)$$
$$\frac{\mathrm{d}y(t)}{\mathrm{d}t} = 0.$$

*Proof.* To prove convergence results on $t \in (-\infty, \infty)$ in Corollary 2, we fix an arbitrary sample path $\{z_0, \{W_i\}_{i=1}^{\infty}\}$. The stability results from Theorem 2 hold. To prove properties on $t \in (-\infty, \infty)$, we first fix an arbitrary $\tau > 0$ and show properties on $\forall t \in [-\tau, \tau]$.

**Definition 12.** $\forall n \in \mathbb{N}$, define $\bar{z}_n(t)$ as the solution to the ODE (53) in $(-\infty, \infty)$ with an initial condition

$$\bar{z}_n(0) = \bar{z}(t(n)).$$

$\bar{z}_n(t)$ can also be written as

$$\bar{z}_n(t) = \bar{z}(t(n)) + \int_0^t (h(\bar{z}_n(s)), 0)ds, \quad \forall t \in (-\infty, \infty). \tag{54}$$

We need to show that the error of Euler's discretization diminishes asymptotically. With (16) and (17), $\forall \tau > 0$, $\forall t \in [-\tau, \tau]$,

$$\bar{z}(t(n) + t) = z_{m(t(n)+t)} \tag{55}$$

$$= \begin{cases} \bar{z}(t(n)) + \sum_{i=n}^{m(t(n)+t)-1} (\alpha(i)H(\bar{z}(t(i)), W_{i+1}), \beta(i)G(\bar{z}(t(i)), W_{i+1})) & \text{if } t \geq 0 \\ \bar{z}(t(n)) - \sum_{i=m(t(n)+t)}^{n-1} (\alpha(i)H(\bar{z}(t(i)), W_{i+1}), \beta(i)G(\bar{z}(t(i)), W_{i+1})) & \text{if } t < 0. \end{cases}$$

Notably, the property (16) that $\forall t < 0, m(t) = 0$ in (55) ensures $\bar{z}(t(n) + t)$ is well-defined when $t(n) + t < 0$. Precisely speaking, $\forall \tau > 0$, $\forall t \in [-\tau, \tau]$, the discretization error is defined as

$$\bar{f}_n(t) \doteq \bar{z}(t(n) + t) - \bar{z}_n(t). \tag{56}$$

and we need $\bar{f}_n(t)$ to diminish to 0 as $n \to \infty$. To this end, we study the following three sequences of functions

$$\{\bar{z}(t(n) + t)\}_{n=0}^{\infty}, \{\bar{z}_n(t)\}_{n=0}^{\infty}, \{\bar{f}_n(t)\}_{n=0}^{\infty}.$$

Equicontinuity in the extended sense on the domain $(-\infty, \infty)$ is defined as following (Section 4.2.1 in Kushner & Yin (2003)).

**Definition 13.** A sequence of functions $\{\gamma_n : (-\infty, \infty) \to \mathbb{R}^K\}$ is equicontinuous in the extended sense on $(-\infty, \infty)$ if $\sup_n \|\gamma_n(0)\| < \infty$ and $\forall \tau > 0$, $\forall \epsilon > 0$, $\exists \delta > 0$ such that

$$\limsup_n \sup_{0 \leq |t_1 - t_2| \leq \delta, |t_1| \leq \tau, |t_2| \leq \tau} \|\gamma_n(t_1) - \gamma_n(t_2)\| \leq \epsilon.$$

We show $\{\bar{z}(t(n) + t)\}$, $\{\bar{z}_n(t)\}$ and $\{\bar{f}_n(t)\}$ are all equicontinuous in the extended sense.

**Lemma 14.1.** The three sequences of functions $\{\bar{z}(t(n) + t)\}_{n=0}^{\infty}$, $\{\bar{z}_n(t)\}_{n=0}^{\infty}$, and $\{\bar{f}_n(t)\}_{n=0}^{\infty}$ are all equicontinuous in the extended sense on $t \in (-\infty, \infty)$.

To prove those lemmas, we need the Gronwall inequality in the reverse time in Appendix 10.2. Compared to lemmas in the main text which have the domain $t \in [0, T + 1)$, lemmas in this section have similar proofs because we first fix an arbitrary $\tau$ and prove properties on the domain $t \in [-\tau, \tau]$. We omit proofs for Lemma 14.1 because they are extremely similar to the proofs of equicontinuity in the fast timescale. Similar to Lemma 5.1, we now construct a particular subsequence of interest.

**Lemma 14.2.** There exists a subsequence $\{n_k\}_{k=0}^{\infty} \subseteq \{0, 1, 2, \dots\}$ and some continuous functions $\bar{f}^{\lim}(t)$ and $\bar{z}^{\lim}(t)$ such that $\forall \tau$, $\forall t \in [-\tau, \tau]$,

$$\lim_{k \to \infty} \bar{f}_{n_k}(t) = \bar{f}^{\lim}(t),$$

$$\lim_{k \to \infty} \bar{z}(T_{n_k} + t) = \bar{z}^{\lim}(t),$$

where both convergences are uniform in $t$ on $[-\tau, \tau]$. Furthermore, let $z^{\lim}(t)$ denote the unique solution to the ODE (53) with the initial condition

$$z^{\lim}(0) = \bar{z}^{\lim}(0),$$

in other words,

$$z^{\lim}(t) = \bar{z}^{\lim}(0) + \int_0^t (h(\bar{z}^{\lim}(s)), 0)ds.$$

Then $\forall \tau$, $\forall t \in [-\tau, \tau]$, we have

$$\lim_{k \to \infty} \bar{z}_{n_k}(t) = z^{\lim}(t),$$

where the convergence is uniform in $t$ on $[-\tau, \tau]$.

Its proof is very similar to the proof of Lemma 5.1 and is omitted. We use the subsequence $\{n_k\}$ intensively in the remaining proofs. Recall that $\bar{f}_n(t)$ denotes the discretization error between $\bar{z}(t(n) + t)$ and $\bar{z}_n(t)$. We now proceed to prove that this discretization error diminishes along $\{n_k\}$. In particular, we aim to prove that $\forall \tau$, $\forall t \in [-\tau, \tau]$,

$$\lim_{k \to \infty} \left\| \bar{f}_{n_k}(t) \right\| = \left\| \bar{f}^{\lim}(t) \right\| = 0.$$

This means $\bar{z}(t(n_k) + t)$ is close to $\bar{z}_{n_k}(t)$ as $k \to \infty$. For $t \in (0, \tau]$, the proof for this part is the same as the proof we have done in Section 12.4. Thus, we only discuss the proof for $t \in [-\tau, 0]$. $\forall \tau$, $\forall t \in [-\tau, 0]$,

$$\lim_{k \to \infty} \left\| \bar{f}_{n_k}(t) \right\|$$

$$= \lim_{k \to \infty} \left\| \bar{z}(t(n_k)) - \sum_{i=m(t(n_k)+t)}^{n_k - 1} (\alpha(i)H(\bar{z}(t(i)), W_{i+1}), \beta(i)G(\bar{z}(t(i)), W_{i+1})) - \bar{z}_{n_k}(t) \right\|$$

$$\text{(by (55) and (56))}$$

$$= \lim_{k \to \infty} \left\| -\sum_{i=m(t(n_k)+t)}^{n_k - 1} (\alpha(i)H(\bar{z}(t(i)), W_{i+1}), \beta(i)G(\bar{z}(t(i)), W_{i+1})) - \int_0^t (h(\bar{z}_{n_k}(s)), 0)ds \right\|$$

$$\text{(by (54))}$$

$$\leq \lim_{k \to \infty} \left\| -\sum_{i=m(t(n_k)+t)}^{n_k - 1} \alpha(i)H(\bar{z}(t(i)), W_{i+1}) - \int_0^t h(\bar{z}^{\lim}(s))ds \right\|$$

$$+ \lim_{k \to \infty} \left\| -\sum_{i=m(t(n_k)+t)}^{n_k - 1} \beta(i)G(\bar{z}(t(i)), W_{i+1}) \right\|$$

$$+ \lim_{k \to \infty} \left\| \int_0^t h(\bar{z}^{\lim}(s))ds - \int_0^t h(\bar{z}_{n_k}(s))ds \right\|. \tag{57}$$

The second term in the RHS of (57) is 0.

**Lemma 14.3.** $\forall \tau$, $\forall t \in [-\tau, 0]$,

$$\lim_{k \to \infty} \left\| -\sum_{i=m(t(n_k)+t)}^{n_k - 1} \beta(i)G(\bar{z}(t(i)), W_{i+1}) \right\| = 0.$$

The proof is very similar to the proof of Lemma 12.2 so is omitted.

The first term in the RHS of (57) is also 0.

**Lemma 14.4.** $\forall \tau$, $\forall t \in [-\tau, 0]$,

$$\lim_{k \to \infty} \left\| -\sum_{i=m(t(n_k)+t)}^{n_k - 1} \alpha(i)H(\bar{z}(t(i)), W_{i+1}) - \int_0^t h(\bar{z}^{\lim}(s))ds \right\| = 0.$$

Its proof is very similar to the proof of Lemma 16.7 and is omitted. This convergence is also simpler than (74) because here we have only a single $(H, h)$. But in (74), we have a sequence $\{(H_{n_k}, h_{n_k})\}$, for which we have to split it to a double limit (75) and then invoke the Moore-Osgood theorem to reduce it to the single $(H, h)$ case.

Lemma 14.4 confirms that the first term in the RHS of (57) is 0. Moreover, it also enables us to rewrite $\bar{z}^{\lim}(t)$ from a summation form to an integral form. $\forall \tau, \forall t \in [-\tau, 0]$

$$\bar{z}^{\lim}(t)$$

$$= \lim_{k \to \infty} \bar{z}(t(n_k)) - \sum_{i=m(t(n_k)+t)}^{n_k-1} (\alpha(i)H(\bar{z}(t(i)), W_{i+1}), \beta(i)G(\bar{z}(t(i)), W_{i+1}))$$

$$= \lim_{k \to \infty} \bar{z}(t(n_k)) + \int_0^t (h(\bar{z}^{\lim}(s)), 0)ds. \qquad \text{(by Lemma 14.4)}$$

Thus, we can show the following diminishing discretization error.

**Lemma 14.5.** $\forall \tau, \forall t \in [-\tau, \tau]$,

$$\lim_{k \to \infty} \left\| \bar{f}_{n_k}(t) \right\| = 0.$$

*Moreover, the convergence is uniform in $t$ on $[-\tau, \tau]$.*

Its proof is very similar to the proof of Lemma 16.9 and is omitted. This immediately implies that for any $t \in (-\infty, \infty)$

$$\lim_{k \to \infty} \bar{z}(t(n_k) + t) = z^{\lim}(t). \qquad (58)$$

Theorem 2 then yields that

$$\sup_{t \in (-\infty, \infty)} \left\| z^{\lim}(t) \right\| < \infty.$$

Let $Z$ be the set of limit points of $\{z_n\}$. By Theorem 2, $\sup_n \|z_n\| < \infty$, so $Z$ is bounded and nonempty. We now prove $Z$ is an invariant set of the ODE (53). For any $z \in Z$, there exists a subsequence $\{z_{n_k}\}$ such that

$$\lim_{k \to \infty} z_{n_k} = z.$$

Since $\{\bar{z}(t(n_k) + t)\}$ is equicontinuous in the extended sense, following the way we arrive at (58), we can construct a subsequence $\{n_k'\} \subseteq \{n_k\}$ such that

$$\lim_{k \to \infty} \bar{z}(t(n_k') + t) = z_*^{\lim}(t), \qquad (59)$$

where $z_*^{\lim}(t)$ is a solution to the ODE (53) and $z_*^{\lim}(0) = z$. The remaining is to show that $z_*^{\lim}(t)$ lies entirely in $Z$. For any $t \in (-\infty, \infty)$, by the piecewise constant nature of $\bar{z}$ in (55), the above limit (59) implies that there exists a subsequence of $\{z_n\}$ that converges to $z_*^{\lim}(t)$, indicating $z_*^{\lim}(t) \in Z$ by the definition of the limit set. We now have proved $\forall z \in Z$, there exists a solution $z_*^{\lim}(t)$ to the ODE (53) such that $z_*^{\lim}(0) = z$ and $\forall t \in (-\infty, \infty), z_*^{\lim}(t) \in Z$. This means $Z$ is an invariant set, by definition. In particular, $Z$ is a bounded invariant set.

We now prove that $\{z_n\}$ converges to $Z$. Let $\{z_{n_k}\}$ be any convergent subsequence of $\{z_n\}$ with its limit denoted by $z$. We must have $z \in Z$ by the definition of the limit set. So we have proved that all convergent subsequences of $\{z_n\}$ converge to a point in the bounded invariant set $Z$. If $\{z_n\}$ does not converge to $Z$, there must exist a subsequence $\left\{ z_{n_k'} \right\}$ such that $\left\{ z_{n_k'} \right\}$ is always away from $Z$ by some small $\epsilon_0 > 0$, i.e., $\forall k$,

$$\inf_{z \in Z} \left\| z_{n_k'} - z \right\| \geq \epsilon_0. \qquad (60)$$

But $\left\{ z_{n_k'} \right\}$ is bounded, so by the Bolzano-Weierstrass Theorem, it must have a convergent subsequence, which, by the definition of the limit set, converges to some point in $Z$. This contradicts (60). So we must have $\{z_n\}$ converge to $Z$, which is a bounded invariant set of the ODE (53). This completes the proof. $\qquad \square$

Since the invariant set of (53) is $\left\{ (\lambda(y), y) : y \in \mathbb{R}^{d_2} \right\}$, we have

$$\lim_{n \to \infty} \|x_n - \lambda(y_n)\| = 0. \qquad (61)$$

We will use this fact to now establish convergence to the unique equilibrium.

## 14.2 SLOW TIMESCALE CONVERGENCE

Returning to slow timescale definitions here.

**Corollary 3.** *Let Assumptions 1 - 6 hold. Then the iterates $\{y_n\}$ generated by (1) converge almost surely to a (sample path dependent) bounded invariant set of the ODE*

$$\frac{\mathrm{d}y(t)}{\mathrm{d}t} = g(\lambda(y(t)), y(t)) \tag{62}$$

*Proof.* To prove convergence results on $t \in (-\infty, \infty)$ in Corollary 3, we fix an arbitrary sample path $\{z_0, \{W_i\}_{i=1}^{\infty}\}$. The stability results from Theorem 2 hold. To prove properties on $t \in (-\infty, \infty)$, we first fix an arbitrary $\tau > 0$ and show properties on $\forall t \in [-\tau, \tau]$.

**Definition 14.** $\forall n \in \mathbb{N}$, *define $\bar{y}_n(t)$ as the solution to the ODE (62) in $(-\infty, \infty)$ with an initial condition*

$$\bar{y}_n(0) = \bar{y}(t(n)).$$

$\bar{y}_n(t)$ can also be written as

$$\bar{y}_n(t) = \bar{y}(t(n)) + \int_0^t g(\lambda(\bar{y}_n(s)), \bar{y}_n(s))ds, \quad \forall t \in (-\infty, \infty). \tag{63}$$

We need to show that the error of Euler's discretization diminishes asymptotically. With (16) and (17), $\forall \tau > 0$, $\forall t \in [-\tau, \tau]$,

$$\bar{z}(t(n) + t) = z_{m(t(n)+t)} \tag{64}$$

$$= \begin{cases} \bar{z}(t(n)) + \sum_{i=n}^{m(t(n)+t)-1} (\alpha(i)H(\bar{z}(t(i)), W_{i+1}), \beta(i)G(\bar{z}(t(i)), W_{i+1})) & \text{if } t \geq 0 \\ \bar{z}(t(n)) - \sum_{i=m(t(n)+t)}^{n-1} (\alpha(i)H(\bar{z}(t(i)), W_{i+1}), \beta(i)G(\bar{z}(t(i)), W_{i+1})) & \text{if } t < 0. \end{cases}$$

Notably, the property (16) that $\forall t < 0, m(t) = 0$ in (64) ensures $\bar{z}(t(n) + t)$ is well-defined when $t(n) + t < 0$. Precisely speaking, $\forall \tau > 0$, $\forall t \in [-\tau, \tau]$, the discretization error is defined as

$$\bar{f}_n(t) \doteq \bar{y}(t(n) + t) - \bar{y}_n(t). \tag{65}$$

and we need $\bar{f}_n(t)$ to diminish to 0 as $n \to \infty$. To this end, we study the following three sequences of functions

$$\{\bar{y}(t(n) + t)\}_{n=0}^{\infty}, \{\bar{y}_n(t)\}_{n=0}^{\infty}, \{\bar{f}_n(t)\}_{n=0}^{\infty}.$$

We show $\{\bar{y}(t(n) + t)\}$, $\{\bar{y}_n(t)\}$ and $\{\bar{f}_n(t)\}$ are all equicontinuous in the extended sense on the domain $(-\infty, \infty)$.

**Lemma 14.6.** *The three sequences of functions $\{\bar{y}(t(n) + t)\}_{n=0}^{\infty}$, $\{\bar{y}_n(t)\}_{n=0}^{\infty}$, and $\{\bar{f}_n(t)\}_{n=0}^{\infty}$ are all equicontinuous in the extended sense on $t \in (-\infty, \infty)$.*

To prove those lemmas, we need the Gronwall inequality in the reverse time in Appendix 10.2. Compared to lemmas in the main text which have the domain $t \in [0, T + 1)$, lemmas in this section have similar proofs because we first fix an arbitrary $\tau$ and prove properties on the domain $t \in [-\tau, \tau]$. We omit proofs for Lemma 14.6 because they are extremely similar to the proofs of equicontinuity in the slow timescale. Similar to Lemma 13.2, we now construct a particular subsequence of interest.

**Lemma 14.7.** *There exists a subsequence $\{n_k\}_{k=0}^{\infty} \subseteq \{0, 1, 2, \dots\}$ and some continuous functions $\bar{f}^{\mathrm{lim}}(t)$ and $\bar{y}^{\mathrm{lim}}(t)$ such that $\forall \tau$, $\forall t \in [-\tau, \tau]$,*

$$\lim_{k \to \infty} \bar{f}_{n_k}(t) = \bar{f}^{\mathrm{lim}}(t),$$

$$\lim_{k \to \infty} \bar{y}(T_{n_k} + t) = \bar{y}^{\mathrm{lim}}(t),$$

*where both convergences are uniform in t on $[-\tau, \tau]$. Furthermore, let $y^{\mathrm{lim}}(t)$ denote the unique solution to the ODE (62) with the initial condition*

$$y^{\mathrm{lim}}(0) = \bar{y}^{\mathrm{lim}}(0),$$

*in other words,*

$$y^{\lim}(t) = \bar{y}^{\lim}(0) + \int_0^t g(\lambda(\bar{y}^{\lim}), \bar{y}^{\lim}(s))ds.$$

*Then $\forall \tau$, $\forall t \in [-\tau, \tau]$, we have*

$$\lim_{k \to \infty} \bar{y}_{n_k}(t) = y^{\lim}(t),$$

*where the convergence is uniform in $t$ on $[-\tau, \tau]$.*

Its proof is very similar to the proof of Lemma 13.2 and is omitted. We use the subsequence $\{n_k\}$ intensively in the remaining proofs. Recall that $\bar{f}_n(t)$ denotes the discretization error between $\bar{y}(t(n) + t)$ and $\bar{y}_n(t)$. We now proceed to prove that this discretization error diminishes along $\{n_k\}$. In particular, we aim to prove that $\forall \tau$, $\forall t \in [-\tau, \tau]$,

$$\lim_{k \to \infty} \left\| \bar{f}_{n_k}(t) \right\| = \left\| \bar{f}^{\lim}(t) \right\| = 0.$$

This means $\bar{y}(t(n_k) + t)$ is close to $\bar{y}_{n_k}(t)$ as $k \to \infty$. For $t \in (0, \tau]$, the proof for this part is the same as the proof we have done in Section 13.4. Thus, we only discuss the proof for $t \in [-\tau, 0]$. $\forall \tau$, $\forall t \in [-\tau, 0]$,

$$\lim_{k \to \infty} \left\| \bar{f}_{n_k}(t) \right\|$$

$$= \lim_{k \to \infty} \left\| \bar{y}(t(n_k)) - \sum_{i=m(t(n_k)+t)}^{n_k-1} \beta(i) G(\bar{z}(t(i)), W_{i+1}) - \bar{y}_{n_k}(t) \right\| \qquad \text{(by (64) and (65))}$$

$$= \lim_{k \to \infty} \left\| - \sum_{i=m(t(n_k)+t)}^{n_k-1} \beta(i) G(\bar{z}(t(i)), W_{i+1}) - \int_0^t g(\lambda(\bar{y}_{n_k}), \bar{y}_{n_k})ds \right\| \qquad \text{(by (63))}$$

$$\leq \lim_{k \to \infty} \left\| - \sum_{i=m(t(n_k)+t)}^{n_k-1} \beta(i) G(\bar{z}(t(i)), W_{i+1}) + \sum_{i=m(t(n_k)+t)}^{n_k-1} \beta(i) G(\lambda(\bar{y}(t(i))), \bar{y}(t(i)), W_{i+1}) \right\|$$

$$+ \lim_{k \to \infty} \left\| - \sum_{i=m(t(n_k)+t)}^{n_k-1} \beta(i) G(\lambda(\bar{y}(t(i))), \bar{y}(t(i)), W_{i+1}) - \int_0^t g(\lambda(\bar{y}_{n_k}), \bar{y}_{n_k})ds \right\|. \qquad (66)$$

The first term in the RHS of (66) is 0.

**Lemma 14.8.** $\forall \tau$, $\forall t \in [-\tau, 0]$,

$$\lim_{k \to \infty} \left\| - \sum_{i=m(t(n_k)+t)}^{n_k-1} \beta(i) G(\bar{z}(t(i)), W_{i+1}) + \sum_{i=m(t(n_k)+t)}^{n_k-1} \beta(i) G(\lambda(\bar{y}(t(i))), \bar{y}(t(i)), W_{i+1}) \right\| = 0.$$

Its proof is very similar to the proof of Lemma 13.7 (except that we use (61) instead of Lemma 13.6) and is omitted. This convergence is also simpler than (74) because here we have only a single $(G, g)$.

**Lemma 14.9.** $\forall \tau$, $\forall t \in [-\tau, \tau]$,

$$\lim_{k \to \infty} \left\| \bar{f}_{n_k}(t) \right\| = 0.$$

*Moreover, the convergence is uniform in $t$ on $[-\tau, \tau]$.*

Its proof is very similar to the proof of the diminishing discretization error in the slow timescale. This immediately implies that for any $t \in (-\infty, \infty)$

$$\lim_{k \to \infty} \bar{z}(t(n_k) + t) = (\lambda(y^{\lim}(t)), y^{\lim}(t)). \tag{67}$$

Theorem 2 then yields that

$$\sup_{t \in (-\infty, \infty)} \left\| (\lambda(y^{\lim}(t)), y^{\lim}(t)) \right\| < \infty.$$

Let $Y$ be the set of limit points of $\{y_n\}$. By Theorem 2, $\sup_n \|y_n\| < \infty$, so $Y$ is bounded and nonempty. We now prove $Y$ is an invariant set of the ODE (62). For any $y \in Y$, there exists a subsequence $\{y_{n_k}\}$ such that

$$\lim_{k \to \infty} y_{n_k} = y.$$

Since $\{\bar{y}(t(n_k) + t)\}$ is equicontinuous in the extended sense, following the way we arrive at (67), we can construct a subsequence $\{n_k'\} \subseteq \{n_k\}$ such that

$$\lim_{k \to \infty} \bar{y}(t(n_k') + t) = y_*^{\lim}(t), \tag{68}$$

where $y_*^{\lim}(t)$ is a solution to the ODE (62) and $y_*^{\lim}(0) = y$. The remaining is to show that $y_*^{\lim}(t)$ lies entirely in $Y$. For any $t \in (-\infty, \infty)$, by the piecewise constant nature of $\bar{y}$ in (64), the above limit (68) implies that there exists a subsequence of $\{y_n\}$ that converges to $y_*^{\lim}(t)$, indicating $y_*^{\lim}(t) \in Y$ by the definition of the limit set. We now have proved $\forall y \in Y$, there exists a solution $y_*^{\lim}(t)$ to the ODE (62) such that $y_*^{\lim}(0) = y$ and $\forall t \in (-\infty, \infty), y_*^{\lim}(t) \in Y$. This means $Y$ is an invariant set, by definition. In particular, $Y$ is a bounded invariant set.

We now prove that $\{y_n\}$ converges to $Y$. Let $\{y_{n_k}\}$ be any convergent subsequence of $\{y_n\}$ with its limit denoted by $y$. We must have $y \in Y$ by the definition of the limit set. So we have proved that all convergent subsequences of $\{y_n\}$ converge to a point in the bounded invariant set $Y$. If $\{y_n\}$ does not converge to $Y$, there must exist a subsequence $\left\{y_{n_k'}\right\}$ such that $\left\{y_{n_k'}\right\}$ is always away from $Y$ by some small $\epsilon_0 > 0$, i.e., $\forall k$,

$$\inf_{y \in Y} \left\| y_{n_k'} - y \right\| \geq \epsilon_0. \tag{69}$$

But $\left\{y_{n_k'}\right\}$ is bounded, so by the Bolzano-Weierstrass Theorem, it must have a convergent subsequence, which, by the definition of the limit set, converges to some point in $Y$. This contradicts (69). So we must have $\{y_n\}$ converge to $Y$, which is a bounded invariant set of the ODE (62). This completes the proof. □

Since the invariant set of (62) is the singleton containing $y^*$, the equilibrium of the map $\lambda$, we have

$$\lim_{n \to \infty} \|y_n - y^*\| = 0.$$

Combined with the fast convergence, we have

$$\lim_{n \to \infty} \|z_n - (\lambda(y^*), y^*)\| = 0.$$

## 15 CONVERGENCE OF TDC WITH ELIGIBILITY TRACE

TDC was first proposed in Sutton et al. (2008b) as a modification of gradient temporal difference learning (GTD) (Sutton et al., 2008a). GTD was developed to break the deadly triad, divergence that can arise when combining off-policy learning, function approximation, and bootstrapping, each of which are critical components in successful RL algorithms. While GTD mitigates the deadly triad, it is slow. TDC, on the other hand, is nearly as fast as regular TD learning and converges. It is also a two-timescale algorithm, as the gradient correction runs on a faster timescale. Although vanilla TDC is known to converge, the best prior work was only able to establish the convergence of projected variants of TDC with eligibility traces (Yu, 2017).

We must also explain the important of eligibility traces: they are a powerful tool for credit assignation, a critical challenge in RL, and have been a fundamental part of RL since the inception of the field (Barto & Sutton, 1981). Although eligibility traces are useful, they introduce difficulties in analysis. Even if the state space of the Markov chain $\{(S_t, A_t)\}$ is finite, with eligibility traces, we have to instead consider the chain $\{(S_t, A_t, e_t)\}$, which now evolves in an uncountable space. And, in the case of off-policy learning, the importance sampling ratio can cause the state space to be unbounded as well. Our results, therefore, are the first to be able to handle the important case of off-policy RL algorithms with eligibility traces. We demonstrate this with TDC.

**Assumption 15.1.** *Both the state space $\mathcal{S}$ and the action space $\mathcal{A}$ are finite. The Markov chain $\{S_t\}$ induced by the behavior policy $\mu$ is irreducible, and $\mu(a|s) > 0$ for all $s, a$.*

TDC with eligibility trace is defined as follows:

$$
\begin{aligned}
e_t =& \lambda\gamma\rho_{t-1}e_{t-1} + \phi_t, \\
\delta_t =& R_{t+1} + \gamma\phi_{t+1}^\top\theta_t - \phi_t^\top\theta_t, \\
\nu_{t+1} =& \nu_t + \alpha_t \left(\rho_t\delta_t e_t - \phi_t\phi_t^\top\nu_t\right), \\
\theta_{t+1} =& \theta_t + \beta_t\rho_t(\phi_t - \gamma\phi_{t+1})e_t^\top\nu_t.
\end{aligned}
$$

We can more compactly express the updates with the following equations:

$$
\nu_{t+1} = \nu_t + \alpha_t \left( \begin{bmatrix} -\phi_t\phi_t^\top & \rho_t e_t(\gamma\phi_{t+1} - \phi_t)^\top \end{bmatrix} \begin{bmatrix} \nu_t \\ \theta_t \end{bmatrix} + [\rho_t R_{t+1} e_t] \right)
$$

$$
\theta_{t+1} = \theta_t + \beta_t \left( \begin{bmatrix} -(\gamma\phi_{t+1} - \phi_t)\rho_t e_t^\top & 0 \end{bmatrix} \begin{bmatrix} \nu_t \\ \theta_t \end{bmatrix} \right).
$$

Now we define the augmented Markov chain $\{W_t\}$ as

$$
W_{t+1} \doteq (S_t, A_t, S_{t+1}, e_t), \quad t = 0, 1, \ldots.
$$

We also define shorthands

$$
\begin{aligned}
x \doteq& \nu, x_t \doteq \nu_t, y \doteq \theta, y_t \doteq \theta_t \\
w \doteq& (s, a, s', e), \\
A(w) \doteq& \rho(s, a)e(\gamma\phi(s') - \phi(s))^\top, \\
b(w) \doteq& \rho(s, a)r(s, a)e, \\
C(w) \doteq& \phi(s)\phi(s)^\top, \\
H(x, y, w) \doteq& \begin{bmatrix} -C(w) & A(w) \end{bmatrix} \begin{bmatrix} x \\ y \end{bmatrix} + b(w), \\
G(x, y, w) \doteq& \begin{bmatrix} -A(w)^\top & 0 \end{bmatrix} \begin{bmatrix} x \\ y \end{bmatrix}.
\end{aligned}
$$

Then TDC can be expressed as

$$
\begin{aligned}
x_{t+1} =& x_t + \alpha_t H(x_t, y_t, W_{t+1}), \\
y_{t+1} =& y_t + \beta_t G(x_t, y_t, W_{t+1}),
\end{aligned}
$$

which reduces to the form of (1) and (2).

**Lemma 15.1.** *(Theorem 2.1 from Yu (2017)) If Assumption 15.1 holds, then*

(i) $\{W_t\}$ *has a unique invariant probability measure, $d_{\mathcal{W}}$.*

(ii) *The expectation with respect to the stationary distribution, $\mathbb{E}_{w \sim d_{\mathcal{W}}}[\gamma(w)] < \infty$ when $\gamma(w)$ is Lipschitz in the trace variable $e$. Additionally, (LLN) holds for $\gamma$.*

Yu (2017) also shows that

$$A \doteq \mathbb{E}_{w \sim d_{\mathcal{W}}} [A(w)] = \Phi^\top D_{\mathcal{W}}(P_{\pi,\lambda} - I)\Phi,$$
$$b \doteq \mathbb{E}_{w \sim d_{\mathcal{W}}} [b(w)] = \Phi^\top D_{\mathcal{W}} r_{\pi,\lambda},$$
$$C \doteq \mathbb{E}_{w \sim d_{\mathcal{W}}} [C(w)] = \Phi^\top D_{\mathcal{W}}\Phi,$$

where we use $D_{\mathcal{W}}$ to denote the diagonal matrix whose diagonal entry is $d_{\mathcal{W}}$.

**Theorem 15.1** (**Theorem 3 Restated**). *Assume A is nonsingular (without this assumption, there is no solution so the algorithm itself would be ill-posed). Then, TDC with eligibility trace converges.*

*Proof.* We show that all the assumptions are satisfied such that Corollary 1 applies.

Assumption 1 follows immediately from Lemma 15.1.

The assumption 2 follows immediately from the choice of appropriate learning rates.

For Assumption 3, define

$$H_\infty(x, y, w) \doteq [-C(w) \quad A(w)] \begin{bmatrix} x \\ y \end{bmatrix},$$
$$G_\infty(x, y, w) \doteq [-A(w)^\top \quad 0] \begin{bmatrix} x \\ y \end{bmatrix}.$$

Then we have

$$H_c(x, y, w) - H_\infty(x, y, w) = \frac{b(w)}{c},$$
$$G_c(x, y, w) - G_\infty(x, y, w) = 0$$

After noticing

$$\|b((s, a, s', e)) - b((s, a, s', e'))\| = \rho(s, a)|r(s, a)|\|e - e'\|, \quad \forall s, a, s', e, e',$$

Assumption 3 follows immediately from Lemma 15.1.

For Assumption 4, it can be easily verified that $H(x, y, w), G(x, y, w), H_\infty(x, y, w)$, and $G_\infty(x, y, w)$ are Lipschitz continuous in $(x, y)$ for each $w$.

Since $A(w), b(w), C(w)$ are Lipschitz continuous in $e$ for each $(s, a, s')$, Lemma 15.1 implies that

$$h(x, y) = [-C \quad A] \begin{bmatrix} x \\ y \end{bmatrix} + b,$$
$$h_\infty(x, y) = [-C \quad A] \begin{bmatrix} x \\ y \end{bmatrix},$$
$$g(x, y) = g_\infty(x, y) = [-A^\top \quad 0] \begin{bmatrix} x \\ y \end{bmatrix},$$
$$L = \max \{ \|[-C \quad A]\|, \|[-A^\top \quad 0]\| \}.$$

Assumption 4 then follows.

For Assumption 5, we have

$$\|h_c(x) - h_\infty(x)\| = \frac{1}{c}\|b\|,$$

so the uniform convergence of $h_c$ to $h_\infty$ follows immediately, and we already know $g(x, y) = g_\infty(x, y)$.

Since $\Phi$ has full column rank and $D_{\mathcal{W}}$ is positive definite, $C = \Phi^\top D_{\mathcal{W}}\Phi$ is positive definite, so $-C$ is negative definite. This implies the global asymptotic stability of these two ODEs:

$$\frac{\mathrm{d}x(t)}{\mathrm{d}t} = -Cx(t) + Ay + b, \quad \frac{\mathrm{d}x(t)}{\mathrm{d}t} = -Cx(t) + Ay.$$

The unique globally asymptotically stable equilibrium of the first is $C^{-1}(Ay + b)$, and of the second is $C^{-1}Ay$. This means that we can define $\lambda(y) = C^{-1}(Ay + b)$ and $\lambda_\infty(y) = C^{-1}Ay$. Note that $\lambda_\infty(cy) = c\lambda_\infty(y)$ as required.

Since $A$ is nonsingular and $C$ is positive definite, $-A^\top C^{-1} A$ is negative definite. This implies the global asymptotic stability of the following two ODEs:

$$\frac{dy(t)}{dt} = -A^\top \lambda(y(t)) = -A^\top C^{-1}(Ay(t) + b), \quad \frac{dy(t)}{dt} = -A^\top \lambda_\infty(y(t)) = -A^\top C^{-1}Ay(t).$$

The unique globally asymptotically stable equilibrium of the first is $-A^{-1}b$, and of the second is $0$. Assumption 5 then follows.

Assumption 6 follows immediately from Lemma 15.1 and Assumption 2.

Corollary 1 then implies that

$$\lim_{t \to \infty} x_t = 0 \quad \text{a.s.}$$
$$\lim_{t \to \infty} y_t = -A^{-1}b \quad \text{a.s.}$$

which completes the proof. $\qquad\square$

## 16 Technical Lemmas

### 16.1 Asymptotic Rate of Change of Functions in Assumption 6

**Lemma 16.1.** *Let Assumptions 1, 2, 4, and 6 hold. Then the asymptotic rate of change of the functions that could be represented by $\gamma$ in Assumption 6 is 0, i.e., for any fixed $\tau > 0$ and $x$, it holds that*

$$\limsup_n \sup_{-\tau \le t_1 \le t_2 \le \tau} \left\| \sum_{i=m(t(n)+t_1)}^{m(t(n)+t_2)-1} \alpha(i) \left[ H(x, y, W_{i+1}) - h(x, y) \right] \right\| = 0 \quad a.s.,$$

$$\limsup_n \sup_{-\tau \le t_1 \le t_2 \le \tau} \left\| \sum_{i=m(t(n)+t_1)}^{m(t(n)+t_2)-1} \alpha(i) \left[ G(x, y, W_{i+1}) - g(x, y) \right] \right\| = 0 \quad a.s.,$$

$$\limsup_n \sup_{-\tau \le t_1 \le t_2 \le \tau} \left\| \sum_{i=m(t(n)+t_1)}^{m(t(n)+t_2)-1} \alpha(i) [L_b(W_{i+1}) - L_b] \right\| = 0 \quad a.s.,$$

$$\limsup_n \sup_{-\tau \le t_1 \le t_2 \le \tau} \left\| \sum_{i=m(t(n)+t_1)}^{m(t(n)+t_2)-1} \alpha(i) [L(W_{i+1}) - L] \right\| = 0 \quad a.s.$$

*and we can replace $\alpha(i)$ with $\beta(i)$ in any of the above.*

The proofs of these results are very similar to the proof of Lemma 9 in Liu et al. (2025) and so are omitted due to their length.

### 16.2 A Uniform Convergence of $H_c, G_c$ to $H_\infty, G_\infty$

**Lemma 16.2.** *Let Assumptions 1, 2, 4, and 6 hold. It then holds that*

$$\lim_{c \to \infty} \sup_{z \in \mathcal{B}} \sup_n \sup_{t \in [0,T]} \left\| \sum_{i=m(T_n)}^{m(T_n+t)-1} \alpha(i) \left[ H_c(z, W_{i+1}) - H_\infty(z, W_{i+1}) \right] \right\| = 0 \quad a.s.,$$

$$\lim_{c \to \infty} \sup_{z \in \mathcal{B}} \sup_n \sup_{t \in [0,T]} \left\| \sum_{i=m(T_n)}^{m(T_n+t)-1} \alpha(i) \left[ G_c(z, W_{i+1}) - G_\infty(z, W_{i+1}) \right] \right\| = 0 \quad a.s.,$$

*where $\mathcal{B}$ denotes an arbitrary compact set of $\mathbb{R}^{d_1+d_2}$. Again, $\alpha(i)$ can be replaced with $\beta(i)$.*

*Proof.* Fix an arbitrary sample path $\{x_0, y_0, \{W_i\}_{i=1}^{\infty}\}$. Use $\mathcal{B}$ to denote an arbitrary compact subset of $\mathbb{R}^{d_1+d_2}$.

$$
\limsup_{c \to \infty} \sup_{z \in \mathcal{B}} \sup_{n} \sup_{t \in [0,T]} \left\| \sum_{i=m(T_n)}^{m(T_n+t)-1} \alpha(i) \left[ H_c(z, W_{i+1}) - H_\infty(z, W_{i+1}) \right] \right\|
$$

$$
= \limsup_{c \to \infty} \sup_{z \in \mathcal{B}} \sup_{n} \sup_{t \in [0,T]} \left\| \sum_{i=m(T_n)}^{m(T_n+t)-1} \alpha(i) \kappa(c) b(z, W_{i+1}) \right\| \qquad \text{(by (11))}
$$

$$
= \lim_{c \to \infty} \kappa(c) \sup_{z \in \mathcal{B}} \sup_{n} \sup_{t \in [0,T]} \left\| \sum_{i=m(T_n)}^{m(T_n+t)-1} \alpha(i) b(z, W_{i+1}) \right\|
$$

$$
= 0 \sup_{z \in \mathcal{B}} \sup_{n} \sup_{t \in [0,T]} \left\| \sum_{i=m(T_n)}^{m(T_n+t)-1} \alpha(i) b(z, W_{i+1}) \right\| \qquad (70)
$$

We now show that the function

$$
z \mapsto \sup_{n} \sup_{t \in [0,T]} \left\| \sum_{i=m(T_n)}^{m(T_n+t)-1} \alpha(i) b(z, W_{i+1}) \right\| \qquad (71)
$$

is Lipschitz continuous. $\forall z, z'$,

$$
\left| \sup_{n} \sup_{t \in [0,T]} \left\| \sum_{i=m(T_n)}^{m(T_n+t)-1} \alpha(i) b(z, W_{i+1}) \right\| - \sup_{n} \sup_{t \in [0,T]} \left\| \sum_{i=m(T_n)}^{m(T_n+t)-1} \alpha(i) b(z', W_{i+1}) \right\| \right|
$$

$$
\leq \sup_{n} \sup_{t \in [0,T]} \left| \left\| \sum_{i=m(T_n)}^{m(T_n+t)-1} \alpha(i) b(z, W_{i+1}) \right\| - \left\| \sum_{i=m(T_n)}^{m(T_n+t)-1} \alpha(i) b(z', W_{i+1}) \right\| \right|
$$
$$
\text{(by } |\sup_z f(z) - \sup_z g(z)| \leq \sup_z |f(z) - g(z)|)
$$

$$
\leq \sup_{n} \sup_{t \in [0,T]} \left\| \sum_{i=m(T_n)}^{m(T_n+t)-1} \alpha(i) b(z, W_{i+1}) - \sum_{i=m(T_n)}^{m(T_n+t)-1} \alpha(i) b(z', W_{i+1}) \right\|
$$

$$
\leq \sup_{n} \sup_{t \in [0,T]} \sum_{i=m(T_n)}^{m(T_n+t)-1} \alpha(i) \| b(z, W_{i+1}) - b(z', W_{i+1}) \|
$$

$$
\leq \sup_{n} \sup_{t \in [0,T]} \left( \sum_{i=m(T_n)}^{m(T_n+t)-1} \alpha(i) L_b(W_{i+1}) \right) \| z - z' \|
$$

By Lemma 16.1 and (102),

$$
\sup_{n} \sup_{t \in [0,T]} \left( \sum_{i=m(T_n)}^{m(T_n+t)-1} \alpha(i) L_b(W_{i+1}) \right) < \infty
$$

can be viewed as the Lipschitz constant. Thus, (71) is a continuous function. Since $\mathcal{B}$ is compact, the extreme value theorems asserts that the supremum of (71) in $\mathcal{B}$ is attainable at some $z_\mathcal{B}$ and is finite. This means the RHS of (70) is 0, so

$$
\limsup_{c \to \infty} \sup_{z \in \mathcal{B}} \sup_{n} \sup_{t \in [0,T]} \left\| \sum_{i=m(T_n)}^{m(T_n+t)-1} \alpha(i) \left[ H_c(z, W_{i+1}) - H_\infty(z, W_{i+1}) \right] \right\| = 0.
$$

The proofs for the other statements follow similar arguments and so are omitted. $\qquad \square$

### 16.3 Definitions and Proofs Relating to Equicontinuity

**Definition 15.** *A sequence of functions $\left\{ \gamma_n : [0,T) \to \mathbb{R}^K \right\}$ is equicontinuous on $[0,T)$ if* $\sup_n \|\gamma_n(0)\| < \infty$ *and* $\forall \epsilon > 0$, $\exists \delta > 0$ *such that*

$$\sup_n \sup_{0 \leq |t_1 - t_2| \leq \delta, \, 0 \leq t_1 \leq t_2 < T} \|\gamma_n(t_1) - \gamma_n(t_2)\| \leq \epsilon.$$

A standard example of a family of equicontinuous functions is a sequence of bounded Lipschitz continuous functions with a common Lipschitz constant. Clearly, if $\{\gamma_n\}$ is equicontinuous, each $\gamma_n$ must be continuous. However, the functions of interest in this work, i.e., $\tilde{z}_n(t)$, $f_n(t)$, are not continuous, so equicontinuity cannot apply. We, therefore, define the following equicontinuity in the extended sense.

**Definition 16.** *A sequence of functions $\left\{ \gamma_n : [0,T) \to \mathbb{R}^K \right\}$ is equicontinuous in the extended sense on $[0,T)$ if* $\sup_n \|\gamma_n(0)\| < \infty$ *and* $\forall \epsilon > 0$, $\exists \delta > 0$ *such that*

$$\limsup_n \sup_{0 \leq |t_1 - t_2| \leq \delta, \, 0 \leq t_1 \leq t_2 < T} \|\gamma_n(t_1) - \gamma_n(t_2)\| \leq \epsilon.$$

The following lemmas establish the desired equicontinuity, where Lemma 16.1 plays a key role.

**Lemma 16.3.** $\{\tilde{z}_n(t)\}_{n=0}^{\infty}$ *is equicontinuous in the extended sense on* $[0, T+1)$.

*Proof.* By (19),

$$\sup_n \|\tilde{z}_n(0)\| \leq 1.$$

Without loss of generality, let $t_1 \leq t_2$.

$$\limsup_n \sup_{0 \leq t_2 - t_1 \leq \delta} \|\tilde{z}_n(t_1) - \tilde{z}_n(t_2)\|$$

$$= \limsup_n \sup_{0 \leq t_2 - t_1 \leq \delta} \left\| \sum_{i=m(T_n+t_1)}^{m(T_n+t_2)-1} (\alpha(i) H_{r_n}(\tilde{z}_n(t(i) - T_n), W_{i+1}), \beta(i) G_{r_n}(\tilde{z}_n(t(i) - T_n), W_{i+1})) \right\|$$

$$\leq \limsup_n \sup_{0 \leq t_2 - t_1 \leq \delta} \left\| \sum_{i=m(T_n+t_1)}^{m(T_n+t_2)-1} \alpha(i) H_{r_n}(\tilde{z}_n(t(i) - T_n), W_{i+1}) \right\|$$

$$+ \limsup_n \sup_{0 \leq t_2 - t_1 \leq \delta} \left\| \sum_{i=m(T_n+t_1)}^{m(T_n+t_2)-1} \beta(i) G_{r_n}(\tilde{z}_n(t(i) - T_n), W_{i+1}) \right\|.$$

We bound each term individually. We start with the first term.

$\forall \xi > 0$, by (95), $\exists \delta_0$, such that $\forall 0 < \delta \leq \delta_0$,

$$\sup_{c \geq 1} \limsup_n \sup_{0 \leq t_2 - t_1 \leq \delta} \left\| \sum_{i=m(T_n+t_1)}^{m(T_n+t_2)-1} \alpha(i) H_c(0, W_{i+1}) \right\| \leq \xi. \tag{72}$$

By (101), $\exists \delta_1$, such that $\forall 0 < \delta \leq \delta_1$,

$$\limsup_n \sup_{0 \leq t_2 - t_1 \leq \delta} \sum_{i=m(T_n+t_1)}^{m(T_n+t_2)-1} \alpha(i) L(W_{i+1}) \leq \xi. \tag{73}$$

Without loss of generality, let $t_1 \leq t_2$. Then $\forall \delta \leq \min\{\delta_0, \delta_1\}$, we have

$$\limsup_n \sup_{0 \leq t_2 - t_1 \leq \delta} \left\| \sum_{i=m(T_n+t_1)}^{m(T_n+t_2)-1} \alpha(i) H_{r_n}(\tilde{z}_n(t(i) - T_n), W_{i+1}) \right\|$$

$$\leq \limsup_n \sup_{0 \leq t_2 - t_1 \leq \delta} \left\| \left\| \sum_{i=m(T_n+t_1)}^{m(T_n+t_2)-1} \alpha(i) H_{r_n}(\tilde{z}_n(t(i) - T_n), W_{i+1}) \right\| - \left\| \sum_{i=m(T_n+t_1)}^{m(T_n+t_2)-1} \alpha(i) H_{r_n}(0, W_{i+1}) \right\| \right\|$$

$$+ \limsup_n \sup_{0 \leq t_2 - t_1 \leq \delta} \left\| \sum_{i=m(T_n+t_1)}^{m(T_n+t_2)-1} \alpha(i) H_{r_n}(0, W_{i+1}) \right\|$$

$$\leq \limsup_n \sup_{0 \leq t_2 - t_1 \leq \delta} \left\| \left\| \sum_{i=m(T_n+t_1)}^{m(T_n+t_2)-1} \alpha(i) H_{r_n}(\tilde{z}_n(t(i) - T_n), W_{i+1}) \right\| - \left\| \sum_{i=m(T_n+t_1)}^{m(T_n+t_2)-1} \alpha(i) H_{r_n}(0, W_{i+1}) \right\| \right\|$$

$$+ \sup_{c \geq 1} \limsup_n \sup_{0 \leq t_2 - t_1 \leq \delta} \left\| \sum_{i=m(T_n+t_1)}^{m(T_n+t_2)-1} \alpha(i) H_c(0, W_{i+1}) \right\|$$

$$\leq \limsup_n \sup_{0 \leq t_2 - t_1 \leq \delta} \left\| \left\| \sum_{i=m(T_n+t_1)}^{m(T_n+t_2)-1} \alpha(i) H_{r_n}(\tilde{z}_n(t(i) - T_n), W_{i+1}) \right\| - \left\| \sum_{i=m(T_n+t_1)}^{m(T_n+t_2)-1} \alpha(i) H_{r_n}(0, W_{i+1}) \right\| \right\|$$

$$+ \xi \qquad\qquad\qquad\qquad\qquad\qquad\qquad\qquad\qquad\qquad\qquad\qquad \text{(by (72))}$$

$$\leq \limsup_n \sup_{0 \leq t_2 - t_1 \leq \delta} \left\| \sum_{i=m(T_n+t_1)}^{m(T_n+t_2)-1} \alpha(i) H_{r_n}(\tilde{z}_n(t(i) - T_n), W_{i+1}) - \sum_{i=m(T_n+t_1)}^{m(T_n+t_2)-1} \alpha(i) H_{r_n}(0, W_{i+1}) \right\|$$

$$+ \xi$$

$$\leq \limsup_n \sup_{0 \leq t_2 - t_1 \leq \delta} \sum_{i=m(T_n+t_1)}^{m(T_n+t_2)-1} \alpha(i) \| H_{r_n}(\tilde{z}_n(t(i) - T_n), W_{i+1}) - H_{r_n}(0, W_{i+1}) \| + \xi$$

$$\leq \limsup_n \sup_{0 \leq t_2 - t_1 \leq \delta} \sum_{i=m(T_n+t_1)}^{m(T_n+t_2)-1} \alpha(i) L(W_{i+1}) \| \tilde{z}_n(t(i) - T_n) \| + \xi$$

$$\leq (C_{\hat{x}} + C_{\hat{y}}) \limsup_n \sup_{0 \leq t_2 - t_1 \leq \delta} \sum_{i=m(T_n+t_1)}^{m(T_n+t_2)-1} \alpha(i) L(W_{i+1}) + \xi \qquad\qquad\quad \text{(by Lemma 17.6)}$$

$$\leq (C_{\hat{x}} + C_{\hat{y}}) \xi + \xi. \qquad\qquad\qquad\qquad\qquad\qquad\qquad\qquad\qquad\qquad\qquad \text{(by (73))}$$

A similar argument bounds the other term, implying that $\{\tilde{z}_n(t)\}$ is equicontinuous in the extended sense. $\qquad\square$

**Lemma 16.4.** $\{z_n(t)\}$ *is equicontinuous on* $[0, T+1)$.

*Proof.* By (19) and (20),

$$\sup_n \| z_n(0) \| \leq 1.$$

Without loss of generality, let $t_1 \le t_2$. Then $\forall \delta > 0$, we have

$$\sup_n \sup_{0 \le |t_1 - t_2| \le \delta,\, 0 \le t_1 \le t_2 < T} \|z_n(t_1) - z_n(t_2)\|$$

$$= \sup_n \sup_{0 \le |t_1 - t_2| \le \delta,\, 0 \le t_1 \le t_2 < T} \left\| \int_{t_1}^{t_2} h_{r_n}(z_n(s)) ds \right\|$$

$$= \sup_n \sup_{0 \le |t_1 - t_2| \le \delta,\, 0 \le t_1 \le t_2 < T} \left\| \int_{t_1}^{t_2} \left[ h_{r_n}(z_n(s)) - h_{r_n}(0) \right] ds + \int_{t_1}^{t_2} h_{r_n}(0) ds \right\|$$

$$\le \sup_n \sup_{0 \le |t_1 - t_2| \le \delta,\, 0 \le t_1 \le t_2 < T} \int_{t_1}^{t_2} \|h_{r_n}(z_n(s)) - h_{r_n}(0)\| ds + \sup_n \sup_{0 \le |t_1 - t_2| \le \delta,\, 0 \le t_1 \le t_2 < T} \int_{t_1}^{t_2} \|h_{r_n}(0)\| ds$$

$$\le \sup_n \sup_{0 \le |t_1 - t_2| \le \delta,\, 0 \le t_1 \le t_2 < T} \int_{t_1}^{t_2} L \|z_n(s)\| ds + \sup_n \sup_{0 \le |t_1 - t_2| \le \delta,\, 0 \le t_1 \le t_2 < T} \int_{t_1}^{t_2} \|h_{r_n}(0)\| ds$$
$$\text{(by Lemma 17.2)}$$

$$\le \delta L C_{\hat{x}} + \sup_n \sup_{0 \le |t_1 - t_2| \le \delta,\, 0 \le t_1 \le t_2 < T} \int_{t_1}^{t_2} \|h_{r_n}(0)\| ds \qquad \text{(by Lemma 17.7)}$$

$$\le \delta (L C_{\hat{x}} + C_H), \qquad\qquad\qquad\qquad \text{(by (104))}$$

which implies that $\{z_n\}$ is equicontinuous. $\qquad\square$

**Lemma 16.5.** $\{f_n(t)\}$ *is equicontinuous in the extended sense on* $[0, T+1)$.

*Proof.*

$$\sup_n f_n(0) = \sup_n \tilde{z}_n(0) - z_n(0) = \sup_n \tilde{z}_n(0) - \tilde{z}_n(0) = 0 < \infty.$$

By Lemma 16.3 and Lemma 16.4, $\forall \epsilon > 0$, $\exists \delta$ such that

$$\limsup_n \sup_{0 \le t_2 - t_1 \le \delta} \|\tilde{z}_n(t_1) - \tilde{z}_n(t_2)\| \le \frac{\epsilon}{2},$$

$$\sup_n \sup_{0 \le t_2 - t_1 \le \delta} \|z_n(t_1) - z_n(t_2)\| \le \frac{\epsilon}{2}.$$

Without loss of generality let $t_1 \le t_2$. Then $\forall \epsilon$, $\exists \delta$ such that

$$\limsup_n \sup_{0 \le t_2 - t_1 \le \delta} \|f_n(t_1) - f_n(t_2)\|$$

$$= \limsup_n \sup_{0 \le t_2 - t_1 \le \delta} \|\tilde{z}_n(t_1) - \tilde{z}_n(t_2) - (z_n(t_1) - z_n(t_2))\|$$

$$\le \limsup_n \sup_{0 \le t_2 - t_1 \le \delta} \|\tilde{z}_n(t_1) - \tilde{z}_n(t_2)\| + \limsup_n \sup_{0 \le t_2 - t_1 \le \delta} \|z_n(t_1) - z_n(t_2)\|$$

$$\le \limsup_n \sup_{0 \le t_2 - t_1 \le \delta} \|\tilde{z}_n(t_1) - \tilde{z}_n(t_2)\| + \sup_n \sup_{0 \le t_2 - t_1 \le \delta} \|z_n(t_1) - z_n(t_2)\|$$

$$\le \epsilon,$$

which implies that $\{f_n\}$ is equicontinuous in the extended sense. $\qquad\square$

### 16.4 PROOF OF LEMMA 5.1

*Proof.* Since $\sup_n r_n = \infty$ and $r_n$ is monotonic, $\lim_{n \to \infty} r_n = \infty$, and every subsequence also converges to infinity.

Since $\{f_{n_{k,0}}\}$ is equicontinuous, by the Arzelà-Ascoli theorem (see Appendix 10.4), there exists a subsequence $n_{k,1} \subseteq n_{k,0}$ such that $\{f_{n_{k,1}}\}$ converges uniformly to a continuous limit $f^{\lim}$. Similarly, since $\{\tilde{z}_{n_{k,1}}(t)\}$ is equicontinuous, there is a subsequence $n_k \subseteq n_{k,1}$ such that $\{\tilde{z}_{n_k}(t)\}$ converges uniformly in $t$ to a continuous limit $\tilde{z}^{\lim}(t)$.

The proof that $\lim_{n \to \infty} z_{n_k}(t) = z^{\lim}(t)$ uniformly is by lemma 17.11. $\qquad\square$

### 16.5   PROOF OF LEMMA 12.2

*Proof.*

$$
\lim_{k\to\infty} \left\| \sum_{i=m(T_{n_k})}^{m(T_{n_k}+t)-1} \beta(i) G_{r_{n_k}}(\tilde{z}_{n_k}(t(i)-T_{n_k}), W_{i+1}) \right\|
$$

$$
\leq \lim_{k\to\infty} \sum_{i=m(T_{n_k})}^{m(T_{n_k}+t)-1} \beta(i) \left\| G_{r_{n_k}}(\tilde{z}_{n_k}(t(i)-T_{n_k}), W_{i+1}) \right\|
$$

$$
\leq \lim_{k\to\infty} \sum_{i=m(T_{n_k})}^{m(T_{n_k}+t)-1} \beta(i) L(W_{i+1}) \| \tilde{z}_{n_k}(t(i)-T_{n_k}) \|
$$

$$
\leq C_{\hat{z}} \lim_{k\to\infty} \sum_{i=m(T_{n_k})}^{m(T_{n_k}+t)-1} \beta(i) L(W_{i+1}) \qquad\qquad \text{(by Lemma 17.6)}
$$

$$
\leq 0. \tag{100}
$$

$\square$

### 16.6   BOUNDING THE DISCRETIZATION ERROR

We will now prove that the first term in the RHS of (24) is 0. We need to show that $\forall t \in [0, T+1)$,

$$
\lim_{k\to\infty} \left\| \sum_{i=m(T_{n_k})}^{m(T_{n_k}+t)-1} \alpha(i) H_{r_{n_k}}(\tilde{z}_{n_k}(t(i)-T_{n_k}), W_{i+1}) - \int_0^t h_{r_{n_k}}(\tilde{z}^{\lim}(s)) ds \right\| = 0. \tag{74}
$$

To evaluate the above, we first fix any $t \in [0, T+1)$ and then compute the following stronger double limit, which implies that the above limit holds.

$$
\lim_{\substack{j\to\infty \\ k\to\infty}} \left\| \sum_{i=m(T_{n_k})}^{m(T_{n_k}+t)-1} \alpha(i) H_{r_{n_j}}(\tilde{z}_{n_k}(t(i)-T_{n_k}), W_{i+1}) - \int_0^t h_{r_{n_j}}(\tilde{z}^{\lim}(s)) ds \right\|. \tag{75}
$$

The Moore-Osgood theorem (Appendix 10.5) will help us compute this double limit by turning it into iterated limits. To invoke the Moore-Osgood theorem, we first prove the uniform convergence in $k$ when $j \to \infty$.

**Lemma 16.6.** $\forall t \in [0, T+1)$,

$$
\lim_{j\to\infty} \left\| \sum_{i=m(T_{n_k})}^{m(T_{n_k}+t)-1} \alpha(i) H_{r_{n_j}}(\tilde{z}_{n_k}(t(i)-T_{n_k}), W_{i+1}) - \int_0^t h_{r_{n_k}}(\tilde{z}^{\lim}(s)) ds \right\|
$$

$$
= \left\| \sum_{i=m(T_{n_k})}^{m(T_{n_k}+t)-1} \alpha(i) H_{\infty}(\tilde{z}_{n_k}(t(i)-T_{n_k}), W_{i+1}) - \int_0^t h_{\infty}(\tilde{z}^{\lim}(s)) ds \right\|
$$

*uniformly in $k$.*

*Proof.* $\forall j, \forall k, \forall t \in [0, T+1)$,

$$\left\| \sum_{i=m(T_{n_k})}^{m(T_{n_k}+t)-1} \alpha(i) H_{r_{n_j}}(\tilde{z}_{n_k}(t(i) - T_{n_k}), W_{i+1}) - \int_0^t h_{r_{n_j}}(\tilde{z}^{\lim}(s)) ds \right\|$$

$$- \left\| \sum_{i=m(T_{n_k})}^{m(T_{n_k}+t)-1} \alpha(i) H_\infty(\tilde{z}_{n_k}(t(i) - T_{n_k}), W_{i+1}) - \int_0^t h_\infty(\tilde{z}^{\lim}(s)) ds \right\|$$

$$\leq \left\| \sum_{i=m(T_{n_k})}^{m(T_{n_k}+t)-1} \alpha(i) H_{r_{n_j}}(\tilde{z}_{n_k}(t(i) - T_{n_k}), W_{i+1}) - \int_0^t h_{r_{n_j}}(\tilde{z}^{\lim}(s)) ds \right.$$

$$\left. - \sum_{i=m(T_{n_k})}^{m(T_{n_k}+t)-1} \alpha(i) H_\infty(\tilde{z}_{n_k}(t(i)), W_{i+1}) + \int_0^t h_\infty(\tilde{z}^{\lim}(s)) ds \right\| \quad \text{(by } |\|a\| - \|b\|| \leq \|a - b\|)$$

$$\leq \left\| \sum_{i=m(T_{n_k})}^{m(T_{n_k}+t)-1} \alpha(i)(H_{r_{n_j}}(\tilde{z}_{n_k}(t(i) - T_{n_k}), W_{i+1}) - H_\infty(\tilde{z}_{n_k}(t(i) - T_{n_k}), W_{i+1})) \right\|$$

$$+ \left\| \int_0^t h_{r_{n_j}}(\tilde{z}^{\lim}(s)) - h_\infty(\tilde{z}^{\lim}(s)) ds \right\|$$

$$\leq \left\| \sum_{i=m(T_{n_k})}^{m(T_{n_k}+t)-1} \alpha(i)(H_{r_{n_j}}(\tilde{z}_{n_k}(t(i) - T_{n_k}), W_{i+1}) - H_\infty(\tilde{z}_{n_k}(t(i) - T_{n_k}), W_{i+1})) \right\|$$

$$+ \int_0^t \left\| h_{r_{n_j}}(\tilde{z}^{\lim}(s)) - h_\infty(\tilde{z}^{\lim}(s)) \right\| ds \tag{76}$$

By Lemma 17.6, $\tilde{z}_{n_k}(t(i) - T_{n_k})$ is in a compact set $\mathcal{B}_{\hat{z}}$. By Lemma 16.2, for the compact set $\mathcal{B}_{\hat{z}}$, $\forall \epsilon > 0, \exists j_1$ such that $\forall j \geq j_1, \forall k, \forall z \in \mathcal{B}, \forall t \in [0, T+1)$,

$$\left\| \sum_{i=m(T_{n_k})}^{m(T_{n_k}+t)-1} \alpha(i) \left[ H_{r_{n_j}}(z, W_{i+1}) - H_\infty(z, W_{i+1}) \right] \right\| \leq \epsilon. \tag{77}$$

Similar to the proof of Lemma 17.10, we have

$$\lim_{j \to \infty} h_{r_{n_j}}(\tilde{z}_k(t)) = h_\infty(\tilde{z}_k(t)) \tag{78}$$

uniformly in $k$ and $t \in [0, T+1)$. By (78), $\forall \epsilon > 0, \exists j_2$ such that $\forall j > j_2, \forall k, \forall t \in [0, T+1)$,

$$\left\| h_{r_{n_j}}(\tilde{z}_k(t)) - h_\infty(\tilde{z}_k(t)) \right\| \leq \epsilon. \tag{79}$$

Define $j_0 \doteq \max\{j_1, j_2\}$. $\forall j \geq j_0$, $\forall k$, $\forall t \in [0, T+1)$,

$$
\left\| \left\| \sum_{i=m(T_{n_k})}^{m(T_{n_k}+t)-1} \alpha(i) H_{r_{n_j}}(\tilde{z}_{n_k}(t(i) - T_{n_k}), W_{i+1}) - \int_0^t h_{r_{n_j}}(\tilde{z}^{\lim}(s))ds \right\| \right.
$$

$$
\left. - \left\| \sum_{i=m(T_{n_k})}^{m(T_{n_k}+t)-1} \alpha(i) H_\infty(\tilde{z}_{n_k}(t(i) - T_{n_k}), W_{i+1}) - \int_0^t h_\infty(\tilde{z}^{\lim}(s))ds \right\| \right\|
$$

$$
\leq \left\| \sum_{i=m(T_{n_k})}^{m(T_{n_k}+t)-1} \alpha(i)(H_{r_{n_j}}(\tilde{z}_{n_k}(t(i) - T_{n_k}), W_{i+1}) - H_\infty(\tilde{z}_{n_k}(t(i) - T_{n_k}), W_{i+1})) \right\| + (T+1)\epsilon
$$

(by (76), (79))

$$
\leq \epsilon + (T+1)\epsilon \qquad\qquad\qquad\qquad\qquad\qquad\qquad\qquad \text{(by (76), (77))}
$$
$$
\leq (T+2)\epsilon.
$$

This completes the proof of uniform convergence. $\qquad\square$

Now, we prove, for each $j$, the convergence with $k \to \infty$.

**Lemma 16.7.** $\forall t \in [0, T+1)$, $\forall j$,

$$
\lim_{k \to \infty} \left\| \sum_{i=m(T_{n_k})}^{m(T_{n_k}+t)-1} \alpha(i) H_{r_{n_j}}(\tilde{z}_{n_j}(t(i) - T_{n_j}), W_{i+1}) - \int_0^t h_{r_{n_j}}(\tilde{z}^{\lim}(s))ds \right\| = 0.
$$

The proof of Lemma 16.7 is very similar to the proof of Lemma 18 in Liu et al. (2025) and is omitted here due to its length.

We are now ready to compute the limit in (74).

**Lemma 16.8.** $\forall t \in [0, T+1)$,

$$
\lim_{k \to \infty} \left\| \sum_{i=m(T_{n_k})}^{m(T_{n_k}+t)-1} \alpha(i) H_{r_{n_k}}(\tilde{z}_{n_k}(t(i) - T_{n_k}), W_{i+1}) - \int_0^t h_{r_{n_k}}(\tilde{z}^{\lim}(s))ds \right\| = 0.
$$

*Proof.* It follows immediately from Lemmas 16.6 & 16.7, the Moore-Osgood theorem, and Lemma 17.12. $\qquad\square$

Lemma 16.8 confirms that the first term in the RHS of (24) is 0. Moreover, it also enables us to rewrite $\tilde{z}^{\lim}(t)$ from a summation form to an integral form.

$$
\tilde{z}^{\lim}(t)
$$
$$
= \lim_{k \to \infty} \tilde{z}_{n_k}(0) + \sum_{i=m(T_{n_k})}^{m(T_{n_k}+t)-1} \alpha(i) H_{r_{n_k}}(\tilde{z}_{n_k}(t(i) - T_{n_k}), W_{i+1})
$$
$$
= \lim_{k \to \infty} \tilde{z}_{n_k}(0) + \int_0^t h_{r_{n_k}}(\tilde{z}^{\lim}(s))ds. \qquad\qquad \text{(by Lemma 16.8)} \quad (80)
$$

This, together with a few Gronwall's inequality arguments, confirms that the discretization error indeed diminishes along $\{n_k\}$.

**Lemma 16.9.** $\forall t \in [0, T+1)$,

$$
\lim_{k \to \infty} \|f_{n_k}(t)\| = 0.
$$

*Proof.* $\forall t \in [0, T+1)$,

$$\lim_{k \to \infty} \|f_{n_k}(t)\|$$

$$\leq \lim_{k \to \infty} \left\| \sum_{i=m(T_{n_k})}^{m(T_{n_k}+t)-1} \alpha(i) H_{r_{n_k}}(\tilde{z}_{n_k}(t(i) - T_{n_k}), W_{i+1}) - \int_0^t h_{r_{n_k}}(\tilde{z}^{\lim}(s))ds \right\|$$

$$+ \lim_{k \to \infty} \left\| \int_0^t h_{r_{n_k}}(\tilde{z}^{\lim}(s))ds - \int_0^t h_{r_{n_k}}(z_{n_k}(s))ds \right\| + 0 \qquad \text{(by (24) and Lemma 12.2)}$$

$$= \lim_{k \to \infty} \left\| \int_0^t h_{r_{n_k}}(\tilde{z}^{\lim}(s))ds - \int_0^t h_{r_{n_k}}(z_{n_k}(s))ds \right\|$$

$$= \left\| \int_0^t h_\infty(\tilde{z}^{\lim}(s))ds - \int_0^t h_\infty(z^{\lim}(s))ds \right\|. \quad \text{(by Lemma 17.13 and Lemma 17.14)} \quad (81)$$

We now show the relationship between $\tilde{z}^{\lim}(t)$ and $z^{\lim}(t)$.

$$\left\| \tilde{z}^{\lim}(t) - z^{\lim}(t) \right\| \tag{82}$$

$$= \left\| \lim_{k \to \infty} \left[ \tilde{z}_{n_k}(0) + \int_0^t h_{r_{n_k}}(\tilde{z}^{\lim}(s))ds \right] - \left[ \tilde{z}^{\lim}(0) + \int_0^t h_\infty(z^{\lim}(s))ds \right] \right\| \quad \text{(by (23) and (80))}$$

$$= \left\| \tilde{z}^{\lim}(0) + \int_0^t h_\infty(\tilde{z}^{\lim}(s))ds - \left[ \tilde{z}^{\lim}(0) + \int_0^t h_\infty(z^{\lim}(s))ds \right] \right\| \qquad \text{(by Lemma 17.13)}$$

$$= \left\| \int_0^t h_\infty(\tilde{z}^{\lim}(s))ds - \int_0^t h_\infty(z^{\lim}(s))ds \right\| \tag{83}$$

$$\leq \int_0^t L \left\| \tilde{z}^{\lim}(s) - z^{\lim}(s) \right\| ds \qquad \text{(by Lemma 17.2)}$$

$$\leq 0. \qquad \text{(by Gronwall inequality in Theorem 10.1)}$$

Thus,

$$\left\| \lim_{k \to \infty} f_{n_k}(t) \right\|$$

$$\leq \left\| \int_0^t h_\infty(\tilde{z}^{\lim}(s))ds - \int_0^t h_\infty(z^{\lim}(s))ds \right\| \qquad \text{(by (81))}$$

$$= \left\| \tilde{z}^{\lim}(t) - z^{\lim}(t) \right\| \qquad \text{(by (83))}$$

$$\leq 0. \qquad \text{(by (82))}$$

$\square$

### 16.7 PROOF OF LEMMA 5.2

*The proof is similar to Lemma 1 from Chapter 3.2 of Borkar (2009).*

*Proof.* By Lyapunov stability, there is a $\delta > 0$ such that any trajectory beginning within $B(\lambda_\infty(y), \delta)$ stays within $\frac{\epsilon}{2}$ of the equilibrium $\lambda_\infty(y)$.

For an initial condition $x$, let $T_x$ be a time at which the trajectory is within $\frac{\delta}{2}$ of the equilibrium. Let $x_1$ be some other initial condition. By definition, Lipschitzness, and the Gronwall inequality, we have

$$\|\eta_\infty^y(t, x) - \eta_\infty^y(t, x_1)\| \leq \|x - x_1\| + L \int_0^t \|\eta_\infty^y(s, x) - \eta_\infty^y(s, x_1)\| ds$$

$$\leq \|x - x_1\| e^{LT_x}$$

for all $t \leq T_x$.

So there is a neighborhood $V_x$ such that for all $x_1 \in V_x$, $\eta_\infty^y(T_x, x_1)$ is within $\delta$ of the equilibrium, which by Lyapunov stability implies that it will always be within $\epsilon$ of the equilibrium after $T_x$.

By compactness, we can cover the set $K$ by a finite number of such intervals and obtain a finite number of times $T_{x_1}, \ldots, T_{x_n}$ and then take the maximum time to be the value of $T_\epsilon$. $\qquad\square$

### 16.8 PROOF OF LEMMA 5.3

*The proof is similar to Lemma 2 from chapter 3.2 of Borkar (2009).*

*Proof.* We have

$$\eta_c^{y'(t)}(t,x) = x + \int_0^t h_c(\eta_c^{y'(s)}(s,x), y'(s))ds,$$

$$\eta_\infty^y(t,x) = x + \int_0^t h_\infty(\eta_\infty^y(s,x), y)ds.$$

Let us define the error term:

$$e(t) = \left\| \eta_c^{y'(t)}(t,x) - \eta_\infty^y(t,x) \right\|.$$

We can bound $e(t)$ by two terms:

$$e(t) \leq \int_0^t \left\| h_c(\eta_c^{y'(s)}(s,x), y'(s)) - h_c(\eta_\infty^y(s,x), y'(s)) \right\| ds$$

$$+ \int_0^t \left\| h_c(\eta_\infty^y(s,x), y'(s)) - h_\infty(\eta_\infty^y(s,x), y) \right\| ds$$

To bound the first term, we use Lipschitzness:

$$\int_0^t \left\| h_c(\eta_c^{y'(s)}(s,x), y'(s)) - h_c(\eta_\infty^y(s,x), y'(s)) \right\| ds \leq L \int_0^t \left\| \eta_c^{y'(t)}(t,x) - \eta_\infty^y(t,x) \right\| ds$$

$$= L \int_0^t e(s)ds.$$

To bound the second term, we use Lipschitzness and Assumption 3:

$$\int_0^t \| h_c(\eta_\infty^y(s,x), y'(s)) - h_\infty(\eta_\infty^y(s,x), y) \| ds \leq \int_0^t \| h_c(\eta_\infty^y(s,x), y'(s)) - h_\infty(\eta_\infty^y(s,x), y'(s)) \| ds$$

$$+ \int_0^t \| h_\infty(\eta_\infty^y(s,x), y'(s)) - h_\infty(\eta_\infty^y(s,x), y) \| ds$$

$$\leq \int_0^t \epsilon(c)ds + L \int_0^t \| y'(s) - y \| ds$$

$$\leq T\epsilon(c) + TL\rho.$$

To conclude, we will use the Gronwall Inequality (Appendix 10.1):

$$e(t) \leq T\epsilon(c) + TL\rho + L \int_0^t e(s)ds$$

$$\leq (L\rho + \epsilon(c))Te^{LT}.$$

$\qquad\square$

## 17 AUXILIARY LEMMAS

**Lemma 17.1.**

$$\forall n, \ T_{n+1} - T_n \geq T,$$
$$\lim_{n \to \infty} T_{n+1} - T_n = T.$$

*Moreover, $\forall \tau > 0, t_1, t_2$ such that $-\tau \leq t_1 \leq t_2 \leq \tau$, we have*

$$\lim_{n \to \infty} \sum_{i=m(t(n)+t_1)}^{m(t(n)+t_2)-1} \alpha(i) = t_2 - t_1, \tag{84}$$

$$\lim_{n \to \infty} \sum_{i=m(t(n)+t_1)}^{m(t(n)+t_2)-1} \beta(i) = 0. \tag{85}$$

*Proof.* $\forall n,$

$$T_{n+1} - T_n$$
$$= t(m(T_n + T) + 1) - T_n$$
$$\geq T_n + T - T_n$$
$$\geq T.$$

Thus,

$$\lim_{n \to \infty} T_{n+1} - T_n \geq T.$$

With

$$\lim_{n \to \infty} T_{n+1} - T_n$$
$$= \lim_{n \to \infty} t(m(T_n + T) + 1) - T_n$$
$$= \lim_{n \to \infty} t(m(T_n + T)) + \alpha(m(T_n + T)) - T_n$$
$$\leq \lim_{n \to \infty} T_n + T + \alpha(m(T_n + T)) - T_n$$
$$= T,$$

by the squeeze theorem, we have $\lim_{n \to \infty} T_{n+1} - T_n = T$.

To prove (84), $\forall \tau, \forall -\tau \leq t_1 \leq t_2 \leq \tau$, it suffices to only consider large $n$ such that $t(n) - \tau \geq 0$. We have

$$\lim_{n \to \infty} \sum_{i=m(t(n)+t_1)}^{m(t(n)+t_2)-1} \alpha(i)$$
$$= \lim_{n \to \infty} t(m(t(n) + t_2)) - t(m(t(n) + t_1))$$
$$\leq \lim_{n \to \infty} t(n) + t_2 - t(m(t(n) + t_1))$$
$$\leq \lim_{n \to \infty} t(n) + t_2 - (t(n) + t_1 - \alpha(m(t(n) + t_1)))$$
$$= t_2 - t_1 + \lim_{n \to \infty} \alpha(m(t(n) + t_1))$$
$$= t_2 - t_1 \tag{by (7)}$$

and

$$\lim_{n\to\infty} \sum_{i=m(t(n)+t_1)}^{m(t(n)+t_2)-1} \alpha(i)$$

$$= \lim_{n\to\infty} t(m(t(n)+t_2)) - t(m(t(n)+t_1))$$

$$\geq \lim_{n\to\infty} t(n) + t_2 - \alpha(m(t(n)+t_2)) - t(m(t(n)+t_1))$$

$$\geq \lim_{n\to\infty} t(n) + t_2 - \alpha(m(t(n)+t_2)) - (t(n)+t_1)$$

$$= \lim_{n\to\infty} t_2 - t_1 - \alpha(m(t(n)+t_2))$$

$$= t_2 - t_1. \qquad\qquad \text{(by (7))}$$

By the squeeze theorem, we have

$$\lim_{n\to\infty} \sum_{i=m(t(n)+t_1)}^{m(t(n)+t_2)-1} \alpha(i) = t_2 - t_1.$$

To prove (85), fix $\epsilon > 0$. By (8), there exists $N_1 \in \mathbb{N}$ such that for all $n > m(t(N_1)+t_1)$,

$$\frac{\beta(n)}{\alpha(n)} < \epsilon. \tag{86}$$

By (84), there exists $N_2 \in \mathbb{N}$ such that for all $n > N_2$,

$$\sum_{i=m(t(n)+t_1)}^{m(t(n)+t_2)-1} \alpha(i) < t_2 - t_1 + \epsilon. \tag{87}$$

Let $N = \max\{N_1, N_2\}$. Then for all $n > N$,

$$\sum_{i=m(t(n)+t_1)}^{m(t(n)+t_2)-1} \beta(i)$$

$$< \epsilon \sum_{i=m(t(n)+t_1)}^{m(t(n)+t_2)-1} \alpha(i) \qquad\qquad \text{(by (86))}$$

$$< \epsilon(t_2 - t_1 + \epsilon). \qquad\qquad \text{(by (87))}$$

Since $t_2 - t_1$ is a constant and $\epsilon$ can be made arbitrarily small, we have

$$\lim_{n\to\infty} \sum_{i=m(t(n)+t_1)}^{m(t(n)+t_2)-1} \beta(i) = 0.$$

$$\qquad\qquad\qquad\qquad\qquad\qquad\qquad\qquad\qquad\qquad\qquad\qquad\qquad \square$$

**Lemma 17.2.** *For any* $x, x', y, y', w, c \geq 1$*, including* $c = \infty$*,*

$$\|H_c(x,y,w) - H_c(x',y',w)\| \leq L(w)\|(x,y) - (x',y')\|, \tag{88}$$

$$\|h_c(x,y) - h_c(x',y')\| \leq L\|(x,y) - (x',y')\|. \tag{89}$$

$$\|G_c(x,y,w) - G_c(x',y',w)\| \leq L(w)\|(x,y) - (x',y')\|, \tag{90}$$

$$\|g_c(x,y) - g_c(x',y')\| \leq L\|(x,y) - (x',y')\|. \tag{91}$$

*Proof.* To prove (88), we first consider $1 \leq c < \infty$,

$$\begin{aligned}
&\|H_c(x, y, w) - H_c(x', y', w)\| \\
&= \left\|\frac{H(cx, cy, w)}{c} - \frac{H(cx', cy, w)}{c}\right\| \quad \text{(by (9))} \\
&\leq \frac{\|H(cx, cy, w) - H(cx', cy', w)\|}{c} \\
&\leq L(w)\frac{\|(cx, cy) - (cx', cy')\|}{c} \quad \text{(by (12))} \\
&= L(w)\|(x, y) - (x', y')\|.
\end{aligned}$$

By (13),

$$\|H_\infty(x, y, w) - H_\infty(x', y', w)\| \leq L(w)\|(x, y) - (x', y')\|.$$

To prove (89), $\forall x, x', y, y', \forall c \geq 1$ including $c = \infty$,

$$\begin{aligned}
&\|h_c(x, y) - h_c(x', y')\| \\
&= \|\mathbb{E}_{w \sim \omega}\left[H_c(x, y, w) - H_c(x', y', w)\right]\| \\
&\leq \mathbb{E}_{w \sim \omega}\left[\|H_c(x, y, w) - H_c(x', y', w)\|\right] \\
&\leq \mathbb{E}_{w \sim \omega}\left[L(w)\|(x, y) - (x', y')\|\right] \\
&\leq L\|(x, y) - (x', y')\|.
\end{aligned}$$

By similar arguments, (90) and (91) also follow. $\qquad\square$

**Lemma 17.3.** $\forall x, y,$

$$\sup_{c \geq 1}\|h_c(0)\| < \infty, \sup_{c \geq 1}\|g_c(0)\| < \infty, \qquad (92)$$

$$\sup_{c \geq 1}\limsup_{n}\sup_{0 \leq t_1 \leq t_2 \leq T_{n+1} - T_n}\left\|\sum_{i=m(T_n+t_1)}^{m(T_n+t_2)-1}\alpha(i)\left[H_c(x, y, W_{i+1}) - h_c(x, y)\right]\right\| = 0 \quad a.s., \quad (93)$$

$$\sup_{c \geq 1}\sup_{n}\sup_{0 \leq t_1 \leq t_2 \leq T_{n+1} - T_n}\left\|\sum_{i=m(T_n+t_1)}^{m(T_n+t_2)-1}\alpha(i)H_c(0, W_{i+1})\right\| < \infty \quad a.s., (94)$$

$$\lim_{\delta \to 0^+}\sup_{c \geq 1}\limsup_{n}\sup_{0 \leq t_2 - t_1 \leq \delta}\left\|\sum_{i=m(T_n+t_1)}^{m(T_n+t_2)-1}\alpha(i)H_c(0, W_{i+1})\right\| = 0 \quad a.s. \quad (95)$$

*By the same arguments, we can show the last three results with the substitutions of $G$ for $H$ and $\beta(i)$ for $\alpha(i)$.*

*Proof.* **Proof of (92):**

$$\sup_{c \geq 1}\|h_c(0)\| = \sup_{c \geq 1}\left\|\frac{h(0)}{c}\right\| \leq \sup_{c \geq 1}\|h(0)\| = \|h(0)\| < \infty, \qquad (96)$$

$$\sup_{c \geq 1}\|g_c(0)\| = \sup_{c \geq 1}\left\|\frac{g(0)}{c}\right\| \leq \sup_{c \geq 1}\|g(0)\| = \|g(0)\| < \infty.$$

**Proof of (93):** $\forall x,$

$$\sup_{c \geq 1} \limsup_{n} \sup_{0 \leq t_1 \leq t_2 \leq T_{n+1}-T_n} \left\| \sum_{i=m(T_n+t_1)}^{m(T_n+t_2)-1} \alpha(i) \left[ H_c(x, y, W_{i+1}) - h_c(x, y) \right] \right\|$$

$$= \sup_{c \geq 1} \limsup_{n} \sup_{0 \leq t_1 \leq t_2 \leq T_{n+1}-T_n} \left\| \sum_{i=m(T_n+t_1)}^{m(T_n+t_2)-1} \alpha(i) \left[ \frac{H(cx, cy, W_{i+1})}{c} - \frac{h(cx, cy)}{c} \right] \right\|$$

$$= \sup_{c \geq 1} \frac{1}{c} \limsup_{n} \sup_{0 \leq t_1 \leq t_2 \leq T_{n+1}-T_n} \left\| \sum_{i=m(T_n+t_1)}^{m(T_n+t_2)-1} \alpha(i) \left[ H(cx, cy, W_{i+1}) - h(cx, cy) \right] \right\|$$

$$\leq \sup_{c \geq 1} \frac{1}{c} \limsup_{n} \sup_{0 \leq t_1 \leq t_2 \leq T+\sup_j \alpha(j)} \left\| \sum_{i=m(T_n+t_1)}^{m(T_n+t_2)-1} \alpha(i) \left[ H(cx, cy, W_{i+1}) - h(cx, cy) \right] \right\|$$

$$(\forall n, T_{n+1} - T_n \leq T + \sup_j \alpha(j))$$

$$= \sup_{c \geq 1} \frac{1}{c} \cdot 0$$

$$= 0. \tag{97}$$

**Proof of (94):**

$$\limsup_{n} \sup_{0 \leq t_1 \leq t_2 \leq T_{n+1}-T_n} \left\| \sum_{i=m(T_n+t_1)}^{m(T_n+t_2)-1} \alpha(i) H(0, W_{i+1}) \right\|$$

$$= \limsup_{n} \sup_{0 \leq t_1 \leq t_2 \leq T_{n+1}-T_n} \left\| \sum_{i=m(T_n+t_1)}^{m(T_n+t_2)-1} \alpha(i) [H(0, W_{i+1}) - h(0) + h(0)] \right\|$$

$$\leq \limsup_{n} \sup_{0 \leq t_1 \leq t_2 \leq T_{n+1}-T_n} \left\| \sum_{i=m(T_n+t_1)}^{m(T_n+t_2)-1} \alpha(i) [H(0, W_{i+1}) - h(0)] \right\|$$

$$+ \limsup_{n} \sup_{0 \leq t_1 \leq t_2 \leq T_{n+1}-T_n} \left\| \sum_{i=m(T_n+t_1)}^{m(T_n+t_2)-1} \alpha(i) h(0) \right\|$$

$$\leq \limsup_{n} \sup_{0 \leq t_1 \leq t_2 \leq T+\sup_j \alpha(j)} \left\| \sum_{i=m(T_n+t_1)}^{m(T_n+t_2)-1} \alpha(i) [H(0, W_{i+1}) - h(0)] \right\|$$

$$+ \limsup_{n} \sup_{0 \leq t_1 \leq t_2 \leq T+\sup_j \alpha(j)} \left\| \sum_{i=m(T_n+t_1)}^{m(T_n+t_2)-1} \alpha(i) h(0) \right\| \quad (\forall n, T_{n+1} - T_n \leq T + \sup_j \alpha(j))$$

$$= \limsup_{n} \sup_{0 \leq t_1 \leq t_2 \leq T+\sup_j \alpha(j)} \left\| \sum_{i=m(T_n+t_1)}^{m(T_n+t_2)-1} \alpha(i) h(0) \right\|$$

$$= \|h(0)\| \limsup_{n} \sup_{0 \leq t_1 \leq t_2 \leq T+\sup_j \alpha(j)} \sum_{i=m(T_n+t_1)}^{m(T_n+t_2)-1} \alpha(i)$$

$$= \|h(0)\| (T + \sup_j \alpha(j)) \qquad \text{(by Lemma 17.1)}$$

$$< \infty. \tag{98}$$

We now consider $c$ in the above bounds. We first get

$$\sup_{c \geq 1} \sup_{n} \sup_{0 \leq t_1 \leq t_2 \leq T_{n+1} - T_n} \left\| \sum_{i=m(T_n+t_1)}^{m(T_n+t_2)-1} \alpha(i) H_c(0, W_{i+1}) \right\|$$

$$= \sup_{c \geq 1} \sup_{n} \sup_{0 \leq t_1 \leq t_2 \leq T_{n+1} - T_n} \left\| \sum_{i=m(T_n+t_1)}^{m(T_n+t_2)-1} \alpha(i) \frac{H(0, W_{i+1})}{c} \right\| \qquad \text{(by (9))}$$

$$= \sup_{n} \sup_{0 \leq t_1 \leq t_2 \leq T_{n+1} - T_n} \left\| \sum_{i=m(T_n+t_1)}^{m(T_n+t_2)-1} \alpha(i) H(0, W_{i+1}) \right\| \qquad \text{(by } c \geq 1\text{)}$$

$$< \infty. \qquad \text{(by (98))}$$

**Proof of (95):**

$$\limsup_{\delta \to 0^+} \sup_{c \geq 1} \limsup_{n} \sup_{0 \leq t_2 - t_1 \leq \delta} \left\| \sum_{i=m(T_n+t_1)}^{m(T_n+t_2)-1} \alpha(i) H_c(0, W_{i+1}) \right\|$$

$$\leq \limsup_{\delta \to 0^+} \sup_{c \geq 1} \limsup_{n} \sup_{0 \leq t_2 - t_1 \leq \delta} \left\| \sum_{i=m(T_n+t_1)}^{m(T_n+t_2)-1} \alpha(i) \left[ H_c(0, W_{i+1}) - h_c(0) \right] \right\|$$

$$+ \limsup_{\delta \to 0^+} \sup_{c \geq 1} \limsup_{n} \sup_{0 \leq t_2 - t_1 \leq \delta} \left\| \sum_{i=m(T_n+t_1)}^{m(T_n+t_2)-1} \alpha(i) h_c(0) \right\|$$

$$\leq 0 + \limsup_{\delta \to 0^+} \sup_{c \geq 1} \limsup_{n} \sup_{0 \leq t_2 - t_1 \leq \delta} \left\| \sum_{i=m(T_n+t_1)}^{m(T_n+t_2)-1} \alpha(i) h_c(0) \right\| \qquad \text{(by (97))}$$

$$\leq 0 + \limsup_{\delta \to 0^+} \sup_{c \geq 1} \limsup_{n} \sup_{0 \leq t_2 - t_1 \leq \delta} \left\| \sum_{i=m(T_n+t_1)}^{m(T_n+t_2)-1} \alpha(i) \frac{h(0)}{c} \right\|$$

$$\leq 0 + \|h(0)\| \limsup_{\delta \to 0^+} \sup_{c \geq 1} \frac{1}{c} \limsup_{n} \sup_{0 \leq t_2 - t_1 \leq \delta} \sum_{i=m(T_n+t_1)}^{m(T_n+t_2)-1} \alpha(i)$$

$$\leq \|h(0)\| \limsup_{\delta \to 0^+} \sup_{c \geq 1} \frac{1}{c} \delta \qquad \text{(by (84))}$$

$$= \|h(0)\| \lim_{\delta \to 0^+} \delta$$

$$= 0.$$

$$\square$$

**Lemma 17.4.**

$$\sup_n \sup_{0 \le t_1 \le t_2 \le T_{n+1} - T_n} \left\| \sum_{i=m(T_n+t_1)}^{m(T_n+t_2)-1} \alpha(i) L(W_{i+1}) \right\| < \infty \quad a.s., \tag{99}$$

$$\lim_{n \to \infty} \sup_{0 \le t_1 \le t_2 \le T_{n+1} - T_n} \left\| \sum_{i=m(T_n+t_1)}^{m(T_n+t_2)-1} \beta(i) L(W_{i+1}) \right\| = 0 \quad a.s., \tag{100}$$

$$\lim_{\delta \to 0^+} \limsup_n \sup_{0 \le t_2 - t_1 \le \delta} \left\| \sum_{i=m(T_n+t_1)}^{m(T_n+t_2)-1} \alpha(i) L(W_{i+1}) \right\| = 0 \quad a.s., \tag{101}$$

$$\sup_n \sup_{0 \le t_1 \le t_2 \le T_{n+1} - T_n} \left\| \sum_{i=m(T_n+t_1)}^{m(T_n+t_2)-1} \alpha(i) L_b(W_{i+1}) \right\| < \infty \quad a.s. \tag{102}$$

These proofs are similar to the proofs of Lemma 17.3 and are thus omitted, except for the proof of (100) (since this is a two-timescale novelty and we need it for Lemma 12.2). For the other statements, we can also replace $\alpha(i)$ with $\beta(i)$.

*Proof.* Here we show 100:

$$\lim_n \sup_{0 \le t_1 \le t_2 \le T_{n+1} - T_n} \left\| \sum_{i=m(T_n+t_1)}^{m(T_n+t_2)-1} \beta(i) L(W_{i+1}) \right\|$$

$$= \lim_n \sup_{0 \le t_1 \le t_2 \le T_{n+1} - T_n} \left\| \sum_{i=m(T_n+t_1)}^{m(T_n+t_2)-1} \beta(i) [L(W_{i+1}) - L + L] \right\|$$

$$\le \lim_n \sup_{0 \le t_1 \le t_2 \le T_{n+1} - T_n} \left\| \sum_{i=m(T_n+t_1)}^{m(T_n+t_2)-1} \beta(i) [L(W_{i+1}) - L] \right\|$$

$$+ \lim_n \sup_{0 \le t_1 \le t_2 \le T_{n+1} - T_n} \left\| \sum_{i=m(T_n+t_1)}^{m(T_n+t_2)-1} \beta(i) L \right\|$$

$$\le \lim_n \sup_{0 \le t_1 \le t_2 \le T + \sup_j \alpha(j)} \left\| \sum_{i=m(T_n+t_1)}^{m(T_n+t_2)-1} \beta(i) [L(W_{i+1}) - L] \right\|$$

$$+ \lim_n \sup_{0 \le t_1 \le t_2 \le T + \sup_j \alpha(j)} \left\| \sum_{i=m(T_n+t_1)}^{m(T_n+t_2)-1} \beta(i) L \right\| \quad (\forall n, T_{n+1} - T_n \le T + \sup_j \alpha(j))$$

$$= \lim_n \sup_{0 \le t_1 \le t_2 \le T + \sup_j \alpha(j)} \left\| \sum_{i=m(T_n+t_1)}^{m(T_n+t_2)-1} \beta(i) L \right\|$$

$$= \|L\| \lim_n \sup_{0 \le t_1 \le t_2 \le T + \sup_j \alpha(j)} \sum_{i=m(T_n+t_1)}^{m(T_n+t_2)-1} \beta(i)$$

$$= 0. \quad \text{(by Lemma 17.1)}$$

$\square$

**Lemma 17.5.** *Fix a sample path $\{x_0, y_0, \{W_i\}_{i=1}^{\infty}\}$, there exist constants $C_H, C_G$ such that*

$$LT \leq C_H, LT \leq C_G \tag{103}$$

$$\sup_{c \geq 1} \|h_c(0)\| \leq \frac{C_H}{T}, \sup_{c \geq 1} \|g_c(0)\| \leq \frac{C_G}{T} \tag{104}$$

$$\sup_{c \geq 1} \sup_{n} \sup_{0 \leq t_1 \leq t_2 \leq T_{n+1}-T_n} \left\| \sum_{i=m(T_n+t_1)}^{m(T_n+t_2)-1} \alpha(i) H_c(0, W_{i+1}) \right\| \leq C_H, \tag{105}$$

$$\sup_{c \geq 1} \sup_{n} \sup_{0 \leq t_1 \leq t_2 \leq T_{n+1}-T_n} \left\| \sum_{i=m(T_n+t_1)}^{m(T_n+t_2)-1} \beta(i) G_c(0, W_{i+1}) \right\| \leq C_G,$$

$$\sup_{n} \sup_{0 \leq t_1 \leq t_2 \leq T_{n+1}-T_n} \sum_{i=m(T_n+t_1)}^{m(T_n+t_2)-1} \alpha(i) L(W_{i+1}) \leq C_H. \tag{106}$$

*We can replace $\alpha(i)$ with $\beta(i)$ in the last statement. Moreover, for convenience of presentation, we denote*

$$C_{\hat{x}} \doteq [1 + C_H] e^{C_H}, C_{\hat{y}} \doteq [1 + C_G] e^{C_G}, C' \doteq C_G + C_H, C_{\hat{z}} \doteq [1 + C'] e^{C'} \tag{107}$$

*Proof.* Fix a sample path $\{x_0, y_0, \{W_i\}_{i=1}^{\infty}\}$,

$$LT < \infty, \qquad\qquad (L \text{ and } T \text{ are constants})$$

$$\sup_{c \geq 1} \|h_c(0)\| T < \infty, \qquad\qquad (\text{by (96)})$$

$$\sup_{c \geq 1} \sup_{n} \sup_{0 \leq t_1 \leq t_2 \leq T_{n+1}-T_n} \left\| \sum_{i=m(T_n+t_1)}^{m(T_n+t_2)-1} \alpha(i) H_c(0, W_{i+1}) \right\| < \infty, \qquad (\text{by (94)})$$

$$\sup_{n} \sup_{0 \leq t_1 \leq t_2 \leq T_{n+1}-T_n} \sum_{i=m(T_n+t_1)}^{m(T_n+t_2)-1} \alpha(i) L(W_{i+1}) < \infty. \qquad (\text{by (99)})$$

Thus, there exists a constant $C_H$ such that

$$LT \leq C_H$$

$$\sup_{c \geq 1} \|h_c(0)\| \leq \frac{C_H}{T},$$

$$\sup_{c \geq 1} \sup_{n} \sup_{0 \leq t_1 \leq t_2 \leq T_{n+1}-T_n} \left\| \sum_{i=m(T_n+t_1)}^{m(T_n+t_2)-1} \alpha(i) H_c(0, W_{i+1}) \right\| \leq C_H,$$

$$\sup_{n} \sup_{0 \leq t_1 \leq t_2 \leq T_{n+1}-T_n} \sum_{i=m(T_n+t_1)}^{m(T_n+t_2)-1} \alpha(i) L(W_{i+1}) \leq C_H.$$

$\square$

**Lemma 17.6.** $\sup_{n, t \in [0, T+1)} \|\tilde{z}_n(t)\| \leq C_{\hat{z}}$.

*Proof.* $\forall n \in \mathbb{N}, t \in [0, T+1)$,

$$
\|\tilde{z}_n(t)\|
$$

$$
= \left\| \tilde{z}_n(0) + \sum_{i=m(T_n)}^{m(T_n+t)-1} (\alpha(i) H_{r_n}(\tilde{z}_n(t(i) - T_n), W_{i+1}), \beta(i) G_{r_n}(\tilde{z}_n(t(i) - T_n), W_{i+1})) \right\|
$$

$$
\leq \|\tilde{z}_n(0)\| + \left\| \sum_{i=m(T_n)}^{m(T_n+t)-1} \alpha(i) \left[ H_{r_n}(\tilde{z}_n(t(i) - T_n), W_{i+1}) - H_{r_n}(0, W_{i+1}) \right] + \sum_{i=m(T_n)}^{m(T_n+t)-1} \alpha(i) H_{r_n}(0, W_{i+1}) \right\|
$$

$$
+ \left\| \sum_{i=m(T_n)}^{m(T_n+t)-1} \beta(i) \left[ G_{r_n}(\tilde{z}_n(t(i) - T_n), W_{i+1}) - G_{r_n}(0, W_{i+1}) \right] + \sum_{i=m(T_n)}^{m(T_n+t)-1} \beta(i) G_{r_n}(0, W_{i+1}) \right\|
$$

$$
\leq \|\tilde{z}_n(0)\| + \sum_{i=m(T_n)}^{m(T_n+t)-1} \alpha(i) \| H_{r_n}(\tilde{z}_n(t(i) - T_n), W_{i+1}) - H_{r_n}(0, W_{i+1}) \| + C_H
$$

$$
+ \sum_{i=m(T_n)}^{m(T_n+t)-1} \beta(i) \| G_{r_n}(\tilde{z}_n(t(i) - T_n), W_{i+1}) - G_{r_n}(0, W_{i+1}) \| + C_G
$$

$$
\leq \|\tilde{z}_n(0)\| + \sum_{i=m(T_n)}^{m(T_n+t)-1} (\alpha(i) + \beta(i)) L(W_{i+1}) \|\tilde{z}_n(t(i) - T_n)\| + C' \qquad \text{(by (105))}
$$

$$
\leq 1 + \sum_{i=m(T_n)}^{m(T_n+t)-1} (\alpha(i) + \beta(i)) L(W_{i+1}) \|\tilde{z}_n(t(i) - T_n)\| + C' \qquad \text{(by (19))}
$$

$$
\leq [1 + C'] e^{\sum_{i=m(T_n)}^{m(T_n+t)-1} (\alpha(i)+\beta(i)) L(W_{i+1})}
$$
$$
\text{(by } \tilde{z}_n(t) = \tilde{z}_n(t(m(T_n + t)) - T_n) \text{ and discrete Gronwall inequality in Theorem 10.3)}
$$

$$
\leq [1 + C'] e^{C'} \qquad \text{(by (106))}
$$
$$
= C_{\hat{z}}. \qquad \text{(by (107))}
$$

□

**Lemma 17.7.** $\sup_{n, t \in [0, T+1)} \|z_n(t)\| \leq C_{\hat{x}}$.

*Proof.* $\forall n, t \in [0, T+1)$,

$$\|z_n(t)\|$$

$$= \left\| z_n(0) + \int_0^t (h_{r_n}(z_n(s)), 0) ds \right\|$$

$$\leq \|z_n(0)\| + \left\| \int_0^t h_{r_n}(z_n(s)) ds \right\|$$

$$\leq \|z_n(0)\| + \int_0^t \|h_{r_n}(z_n(s)) - h_{r_n}(0)\| ds + \int_0^t \|h_{r_n}(0)\| ds$$

$$\leq \|z_n(0)\| + \int_0^t L\|z_n(s)\| ds + \int_0^t \|h_{r_n}(0)\| ds \qquad \text{(by Lemma 17.2)}$$

$$\leq \|z_n(0)\| + \int_0^t L\|z_n(s)\| ds + (T+1)\|h_{r_n}(0)\|$$

$$\leq \|z_n(0)\| + \int_0^t L\|z_n(s)\| ds + (T+1)\frac{C_H}{T+1} \qquad \text{(by (104))}$$

$$\leq 1 + \int_0^t L\|z_n(s)\| ds + C_H \qquad \text{(by (19), (20))}$$

$$\leq [1 + C_H] e^{L(T+1)} \qquad \text{(by Gronwall inequality in Theorem 10.1)}$$

$$\leq [1 + C_H] e^{C_H} \qquad \text{(by (103))}$$

$$= C_{\hat{x}} \qquad \text{(by (107))}$$

$\square$

**Lemma 17.8.** $\sup_{n,t \in [0,T+1)} \|h_{r_n}(z_n(t))\| < \infty$.

*Proof.* $\forall n, \forall t \in [0, T+1)$,

$$\|h_{r_n}(z_n(t))\|$$
$$\leq \|h_{r_n}(z_n(t)) - h_{r_n}(0)\| + \|h_{r_n}(0)\|$$
$$\leq L\|z_n(t)\| + \|h_{r_n}(0)\| \qquad \text{(by Lemma 17.2)}$$
$$\leq LC_{\hat{x}} + \|h_{r_n}(0)\| \qquad \text{(by Lemma 17.7)}$$
$$\leq LC_{\hat{x}} + \frac{C_H}{T}. \qquad \text{(by (18) and (104))}$$

Thus, because $C_{\hat{x}}, C_H$ are independent of $n, t$, $\sup_{n,t \in [0,T+1)} \|h_{r_n}(z_n(t))\| < \infty$.

$\square$

**Lemma 17.9.** $\sup_{t \in [0,T+1)} \left\| z^{\lim}(t) \right\| \leq C_{\hat{x}}$.

*Proof.* $\forall t \in [0, T+1)$,

$$\left\|z^{\lim}(t)\right\|$$

$$=\left\|z^{\lim}(0) + \int_0^t (h_\infty(z^{\lim}(s)), 0)ds\right\|$$

$$\leq \left\|z^{\lim}(0)\right\| + \left\|\int_0^t h_\infty(z^{\lim}(s))ds\right\|$$

$$=\left\|z^{\lim}(0)\right\| + \left\|\int_0^t \left[h_\infty(z^{\lim}(s)) - h_\infty(0)\right] ds + \int_0^t h_\infty(0)ds\right\|$$

$$\leq \left\|z^{\lim}(0)\right\| + \int_0^t \left\|h_\infty(z^{\lim}(s)) - h_\infty(0)\right\|ds + \int_0^t \|h_\infty(0)\|ds$$

$$\leq \left\|z^{\lim}(0)\right\| + \int_0^t L\left\|z^{\lim}(s)\right\|ds + \int_0^t \|h_\infty(0)\|ds \qquad \text{(by Lemma 17.2)}$$

$$\leq 1 + \int_0^t L\left\|z^{\lim}(s)\right\|ds + \int_0^t \|h_\infty(0)\|ds \qquad \text{(by (19), (20))}$$

$$\leq 1 + \int_0^t L\left\|z^{\lim}(s)\right\|ds + (T+1)\|h_\infty(0)\|$$

$$\leq 1 + \int_0^t L\left\|z^{\lim}(s)\right\|ds + C_H \qquad \text{(by Assumption 5 and (104))}$$

$$\leq [1 + C_H] e^{\int_0^t L ds} \qquad \text{(by Gronwall inequality in Theorem 10.1)}$$

$$\leq [1 + C_H] e^{L(T+1)}$$

$$\leq C_{\hat{x}}. \qquad \text{(by (103), (107))}$$

$\square$

**Lemma 17.10.** $\lim_{k\to\infty} h_{r_{n_k}}(z^{\lim}(t)) = h_\infty(z^{\lim}(t))$ *uniformly in* $t \in [0, T+1)$.

*Proof.* By Assumption 5, $\lim_{k\to\infty} h_{r_{n_k}}(v) = h_\infty(v)$ uniformly in a compact set $\{v | v \in \mathbb{R}^d, \|v\| \leq C_x\}$. By Lemma 17.9, $\{z^{\lim}(t) | t \in [0, T+1)\} \subseteq \{v | v \in \mathbb{R}^d, \|v\| \leq C_x\}$. Therefore, $\lim_{k\to\infty} h_{r_{n_k}}(z^{\lim}(t)) = h_\infty(z^{\lim}(t))$ uniformly in $\{z^{\lim}(t) | t \in [0, T+1)\}$ and in $t \in [0, T+1)$. $\square$

**Lemma 17.11.** $\forall t \in [0, T+1)$, *we have*

$$\lim_{k\to\infty} z_{n_k}(t) = z^{\lim}(t).$$

*Moreover, the convergence is uniform in* $t$ *on* $[0, T+1)$.

*Proof.* By (22), $\forall \delta > 0$, there exists a $k_1$ such that $\forall k \geq k_1, \forall t \in [0, T+1)$,

$$\left\|\tilde{z}_{n_k}(t) - \tilde{z}^{\lim}(t)\right\| \leq \delta. \qquad (108)$$

By Lemma 17.10, there exists a $k_2$ such that $\forall k \geq k_2, \forall t \in [0, T+1)$,

$$\left\|h_{r_{n_k}}(z^{\lim}(t)) - h_\infty(z^{\lim}(t))\right\| \leq \delta. \qquad (109)$$

$\forall k \geq \max\{k_1, k_2\}, \forall t \in [0, T+1)$

$$\left\| z_{n_k}(t) - z^{\lim}(t) \right\|$$

$$= \left\| \tilde{z}_{n_k}(0) + \int_0^t (h_{r_{n_k}}(z_{n_k}(s)), 0)ds - \tilde{z}^{\lim}(0) - \int_0^t (h_\infty(z^{\lim}(s)), 0)ds \right\|$$

$$\leq \left\| \tilde{z}_{n_k}(0) - \tilde{z}^{\lim}(0) \right\| + \left\| \int_0^t h_{r_{n_k}}(z_{n_k}(s))ds - \int_0^t h_\infty(z^{\lim}(s))ds \right\|$$

$$\leq \delta + \left\| \int_0^t h_{r_{n_k}}(z_{n_k}(s)) - h_\infty(z^{\lim}(s))ds \right\| \qquad \text{(by (108))}$$

$$\leq \delta + \int_0^t \left\| h_{r_{n_k}}(z_{n_k}(s)) - h_{r_{n_k}}(z^{\lim}(s)) \right\| ds + \int_0^t \left\| h_{r_{n_k}}(z^{\lim}(s)) - h_\infty(z^{\lim}(s)) \right\| ds$$

$$\leq \delta + L \int_0^t \left\| z_{n_k}(s) - z^{\lim}(s) \right\| ds + \int_0^t \left\| h_{r_{n_k}}(z^{\lim}(s)) - h_\infty(z^{\lim}(s)) \right\| ds \quad \text{(by Lemma 17.2)}$$

$$\leq \delta + t\delta + L \int_0^t \left\| z_{n_k}(s) - z^{\lim}(s) \right\| ds \qquad \text{(by (109))}$$

$$\leq (\delta + t\delta)e^{Lt} \qquad \text{(by Gronwall inequality in Theorem 10.1)}$$

$$\leq (\delta + (T+1)\delta)e^{L(T+1)},$$

which completes the proof. $\qquad \square$

**Lemma 17.12.** *For any function $f : \mathbb{R} \times \mathbb{R} \to \mathbb{R}$, if $\lim_{\substack{a \to \infty \\ b \to \infty}} f(a, b) = L$ then $\lim_{c \to \infty} f(c, c) = L$ where $L$ is a constant.*

*Proof.* By definition, $\forall \epsilon > 0, \exists a_0, b_0$ such that $\forall a > a_0, b > b_0, \|f(a, b) - L\| < \epsilon$. Thus, $\forall \epsilon > 0, \exists c_0 = \max\{a_0, b_0\}$ such that $\forall c > c_0, \|f(c, c) - L\| < \epsilon$. $\qquad \square$

**Lemma 17.13.** $\forall t \in [0, T+1)$,

$$\lim_{k \to \infty} \int_0^t h_{r_{n_k}}(\tilde{z}^{\lim}(s))ds = \int_0^t h_\infty(\tilde{z}^{\lim}(s))ds.$$

*Proof.* From Lemma 17.6, it is easy to see that

$$\sup_{t \in [0, T+1)} \left\| \tilde{z}^{\lim}(t) \right\| < \infty,$$

which, similar to Lemma 17.8, implies that

$$\sup_{k, t \in [0, T+1)} \left\| h_{r_{n_k}}\left( \tilde{z}^{\lim}(t) \right) \right\| < \infty.$$

By the dominated convergence theorem, $\forall t \in [0, T+1)$,

$$\lim_{k \to \infty} \int_0^t h_{r_{n_k}}(\tilde{z}^{\lim}(s))ds = \int_0^t \lim_{k \to \infty} h_{r_{n_k}}(\tilde{z}^{\lim}(s))ds = \int_0^t h_\infty(\tilde{z}^{\lim}(s))ds,$$

which completes the proof. $\qquad \square$

**Lemma 17.14.** $\forall t \in [0, T+1)$,

$$\lim_{k \to \infty} \int_0^t h_{r_{n_k}}(z_{n_k}(s))ds = \int_0^t h_\infty(z^{\lim}(s))ds.$$

*Proof.* $\forall \epsilon > 0$, by Lemma 17.10, $\exists k_0$ such that $\forall k \geq k_0, \forall t \in [0, T)$,

$$\left\| h_{r_{n_k}}(z^{\lim}(s)) - h_\infty(z^{\lim}(s)) \right\| \leq \epsilon. \qquad (110)$$

By Lemma 17.11, $\exists k_1$ such that $\forall k \geq k_1$, $\forall t \in [0, T+1)$,

$$\left\| z_{n_k}(t) - z^{\lim}(t) \right\| \leq \epsilon. \tag{111}$$

Thus, $\forall k \geq \max\{k_0, k_1\}$, $\forall t \in [0, T+1)$,

$$\left\| \int_0^t h_{r_{n_k}}(z_{n_k}(s)) ds - \int_0^t h_\infty(z^{\lim}(s)) ds \right\|$$

$$\leq \left\| \int_0^t h_{r_{n_k}}(z_{n_k}(s)) ds - \int_0^t h_{r_{n_k}}(z^{\lim}(s)) ds \right\| + \left\| \int_0^t h_{r_{n_k}}(z^{\lim}(s)) ds - \int_0^t h_\infty(z^{\lim}(s)) ds \right\|$$

$$\leq \int_0^t \left\| h_{r_{n_k}}(z_{n_k}(s)) - h_{r_{n_k}}(z^{\lim}(s)) \right\| ds + \int_0^t \left\| h_{r_{n_k}}(z^{\lim}(s)) - h_\infty(z^{\lim}(s)) \right\| ds$$

$$\leq \int_0^t \left\| h_{r_{n_k}}(z_{n_k}(s)) - h_{r_{n_k}}(z^{\lim}(s)) \right\| ds + (T+1)\epsilon \qquad \text{(by (110))}$$

$$\leq \int_0^t L \left\| z_{n_k}(s) - z^{\lim}(s) \right\| ds + T\epsilon \qquad \text{(by Lemma 17.2)}$$

$$\leq L(T+1)\epsilon + (T+1)\epsilon. \qquad \text{(by (111))}$$

Thus, $\forall t \in [0, T+1)$,

$$\lim_{k \to \infty} \int_0^t h_{r_{n_k}}(z_{n_k}(s)) ds = \int_0^t h_\infty(z^{\lim}(s)) ds.$$

$\square$

**Lemma 17.15.** $\forall n, \forall t \in [0, T_{n+1} - T_n]$,

$$\| \bar{z}(T_n + t) \| \leq \left( \| \bar{z}(T_n) \| C' + C' \right) e^{C'} + \| \bar{z}(T_n) \|,$$

*and in particular,*

$$\| \bar{z}(T_{n+1}) \| \leq \left( \| \bar{z}(T_n) \| C' + C' \right) e^{C'} + \| \bar{z}(T_n) \|$$

*where $C'$ is a positive constant defined in Lemma 17.5.*

*Proof.* We first show the difference between $\bar{z}(T_{n+1})$ and $\bar{z}(T_n)$ by the following derivations. $\forall n$, $\forall t \in [0, T_{n+1} - T_n]$,

$$\|\bar{z}(T_n + t) - \bar{z}(T_n)\|$$

$$=\|\bar{z}(t(m(T_n + t))) - \bar{z}(T_n)\|$$

$$=\left\|\bar{z}(T_n) + \sum_{i=m(T_n)}^{m(T_n+t)-1} (\alpha(i)H(\bar{z}(t(i)), W_{i+1}), \beta(i)G(\bar{z}(t(i)), W_{i+1})) - \bar{z}(T_n)\right\|$$

$$=\left\|\sum_{i=m(T_n)}^{m(T_n+t)-1} (\alpha(i)H(\bar{z}(t(i)), W_{i+1}), \beta(i)G(\bar{z}(t(i)), W_{i+1}))\right\|$$

$$\leq \sum_{i=m(T_n)}^{m(T_n+t)-1} \alpha(i)\|H(\bar{z}(t(i)), W_{i+1}) - H(\bar{z}(T_n), W_{i+1})\| + \beta(i)\|G(\bar{z}(t(i)), W_{i+1}) - G(\bar{z}(T_n), W_{i+1})\|$$

$$+ \left\|\sum_{i=m(T_n)}^{m(T_n+t)-1} \alpha(i)H(\bar{z}(T_n), W_{i+1})\right\| + \left\|\sum_{i=m(T_n)}^{m(T_n+t)-1} \beta(i)G(\bar{z}(T_n), W_{i+1})\right\|$$

$$\leq \sum_{i=m(T_n)}^{m(T_n+t)-1} (\alpha(i) + \beta(i))L(W_{i+1})\|\bar{z}(t(i)) - \bar{z}(T_n)\|$$

$$+ \left\|\sum_{i=m(T_n)}^{m(T_n+t)-1} \alpha(i)H(\bar{z}(T_n), W_{i+1})\right\| + \left\|\sum_{i=m(T_n)}^{m(T_n+t)-1} \beta(i)G(\bar{z}(T_n), W_{i+1})\right\|$$

$$\leq \sum_{i=m(T_n)}^{m(T_n+t)-1} (\alpha(i) + \beta(i))L(W_{i+1})\|\bar{z}(t(i)) - \bar{z}(T_n)\| + \sum_{i=m(T_n)}^{m(T_n+t)-1} (\alpha(i) + \beta(i))L(W_{i+1})\|\bar{z}(T_n)\|$$

$$+ \left\|\sum_{i=m(T_n)}^{m(T_n+t)-1} \alpha(i)H(0, W_{i+1})\right\| + \left\|\sum_{i=m(T_n)}^{m(T_n+t)-1} \beta(i)G(0, W_{i+1})\right\| \qquad \text{(by Assumption 4)}$$

$$= \sum_{i=m(T_n)}^{m(T_n+t)-1} (\alpha(i) + \beta(i))L(W_{i+1})\|\bar{z}(t(i)) - \bar{z}(T_n)\| + \|\bar{z}(T_n)\| \sum_{i=m(T_n)}^{m(T_n+t)-1} (\alpha(i) + \beta(i))L(W_{i+1})$$

$$+ \left\|\sum_{i=m(T_n)}^{m(T_n+t)-1} \alpha(i)H(0, W_{i+1})\right\| + \left\|\sum_{i=m(T_n)}^{m(T_n+t)-1} \beta(i)G(0, W_{i+1})\right\|$$

$$\leq \sum_{i=m(T_n)}^{m(T_n+t)-1} (\alpha(i) + \beta(i))L(W_{i+1})\|\bar{z}(t(i)) - \bar{z}(T_n)\| + \|\bar{z}(T_n)\|C' + \left\|\sum_{i=m(T_n)}^{m(T_n+t)-1} \alpha(i)H(0, W_{i+1})\right\|$$

$$+ \left\|\sum_{i=m(T_n)}^{m(T_n+t)-1} \beta(i)G(0, W_{i+1})\right\| \qquad \text{(by (106))}$$

$$\leq \sum_{i=m(T_n)}^{m(T_n+t)-1} (\alpha(i) + \beta(i))L(W_{i+1})\|\bar{z}(t(i)) - \bar{z}(T_n)\| + [\|\bar{z}(T_n)\|C' + C'] \qquad \text{(by (105))}$$

$$\leq [\|\bar{z}(T_n)\|C' + C'] e^{\sum_{i=m(T_n)}^{m(T_n+t)-1} (\alpha(i)+\beta(i))L(W_{i+1})}$$

$$\text{(by discrete Gronwall inequality in Theorem 10.3)}$$

$$\leq [\|\bar{z}(T_n)\|C' + C'] e^{C'} \qquad \text{(by (106))}$$

