# OpenReview forum: "The Stability and Convergence of Two-Timescale Stochastic Approximation with Markovian Noise for Reinforcement Learning"
_ICLR.cc/2026/Conference — Submitted to ICLR 2026_

### Official Review · Reviewer_gcT6 · 2025-10-29

**Soundness:** 3
**Presentation:** 3
**Contribution:** 3
**Rating:** 6
**Confidence:** 3

**Summary:**

This paper establishes theoretical guarantees for stability and convergence of two-timescale stochastic approximation (SA) under Markovian noises. It then applied the results to temporal difference learning with gradient correction (TDC)

**Strengths:**

S1. The major strength is to establish boundedness of iterates for general two-timescale SA with Markovian noise, without projections or compactness assumptions (Theorem 2).

S2. It further allows for establishing almost sure convergence of two-timescale SA in unbounded state spaces (Corollary 1)

S3. The general framework is further applied to TDC for establishing convergence.

**Weaknesses:**

W1. The main weakness is that there is no experiment to validate any results.

W2. Only asymptotic results are given, and less is discussed for discretization error.

**Questions:**

Q1. Can the authors illustrate more on the technical difference in Markovian and iid noise case? Intuitively if the Markovian runs into stationary distribution, and during the transition time the iterates stay bounded, does the analysis links with iid noise case?

---

> ### Author Response · Authors · 2025-11-23
>
> We appreciate the reviewer’s comments and are pleased to see the reviewer has identified our key contributions of establishing stability and convergence of two-timescale stochastic approximation with Markovian noise and the application to TDC with eligibility traces. We will now address the concerns.
>
> >The main weakness is that there is no experiment to validate any results. Only asymptotic results are given, and less is discussed for discretization error.
>
> We thank the reviewer for their perspective. The scope of this work focuses on the theory of stochastic approximation (often in the SA theory literature experiments are not conducted, see e.g. Liu et al. 2025, Borkar 2009) and asymptotic results specifically as even this is a large gap in the two-timescale stochastic approximation literature when the noise is Markovian, and we leave results about convergence rate to future work. We can add some more detail on the discretization error, particularly in the slow timescale where there is more novelty in our approach.
>
> >Can the authors illustrate more on the technical difference in Markovian and iid noise case? Intuitively if the Markovian runs into stationary distribution, and during the transition time the iterates stay bounded, does the analysis links with iid noise case?
>
> We appreciate the chance to highlight how the Markovian case is different from the iid case. Even if the chain mixes, the key difference is that in the Markovian case, sampling is not independent while it always is in the iid case. So we can’t just treat it like Martingale difference noise like we could in the iid case, which is why the machinery required for the analysis of the Markovian case isn’t just a simple extension of the existing methods for addressing the iid case.
>
> We again thank the reviewer for their time, hope we have adequately addressed the concerns, and would be glad to continue the discussion if the reviewer would find it helpful.

---

### Official Review · Reviewer_Cryz · 2025-10-31

**Soundness:** 2
**Presentation:** 2
**Contribution:** 2
**Rating:** 2
**Confidence:** 3

**Summary:**

The paper claims to present the first general stability and convergence results for two-timescale stochastic approximation (SA) schemes under Markovian noise, and to apply these to prove the convergence of the Temporal Difference with Correction (TDC) algorithm with eligibility traces. The work positions itself as filling a long-standing theoretical gap in stochastic approximation theory for reinforcement learning (RL). It uses the ODE method to derive results without requiring projections or compact noise spaces.

**Strengths:**

The manuscript is logically organized, with clear sections on introduction, related works, assumptions, main results and appendices containing proofs.

**Weaknesses:**

1. Some sections copy the material from [Liu et al, 2025] with minor cosmetic changes. For instance,
Sections 12.1-12.2 are almost the same as the material by [Liu et al, 2025] on page 10
Section 12.4 is a less detailed version of Section 4.4 in [Liu et al, 2025]
2. Some of proofs (e.g. Lemmas 12.3, 12.4) coincide with the ones in [Section 3.2, Borkar, 2009] modulo cosmetic changes (see Lemmas 1, 2 in Section 3.2)
3. No simulations or synthetic examples are provided to show the implications of the results. The absence of any such illustration makes it difficult to assess practical relevance.
4. The assumptions are described as “mild, minimal” but in practice, they are restrictive (unique stationary distribution, Lipschitz continuity, globally asymptotically stable equilibria). The theoretical claims are therefore less general than the rhetoric suggests.
Table 1 serves as a key motivation, though the discussion would benefit from a clearer explanation of why earlier Markovian SA results (e.g., Karmakar & Bhatnagar 2018, 2021) cannot be straightforwardly extended.

**Questions:**

1. Several sections (e.g., Sections 12.1, 12.2, 12.4) appear to closely follow the structure and content of Liu et al. (2025) and Borkar (2009). Could the authors clarify which parts of the analysis are genuinely novel and what specific methodological innovations distinguish this work from those prior results?
2. The assumptions (e.g., unique stationary distribution, Lipschitz continuity, and globally asymptotically stable equilibria) are described as “mild, minimal” but seem rather restrictive in realistic reinforcement learning settings. Could the authors discuss how sensitive their results are to these assumptions, and whether they can be relaxed in future work?
3. Can you provide an explicit example (even synthetic) of a two-timescale Markovian SA scheme where your results apply but all prior results fail?
4. Would the authors consider including a simple numerical example or experiment to illustrate the stability and convergence behavior implied by the theory?
5. Table 1 is used as a central motivation, but it remains somewhat unclear why earlier Markovian SA results cannot be directly adapted to the present setting. Could the authors elaborate on the specific technical challenges or assumptions that prevent such adaptation?
6. Why is projection-based stability considered fundamentally inferior here? In practice, most RL implementations employ normalization, clipping, or other mechanisms that implicitly act as projections, mitigating divergence issues even without formal projection steps.
7. Could the results be extended to stochastic approximation with function approximation beyond linear settings, or to non-Lipschitz dynamics?
8. How do your results compare to those of Liu et al. (2025) in terms of convergence rate guarantees?

**Details Of Ethics Concerns:**

-

---

> ### Author Response · Authors · 2025-11-23
>
> We appreciate the reviewer’s thorough examination of our work and respond to all the points below.
>
> >Some sections copy the material from [Liu et al, 2025] with minor cosmetic changes. For instance, Sections 12.1-12.2 are almost the same as the material by [Liu et al, 2025] on page 10 Section 12.4 is a less detailed version of Section 4.4 in [Liu et al, 2025]
> Some of proofs (e.g. Lemmas 12.3, 12.4) coincide with the ones in [Section 3.2, Borkar, 2009] modulo cosmetic changes (see Lemmas 1, 2 in Section 3.2).
>
> We thank the reviewer for raising this concern. In our work, there are some significant differences in the problems we solve (two-timescale versus single-timescale, Markovian versus iid) and the methods we use compared to prior work, but there are sections where we can invoke prior lemmas. In appendix 16.7 and 16.8, we mention those lemmas and proofs are similar to Borkar, but we did not mention it in 12.3 and 12.4 where the lemmas are stated. We will more thoroughly ensure that these are annotated and clarified. Those sections which are very similar to those from Liu et al. are included for completeness. There are some places where our steps are very similar and others where some steps and expressions are different (e.g. extra terms), e.g. Lemma 12.2 deals with the novel term in the discretization error in the fast timescale but the other terms similar to Liu et al. However, these only appear in the appendix and not in the main text. Nevertheless, we appreciate the reviewer’s attention to this matter and will better clarify which results are wholly original and which come from prior work.
>
> >No simulations or synthetic examples are provided to show the implications of the results. The absence of any such illustration makes it difficult to assess practical relevance.
> >Would the authors consider including a simple numerical example or experiment to illustrate the stability and convergence behavior implied by the theory?
>
> We appreciate the reviewer providing their concern on practical relevance. The scope of this work focuses on the theory of stochastic approximation (often in the SA theory literature experiments are not conducted, see e.g. Liu et al. 2025, Borkar 2009), so no simulations were run. For practical relevance, we would point to the proof of the convergence of TDC with traces. We also refer to this Sutton paper: https://icml.cc/Conferences/2009/papers/546.pdf, where TDC (albeit with no traces) was first presented and has experiments showing that it seems to converge. We do want our work to be useful to the community and appreciate the reviewer bringing this up.
>
> >The assumptions are described as “mild, minimal” but in practice, they are restrictive (unique stationary distribution, Lipschitz continuity, globally asymptotically stable equilibria). The theoretical claims are therefore less general than the rhetoric suggests. Table 1 serves as a key motivation, though the discussion would benefit from a clearer explanation of why earlier Markovian SA results (e.g., Karmakar & Bhatnagar 2018, 2021) cannot be straightforwardly extended.
>
>
> We thank the reviewer for bringing this up. The assumptions are no more restrictive than the previous work in the theory literature (see Lakshminarayanan and Bhatnagar 2017, Liu et al 2025); even with the same or weaker assumptions we are able to get stronger results. However, what is more important is the fact that these assumptions are verifiable, unlike the common assumption that the iterates are bounded almost surely, which is not (see e.g., Borkar 1997, 2009; Tadic 2004). This allows us to prove that TDC with traces converges without needing to just assume boundedness or resorting to projections to guarantee boundedness. We will make this point clearer in the revision.
> The reason the earlier Markovian SA results can’t be easily extended is that they don’t have the coupled dynamics where the fast and slow timescale affect each other, or where the noise must be in a compact space, so the methodology is developed for a different case and is not applicable (i.e. those restrictions are fundamental in those methods). We again thank the reviewer for bringing attention to the restrictiveness of the assumptions.

---

> > ### Author Response · Authors · 2025-11-23
> >
> > >Could the authors discuss how sensitive their results are to these assumptions, and whether they can be relaxed in future work?
> >
> > It is good of the reviewer to bring up how the assumptions may be relaxed. Some of the assumptions can be relaxed. For instance the first assumption, which concerns the existence of the invariant measure can be relaxed–see Liu et al. for a longer discussion on this. However, we chose to present the assumption this way since it is much easier to state and is commonly assumed. It also may be possible to relax the global asymptotically stable equilibria to local (if the algorithm doesn’t have any stable equilibria it likely is not a convergent algorithm). Relaxing Lipschitz continuity would likely be more challenging since there are many steps relying on applying the Gronwall inequality to a sum of terms bounded by the Lipschitz term.
> >
> > >Can you provide an explicit example (even synthetic) of a two-timescale Markovian SA scheme where your results apply but all prior results fail?
> >
> > We thank the reviewer for this question, but would kindly appreciate some clarification on what is meant by a synthetic example. The application to TDC with eligibility trace is precisely the explicit example we provide. It demonstrates a real two-timescale Markovian SA algorithm in the field of reinforcement learning where our results apply and all prior results fail. This is an important point for the reviewer to bring up, and this algorithm shows a specific, important case where our results apply and other results do not.
> >
> > >Why is projection-based stability considered fundamentally inferior here? In practice, most RL implementations employ normalization, clipping, or other mechanisms that implicitly act as projections, mitigating divergence issues even without formal projection steps.
> >
> > The reviewer brings up a great point about how in practice, many RL implementations implicitly rely on projections through the form of clipping and other methods. Unfortunately, projections can result in convergence to the incorrect value. TD methods should converge to the TD fixed point, but if you use a projected scheme, they may not. In section 5.4 of Borkar 2009, he discusses the theoretical complications and difficulties of using projected schemes in more detail; we will mention this in the revision. We appreciate the chance to clarify why projection-free schemes are preferable.
> >
> > >Could the results be extended to stochastic approximation with function approximation beyond linear settings, or to non-Lipschitz dynamics?
> >
> > This is a great question. The function approximation beyond linear settings should be possible; it may complicate the process of proving that a particular algorithm satisfies the assumptions, but there should be other types of function approximation besides linear approximation that would satisfy them. However, losing Lipschitzness would probably be a much more difficult hurdle to surmount due to the sheer number of steps throughout the analysis that rely on it.
> >
> > >How do your results compare to those of Liu et al. (2025) in terms of convergence rate guarantees?
> >
> > We appreciate the reviewer’s question. Neither Liu et al. nor this paper have any claims related to convergence rate–both are foremost concerned with asymptotic stability and convergence results. Convergence rates are certainly an important future step for better guarantees within the two-timescale SA theory, but it was important to fill the gap of asymptotic results.
> >
> > We hope we were able to adequately respond to all of the concerns, and we are happy to continue the discussion if the reviewer would find it to be of assistance.

---

### Official Review · Reviewer_bWeS · 2025-11-01

**Soundness:** 2
**Presentation:** 3
**Contribution:** 2
**Rating:** 4
**Confidence:** 3

**Summary:**

The paper develops stochastic approximation theorem for two-time scale method with Markovian noise. The authors establish a stability theorem, which shows the boundedness of the iterates, and then the convergence, which is followed by the asymptotic behavior of the corresponding ODE. The key novelty in the proof is argued to be a particular relation between the two iterates.

**Strengths:**

1. The authors provide a useful result the the community as two-time sclae based stochastic approximation with Markovian nosie is a widely considered scenario in the community. A convergence result for TDC is provided as an example, which shows the usefulness of the developed theorem.


2. The authors formalize a useful inequality (Theorem 1) which could be of independent interest,  aside from the stability or convergence result, which establishes fo relation between the two iterates.

**Weaknesses:**

1. The main concern is on the novelty of the proof and result of Theorem 1. The proof seems to follow that of Borkar and Meyn Theorem or Lakshminarayanan and Bhatnagar, defining the rescaled iterate to show boundedness. In particular the proof method of Lakshminarayanan and Bhatnagar also relies on the relation between the iterate $x_n$ and $y_n$. Therefore, as for now, it is not clear how the proof technique differ except for the difference of Markovian and i.i.d. case.

2. The presentation of the paper feels not friendly for several reasons. 1) The Lemmas are somewhat enumerated in serial manner, making it difficult to grasp the overall picture of the proof; 2) In Section 5, the authors assume $\sup_n r_n=\infty$ to obtain contradiction. But this is never mentioned again till the end of Section 6, which makes the overall picture of the proof unclear.

**Questions:**

1. In addition to the result of Lakshminarayanan and Bhatnagar, there has been quite a literature on two-time scale methods dealing with its non-asymptotic analysis. Considering such literature, is still the inequality in Theorem 1, can be considered a unique result?

2. How does the proof of showing the discreteization error of the rescaled iteration and its corresponding ODE going to zero differ from that of the stnadrad Borkar and Meyn Theorem?

---

> ### Author Response · Authors · 2025-11-23
>
> We thank the reviewer for the comments on our paper! We are encouraged to see that the results are useful since two-timescale stochastic approximation with Markovian noise is important to the community and that Theorem 1 is a useful result independently. We would now like to discuss the concerns:
>
> >The main concern is on the novelty of the proof and result of Theorem 1. The proof seems to follow that of Borkar and Meyn Theorem or Lakshminarayanan and Bhatnagar, defining the rescaled iterate to show boundedness. In particular the proof method of Lakshminarayanan and Bhatnagar also relies on the relation between the iterate
>  and
> . Therefore, as for now, it is not clear how the proof technique differ except for the difference of Markovian and i.i.d. Case.
>
> Thanks for bringing up this point. The relation between the iterates that was proved is different from Lakshminarayanan and Bhatnagar–Theorem 1 states that the fast iterate is controlled by the maximum of the previous slow iterates. This has consequences for the slow timescale analysis, like choosing the rescale factor $r_n$ in the slow timescale to be the maximum over the norms of all previous iterates, which is unique to our analysis, and the results required to link the timescales (Lemmas 5.7 and 6.1). While the general frame uses rescaled iterates like Borkar and Meyn, it has been heavily extended to tackle a much more general problem. Thanks again for mentioning this, we will make sure to better clarify the innovation in the proof technique over prior work.
>
> >The presentation of the paper feels not friendly for several reasons. 1) The Lemmas are somewhat enumerated in serial manner, making it difficult to grasp the overall picture of the proof; 2) In Section 5, the authors assume
>  to obtain contradiction. But this is never mentioned again till the end of Section 6, which makes the overall picture of the proof unclear.
>
> We appreciate the reviewer’s perspective on how the presentation could be improved. Since the work is complex, striking the balance between trying to be thorough by having line by line proofs for all the results and omitting enough to allow the reader to grasp the broad strokes has been challenging. We endeavored to minimize the number of lemmas stated in the main text so that the main thread was able to be understood, but this has caused our main text lemma numbering to be out of sync with the appendix, where the proofs all reside. We will work to improve this in the revision.
>
> As for the contradiction, we will improve the wording. Strictly speaking, in section 5, we assume $r_n$ goes to infinity because if it doesn’t the result is trivially true, so one must show that even if $r_n$ goes to infinity, the result holds. In section 6 it is accurate to say we assume $r_n$ goes to infinity for contradiction. We will clarify the language in both sections to lessen potential confusion–thanks for bringing this to our attention.
>
> >In addition to the result of Lakshminarayanan and Bhatnagar, there has been quite a literature on two-time scale methods dealing with its non-asymptotic analysis. Considering such literature, is still the inequality in Theorem 1, can be considered a unique result?
>
> We appreciate the reviewer bringing attention to the large literature on non-asymptotic analysis of two-timescale methods. We did examine the non-asymptotic analyses in the literature specifically to see if a result similar to Theorem 1 was present, and to our knowledge, Theorem 1 (and the idea of bounding the fast iterate with the maximal slow iterate seen so far) is still unique.

---

> > ### Author Response · Authors · 2025-11-23
> >
> > >How does the proof of showing the discreteization error of the rescaled iteration and its corresponding ODE going to zero differ from that of the stnadrad Borkar and Meyn Theorem?
> >
> > We thank the reviewer for allowing us to clarify where our work innovates. The two-timescale setting results in several differences from the Borkar and Meyn theorem.
> >
> > In Borkar and Meyn, they immediately show the discretization error diminishes across the entire sequence. Since we are using the Arzela-Ascoli theorem, we only immediately get convergence along a subsequence. But, due to the two-timescale setting (so unlike Liu et al.) we still need convergence along the whole sequence, which we get using a result from real analysis (Lemma 10.1).
> >
> > Another distinction concerns Lemma 5.7. The two-timescale setting also means that we have to deal with coupled ODEs since the timescale affect each other, fundamentally increasing the complexity of the dynamics we need to handle. So, we need some results about ODEs with external inputs. This takes some work to ensure some results concerning the uniform stability of these ODEs (including Lemmas 12.8 to 12.10).
> >
> > The final distinction we will highlight here concerns Lemma 6.1. This lemma, regarding how the timescales are linked, is a unique novelty of the two timescale case and its necessity results from the fact that we take r_n to be the max of all slow iterates seen previously (itself necessary due to what we prove in Theorem 1, namely that is relates the fast iterates to the maximum of the previous slow iterates).
> >
> > We again thank the reviewer for mentioning this concern. We will better emphasize our novel contributions in the revision to highlight the unique results.
> >
> > We hope our clarifications addressed all the reviewer’s concerns and are happy to participate in further discussion if it would be helpful.

---

### Official Review · Reviewer_FLRW · 2025-11-01

**Soundness:** 3
**Presentation:** 3
**Contribution:** 3
**Rating:** 6
**Confidence:** 4

**Summary:**

The paper analyzes two-timescale stochastic approximation with Markovian noise, providing stability (boundedness) and asymptotic convergence for general coupled updates without projections or compactness assumptions on the noise space. The framework targets RL settings with eligibility traces and off-policy importance ratios. As an application, the authors prove convergence of TDC with eligibility traces under standard conditions.

**Strengths:**

- **Closes a gap:** Prior Markov-noise analyses often require compactness or projections; this work handles unprojected iterates in non-compact noise spaces.
- First convergence proof (to my knowledge) for TDC with eligibility traces without projections.

**Weaknesses:**

There aren't any major weaknesses. However, a large part of the analysis seems to be following the standard existing analyses except for the part on the rescaling factor r_n. It would be helpful to explicitly explicitly articulate the analysis novelty versus existing proofs: isolate the steps that hinge on the rescaling factor.

Missing reference on multi-timescale SA: https://arxiv.org/pdf/2112.03515

**Questions:**

- Can you explicitly articulate the analysis novelty versus existing proofs, and isolate the steps that hinge on the rescaling factor to better unserstand the novelty in the paper wrt existing work

- Some typos: we now prove results on $t\in (\infty, \infty)$, in Assumption 2:  $n\rightarrow \infty$

---

> ### Author Response · Authors · 2025-11-23
>
> We appreciate the reviewer taking the time to examine our work and provide feedback. We are glad to see the reviewer recognized that our work closes a gap in the stochastic approximation theory and provides the first convergence proof of TDC with eligibility traces without projections. We address the concerns brought up below:
>
> >a large part of the analysis seems to be following the standard existing analyses except for the part on the rescaling factor r_n. It would be helpful to explicitly explicitly articulate the analysis novelty versus existing proofs: isolate the steps that hinge on the rescaling factor.
>
> Thanks for pointing out a way in which we can improve the presentation. The novelty includes the following. The result in Theorem 1 states the size of the fast iterates are controlled by the maximum of the previous slow iterates; that has a lot of consequences for the slow timescale proof being different. It requires some different treatment in the stability proof and in linking the timescales, in particular in the proof of Lemma 6.1 mentioned in the slow timescale stability section. We stated in the introduction that the use of controlling the size of the fast iterates with the maximum of the previous slow iterates is novel but we will emphasize this more and provide more pointers to those areas of novelty.
>
> >Missing reference on multi-timescale SA: https://arxiv.org/pdf/2112.03515
>
> Thanks for pointing this out! The reference extends the Lakshminarayanan and Bhatnagar analysis of two-timescale stability in the iid case to multi-timescale stability (i.e. greater than two timescales). We will add this reference in the revision; thanks again for the reference!
>
> >Some typos…
>
> Thanks for catching these! We will correct these typos in the revision.
>
> We appreciate the encouraging feedback, hope the clarifications were helpful, and are happy to discuss further if this would be beneficial.

---

### Meta-Review · Area_Chair_1ujw · 2026-01-16

**Summary:**

This paper focuses on two time scale stochastic approximation (SA) algorithms under Markovian noise which is a realistic model since such algorithms are widely used in RL where the noise is not iid. The authors are using the ODE framework to show asymptotic convergence guarantees as well as stability guarantees (boundedness of the iterates). The latter guarantee allows the authors to not require compact domains for the problem and projections. The application of the result is TD learning with gradient correction and an off-policy algorithm with linear approximation and elibility traces. Even though all the reviewers agree that the work fills a gap in the literature, the main concerns are, and remain to be, the unclarity with respect to the technical contribution of the work due to the lack of a proper positioning by the work. In particular, as Reviewer Cryz diligently noted, the authors have sections that are almost the same as Liu et al., 2025 without proper justification -- that is, without properly acknowledging they use parts of the text from this earlier paper and without making clear what the distinctions are. The authors respond by saying that this section is in the appendix but this is not satisfactory because this appendix section is indeed the proof of the main theorem, that is, Theorem 1. I saw more examples to this, for example Lemma 13.1 in the submission and Appendix B.2 in Liu et al., 2025. This section of the submission is related to the proof of the other main result, Theorem 2.

These, by themselves, don't need to be a reason for rejection, however the lack of proper explanations and positioning by the authors and issues with presentation lead me to recommend rejection. Moreover, the reviewers' questions about novelty are not explained well, the authors rebuttal mostly states generic statements such as "While the general frame uses rescaled iterates like Borkar and Meyn, it has been heavily extended to tackle a much more general problem". We need more precise and concrete explanations to be convinced of the novelty. It also doesn't help that the proofs are quite difficult to read in the 50-page appendix. For example, all the pages after page 43 contain expressions which are not polished and quite difficult to read. I recommend the authors to put more effort into making their proofs more readable and polished.

**Reviewer Concerns:**

Reviewer bWeS and Cryz had concerns about the novelty of the technical contribution compared to the work of Liu et al., 2025. Particularly, Reviewer Cryz also pointed out that parts of the text are almost the same as Liu et al., 2025 which is problematic. The authors provided rather generic responses to the questions about the novelty but I do not find these precise enough to address the concerns. I think the concerns of the reviewers about the lack of numerical experiments are addressed well since the authors pointed out that such experiments are not done in the particular sub-area their papers belong to.

**Reviewer Scores:**

As I stated before, the concerns of Reviewer bWeS and Cryz about the novelty and the comparison with Liu et al., 2025 are not addressed in my view. As a result, I think that these reviewers would not have changed their scores even with a longer discussion period.

Reviewer FLRW had issues with novelty and gcT6 had issues about numerics. I believe that these reviewers would keep their scores the same since they have scores of 6, which suggests weak acceptance with reservations. Especially the reservations of Reviewer FLRW about novelty are not addressed well as I mentioned above.

---

### Decision · Program_Chairs · 2026-01-26

Reject